# Organization of corticocortical and thalamocortical top-down inputs in the primary visual cortex

Yanmei Liu[1,2,4], Jiahe Zhang[1,2,4], Zhishan Jiang[1,2], Meiling Qin[3], Min Xu ®[3], Siyu Zhang ®[1,2] ✉ & Guofen Ma ®[1,2] ✉

Unified visual perception requires integration of bottom-up and top-down inputs in the primary visual cortex (V1), yet the organization of top-down inputs in V1 remains unclear. Here, we used optogenetics-assisted circuit mapping to identify how multiple top-down inputs from higher-order cortical and thalamic areas engage V1 excitatory and inhibitory neurons. Top-down inputs overlap in superficial layers yet segregate in deep layers. Inputs from the medial secondary visual cortex (V2M) and anterior cingulate cortex (ACA) converge on L6 Pyrs, whereas ventrolateral orbitofrontal cortex (ORBvl) and lateral posterior thalamic nucleus (LP) inputs are processed in parallel in Pyr-type-specific subnetworks (Pyr$_{\leftarrow ORBvl}$ and Pyr$_{\leftarrow LP}$) and drive mutual inhibition between them via local interneurons. Our study deepens understanding of the top-down modulation mechanisms of visual processing and establishes that V2M and ACA inputs in L6 employ integrated processing distinct from the parallel processing of LP and ORBvl inputs in L5.

The visual system is hierarchically organized, with functionally related areas connecting to each other in specific laminar patterns[1-4]. Processing of various dimensions of the complex visual environment, such as color, depth, shape, and motion, occurs in parallel pathways starting from the retina and continues in specialized visual areas, followed by integration in higher-order areas to form a unified perception[5-11]. This involves both bottom-up inputs, which flow from the retina to higher-order areas, and top-down modulation, where signals from higher-order areas adapt visual processing to meet the requirement of the current task[5-10].

V1 is the initial cortical area for visual information processing; it receives condensed and parallel bottom-up signals from the retino-geniculo-cortical pathways, extracts relevant information, and further elaborates and integrates this information with top-down inputs from higher-order cortical and thalamic areas, contributing to the formation of a unified perceptual experience[5,8,11–13]. In more detail, thalamocortical (TC) bottom-up inputs from the dorsal lateral geniculate thalamic

nucleus (dLGN) preferentially target the medial layer (L4), while corticocortical (CC) and TC top-down inputs target superficial and deep layers in V1[4,14-16].

Previous studies have identified multiple higher-order cortical and thalamic areas that provide top-down inputs to V1, such as the secondary visual cortex (V2), posterior parietal cortex (PTLp), retrosplenial cortex (RSP), anterior cingulate cortex (ACA), ventrolateral orbitofrontal cortex (ORBvl), and lateral dorsal (LD) and lateral posterior thalamic nuclei (LP)[4,15,17,18]. These top-down inputs convey a rich array of information, including attention[19-21], expectation[22,23], perceptual tasks[24,25], and motor commands[26,27].

The signals from these top-down inputs are differentially processed in V1[2,21,24,28–30]. For instance, V2 inputs are known to enhance the accuracy of visual information decoding in V1 neurons without altering the average response[31]. ACA and LP inputs increase V1 neuron response to task-relevant visual information[20,21], whereas ORBvl inputs suppress

[1]Songjiang Hospital and Songjiang Research Institute, Shanghai Key Laboratory of Emotions and Affective Disorders, Shanghai Jiao Tong University School of Medicine, Shanghai 201600, China. [2]Department of Anatomy and Physiology, Shanghai Jiao Tong University School of Medicine, Shanghai 200025, China. [3]Institute of Neuroscience, CAS Center for Excellence in Brain Science and Intelligence Technology, State Key Laboratory of Neuroscience, Chinese Academy of Sciences, Shanghai 200031, China. [4]These authors contributed equally: Yanmei Liu, Jiahe Zhang. ✉e-mail: zhang_siyu@sjtu.edu.cn; guofenma@sjtu.edu.cn

V1 neuron responses to filter out irrelevant visual information[24]. The diverse effects of these top-down modulations on visual processing imply sophisticated mechanisms within local circuits of V1, presumably controlled by distinct sets of excitatory and inhibitory neurons across different layers. However, knowledge about how these top-down inputs differentially engage excitatory and inhibitory neurons in V1 is quite limited, particularly with regard to their innervation patterns of neurons in deep layers.

The mouse V1 consists of six distinct layers, with each layer possessing unique connection and functional properties[7,8,13]. While the cell-sparse L1 completely lacks excitatory neurons and contains unique inhibitory neurons (L1-INs), all the other layers consist of excitatory Pyrs and three major types of inhibitory neurons (parvalbumin-positive, PV+; somatostatin-positive, SST+; and vasoactive intestinal peptide-positive, VIP+ neurons)[32–36]. The top-down modulation signals and bottom-up visual signals are processed and integrated into the V1 local circuits, and then transmitted to downstream targets by heterogeneous long-range projection Pyrs, including pyramidal tract (PT) and intratelencephalic (IT) neurons[4,7,37,38]. PT neurons mediate subcortical output, while IT neurons mediate intracortical output[4,39,40].

In this study, we profiled the layer- and cell-type-specific innervation patterns of multiple CC and TC top-down inputs in V1, including three CC inputs (V2M, ACA, and ORBvl) and one TC input (LP). We found distinct layer- and cell-type-specific innervation profiles for each top-down input, with profiles partially overlapping in superficial layers, bypassing L4, and clearly segregating in deep layers. Specifically, V2M and ACA inputs preferentially activate L6 Pyrs, while ORBvl and LP inputs activate L5 Pyrs. We also characterized the layer-specificity of top-down inputs on inhibitory neurons, revealing that L1-INs are strongly activated in L1 and that VIP+ neurons are strongly activated in both superficial and deep layers, while PV+ and SST+ neurons are specifically activated in the deep layers. These results provide a valuable resource for the layer- and cell-type-specific organization of top-down inputs in V1. We subsequently investigated how these inputs interact within their strongest receptive layers, L5 and L6. Using independent optogenetic activation on the same brain slice, we found that V2M and ACA inputs converge on L6 Pyrs, whereas ORBvl and LP inputs selectively activate two distinct types of L5 Pyrs: Pyr$_{\leftarrow ORBvl}$ and Pyr$_{\leftarrow LP}$ neurons, each characterized by specific electrophysiological properties and gene-expression profiles. Retrograde tracing revealed that Pyr$_{\leftarrow ORBvl}$ neurons preferentially innervate subcortical areas and Pyr$_{\leftarrow LP}$ neurons innervate cortical areas, indicating parallel processing of the ORBvl and LP inputs in Pyr-type-specific subnetworks. We also found that ORBvl and LP inputs drive mutual inhibition, mediated by local inhibitory neurons, between these two subnetworks in L5 of V1. These findings deepen our understanding of neuronal mechanisms of top-down modulation of visual processing by revealing interactions of modulation signals within V1 local circuits.

## Results

### Systematic characterization of the layer-specificities of CC and TC top-down inputs in V1 Pyrs

Multiple higher-order cortical and thalamic areas directly innervate V1, thus enabling extensive capacity for top-down modulation of visual processing[4,15–18]. The axons of CC and TC top-down inputs primarily target superficial and deep layers in V1[4]. Our research group has had a longstanding interest in top-down modulation of visual processing[17,18,21,41]; we found it conspicuous that most studies on the neuronal mechanisms of top-down modulation to date have focused on superficial layers (i.e., L1 and L2/3)[17,21,34,36,42]. We, therefore, undertook a systematic effort to characterize the innervation patterns of multiple CC and TC top-down inputs across all layers in V1, including both superficial and deep layers. Since the cell-sparse L1 completely lacks excitatory neurons, we started by examining the input strengths of V2M, ACA, ORBvl, and LP inputs in Pyrs across L2-L6 of V1.

To optogenetically activate V2M inputs in V1, we injected AAV expressing ChR2 in excitatory neurons (AAV-CaMKIIα-ChR2-EYFP) into the V2M (Fig. 1a, b). Expression of ChR2-EYFP in the V2M resulted in bright axonal fluorescence in L1 and L6 of V1, with L1 and L6 receiving 13 and 50% of the total fluorescent signal, respectively (Fig. 1c). We conducted whole-cell recordings of V1 Pyrs and recorded the monosynaptic excitatory postsynaptic potential (EPSP) elicited by optogenetic activation of V2M axons, while blocking local neuron spikes with TTX[43]. Consistent with a previous study[28], we found that V2M inputs activated Pyrs in L2/3 and L6 of V1 (defined as averaged EPSP amplitude >3 mV), with the strongest input strengths in L6 (Fig. 1d, e, Supplementary Figs. 1–3, and Supplementary Data 1, 2).

The distribution patterns of axons from ACA inputs (L1 received 22% and L6 received 49%) were similar to those of V2M inputs, as was their layer-specificity (Fig. 1f–j). Unlike V2M and ACA inputs, the axons from ORBvl inputs ramified densely in L5 (52% in L5), consistent with their strongest input strength in L5 Pyrs (Fig. 1k–o). The axons from LP inputs were primarily distributed in L1, L5, and L6 (15% in L1, 35% in L5, and 38% in L6), with the strongest input strength observed in L5 Pyrs (Fig. 1p–t). LP inputs also activate L6 Pyrs, but with weaker input strength as compared to L5 Pyrs (46% of the input strength in L5 Pyrs) (Fig. 1s, t). Together, when considering the input strengths of top-down inputs across all layers, our results show that the examined CC top-down inputs (V2M, ACA, and ORBvl) and TC top-down inputs (LP) inputs are partially overlapped in L2/3 with relatively weak input strength in L2/3 Pyrs. These inputs then bypass L4 Pyrs before segregating in L5 and L6: V2M and ACA inputs strongly activate L6 Pyrs (defined as averaged EPSP amplitude >7 mV), while ORBvl and LP inputs strongly activate L5 Pyrs.

### Layer-specificities of CC and TC top-down inputs in V1 inhibitory neurons

Despite representing a minority of cortical neurons, inhibitory neurons play a critical role in providing rapid and dynamic modulation of the output of excitatory neurons across different layers[32,34,36,39,44,45]. Besides unique inhibitory neurons in L1 (L1-INs), all the other layers are known to contain three major types of inhibitory neurons (PV+, SST+, and VIP+ neurons)[32,33,35]. We next examined the input patterns of multiple CC and TC top-down inputs in each type of inhibitory neurons. L1-INs were identified by their locations. PV+, SST+, and VIP+ neurons were identified by breeding PV-, SST-, or VIP-Cre mice with loxP-flanked tdTomato reporter mice (Ai14 mice, Supplementary Fig. 4). We observed distinct innervation patterns of cortical inhibitory neurons for each of CC and TC top-down input.

Optogenetic activation of V2M inputs resulted in strong activation in L1-INs within L1 and in strong activation in VIP+ and PV+ neurons in L6 ("strong activation" in inhibitory neurons also defined as averaged EPSP amplitude >7 mV) (Fig. 2a–d, Supplementary Fig. 4, and Supplementary Data 3). Activation of ACA inputs resulted in strong activation of L1-INs in L1, VIP+ neurons in L2/3, as well as in VIP+, PV+, and SST+ neurons in L6 (Fig. 2e–h). Notably, ORBvl inputs were weak in all types of inhibitory neurons across layers, although some L5 VIP+, PV+, and SST+ neurons were activated (ranging from 25 to 34% of each recorded inhibitory neuron type) (Fig. 2i–l and Supplementary Fig. 4). LP inputs strongly activated L1-INs in L1, VIP+ neurons from L2 to L6, and PV+ neurons in L5 (Fig. 2m–p).

Our profiling data for multiple CC (V2M, ACA, and ORBvl) and TC (LP) top-down inputs show that the innervation patterns of V1 inhibitory neurons are partially overlapped in superficial layers, with V2M, ACA, and LP inputs strongly activated L1-INs in L1, and ACA and LP inputs strongly activated VIP+ neurons in L2/3 (Fig. 2q, r). V2M and ACA inputs bypassed L4, whereas LP inputs strongly activated VIP+ neurons in L4. We also observed that the innervation patterns of top-down inputs were segregated in deep layers. L5 inhibitory neurons (VIP+ and PV+ neurons) were strongly activated by LP inputs

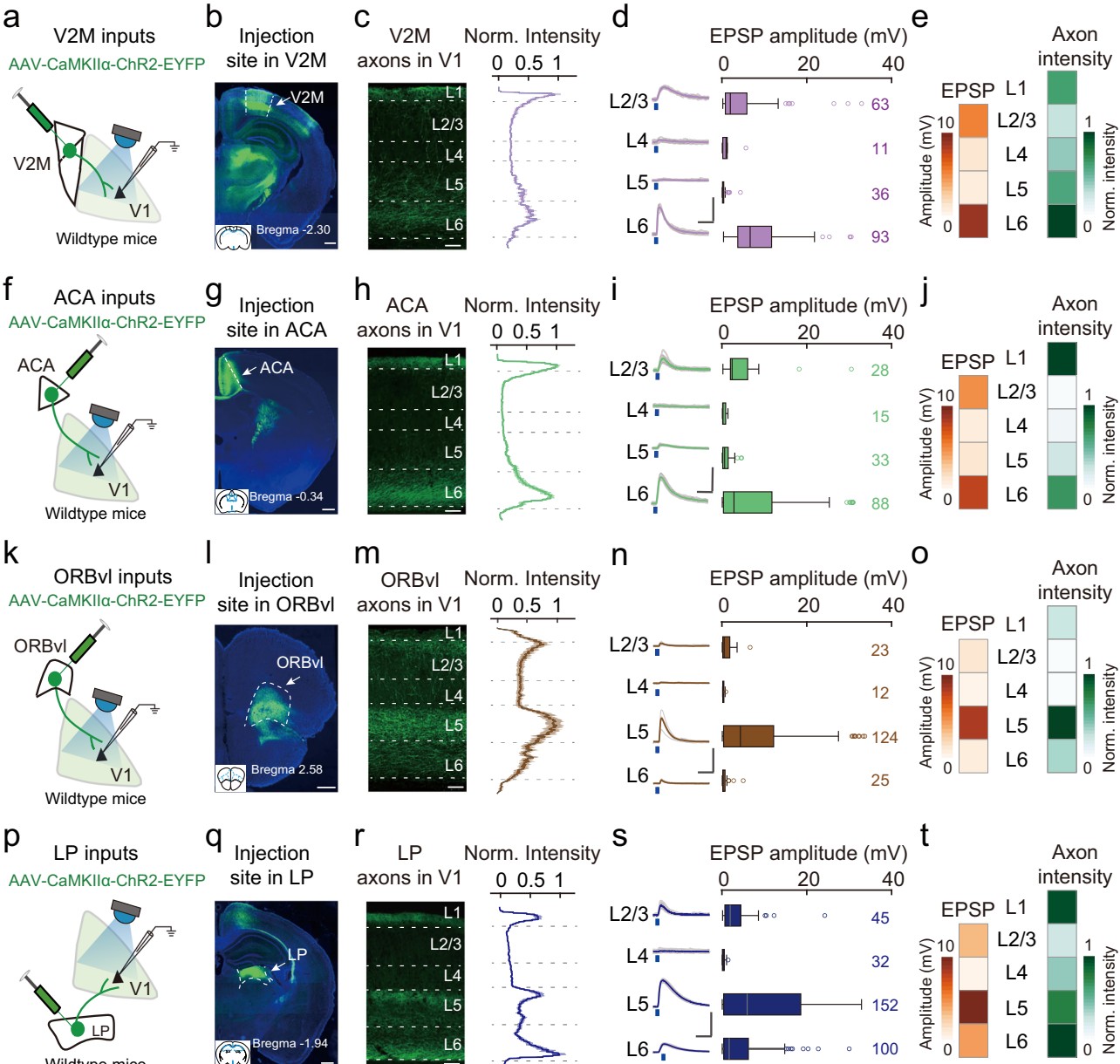

**Fig. 1 | Characterization of the strengths of CC and TC top-down inputs in V1 Pyrs across layers. a** Slice experiment schematic, with whole-cell recording of V1 Pyrs and optogenetic activation of V2M inputs. **b** Injection sites in the V2M. Scale bar, 500 μm. Arrowhead, AAV injection site. Blue, DAPI staining. Inset, coronal section location. **c** Distribution of V2M axons in V1. Left, fluorescence image showing V2M axons (green) in V1. Scale bar, 50 μm. Right, normalized green fluorescence intensity from L1 to L6 (normalized by the peak intensity in V1). **d** Monosynaptic EPSP amplitudes in Pyrs across layers. Left, monosynaptic EPSPs from one example Pyr in each layer with TTX and 4-AP treatment to block local neuron spiking (gray, raw traces; purple, averaged traces). Blue dots, 5-ms blue light stimulation (474 nm, 3.5 mW). Scale bars, 50 ms, 5 mV. Right, box plot for EPSP amplitude from Pyrs in the indicated layers. Edges, 25th and 75th percentiles; central line, median; whiskers, 1.5× the interquartile range of the edges. Black

circles are outliers (values more than three times the interquartile range (IQR) from the median). V2M inputs activated Pyrs in L2/3 and L6, with the strongest input strength in L6 (L6 vs. other layers, *P* < 0.004, Tukey's HSD test). **e** Matrix of EPSP amplitude (left) and normalized axon intensity (right) across V1 layers for V2M inputs. **f–j** Similar to (**a–e**), but for ACA inputs. ACA inputs activated Pyrs in L2/3 and L6, with stronger input strength in L6 (L6 vs. L4 and L5, *P* < 0.007; L6 vs. L2/3, *P* = 0.23). **k–o** Similar to (**a–e**), but for ORBvl inputs. ORBvl inputs only activated Pyrs in L5 (L5 vs. other layers, *P* < 0.02). **p–t** Similar to (**a–e**), but for LP inputs. LP inputs activated Pyrs in L2/3, L5, and L6, with the strongest input strength in L5 (L5 vs. other layers, *P* < 7 × 10⁻⁵). The number of neurons in each group is indicated. See Supplementary Data 1, 2 for input strengths and ANOVA parameters. Source data are provided as a Source Data file.

exclusively. L6 VIP+ neurons were strongly activated by V2M, ACA, and LP inputs, while L6 PV+ neurons were strongly activated by V2M and ACA inputs, and L6 SST+ neurons were strongly activated specifically by ACA inputs (Fig. 2q, r). Note that the layer-and cell-type specificity of these top-down inputs is evident based on both the raw and normalized input strengths (Supplementary Figs. 5, 6 and Supplementary Data 4–7). Thus, our profiling of layer- and cell-type-

specific innervation patterns of multiple CC and TC top-down inputs in V1 represents a valuable resource for understanding the neuronal mechanisms of top-down modulation of visual processing. Interestingly, in contrast to similar innervation properties in superficial layers, the examined CC and TC inputs are clearly segregated in deep layers, which opens up the possibility of testing hypotheses about their layer-specific functions.

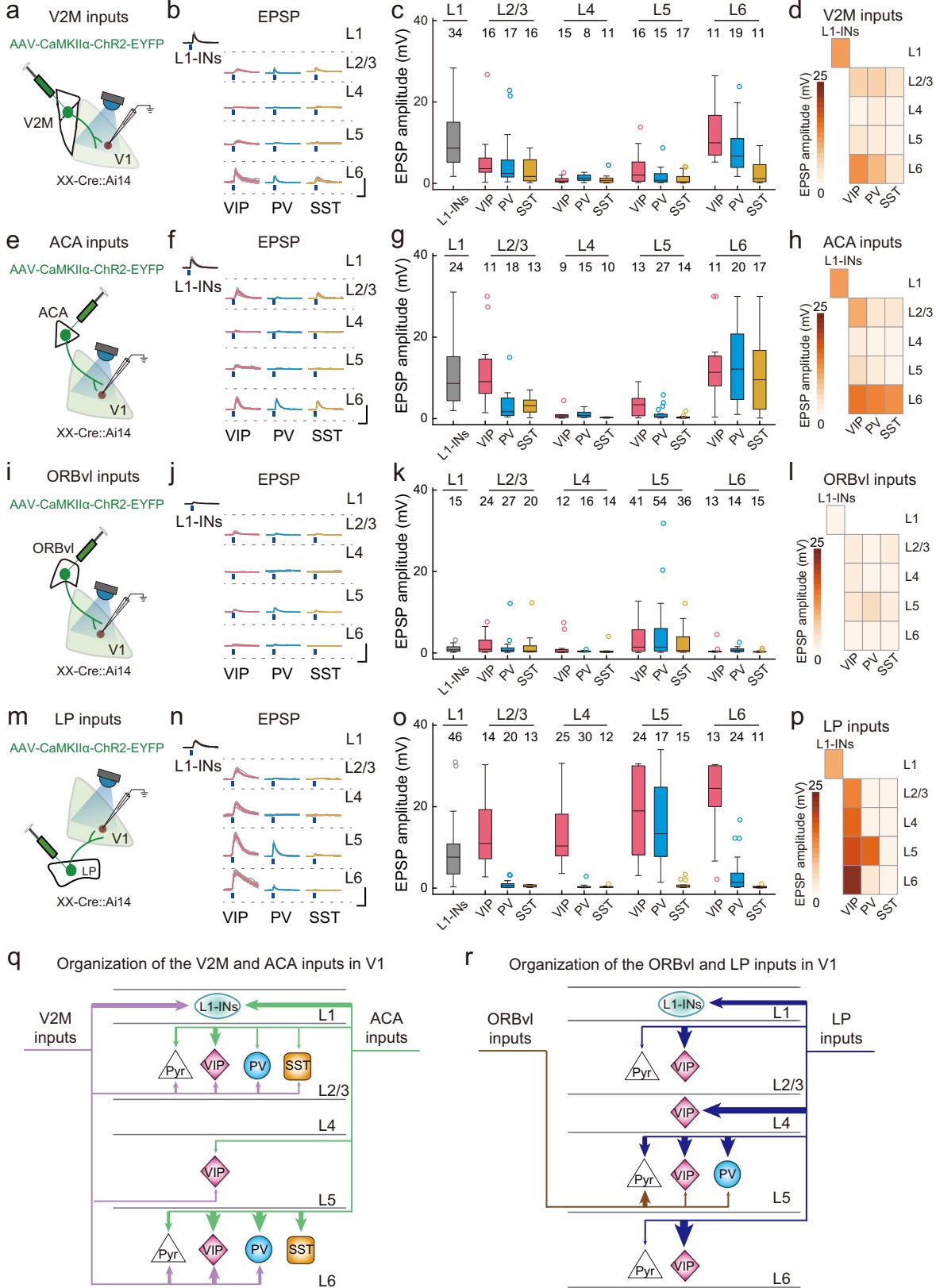

## Integrated processing of V2M and ACA inputs in L6 Pyrs versus parallel processing of LP and ORBvl inputs in two distinct L5 Pyr populations

The examined CC and TC top-down inputs are strongest in the deep layers, with L6 being most responsive to V2M and ACA inputs, while L5 is most responsive to LP and ORBvl inputs. Previous studies have

reported that deep-layer Pyrs are a heterogeneous population, with functionally distinct subnetworks[39,46–48]. To investigate the interactions among these top-down inputs in their most responsive layers, we employed independent optogenetic activation[49]. Specifically, for V2M and ACA inputs, we alternatively activated these inputs using ChR2 and ChrimsonR in the same brain slice and measured their input strengths

**Fig. 2 | Input strengths of CC and TC top-down inputs in four types of V1 inhibitory neurons across different layers. a** Schematic of the slice experiment, with whole-cell recording of V1 inhibitory neurons and optogenetic activation of V2M inputs. **b** Monosynaptic EPSPs from one example inhibitory neuron (L1-IN, VIP +, PV+, and SST+ neurons) in each layer with TTX and 4-AP treatment to block local neuron spiking (gray traces, raw traces; colored traces, averaged traces). Blue dots, 5-ms blue light stimulation (474 nm, 3.5 mW). Scale bars, 50 ms, 10 mV. **c** Distribution of EPSP amplitude from inhibitory neurons in different layers. Box plots indicate the median and the respective quartiles. Black dots are outliers, defined as values above 3 IQR from the median. V2M inputs strongly activated the L1-INs in L1 and VIP+ and PV+ neurons in L2/3 and L6. **d** Matrix of EPSP amplitude in different types of inhibitory neurons across layers of V1 for V2M inputs. **e–h** Similar

to (**a–d**), but for ACA inputs. **f** Scale bars, 50 ms, 10 mV. **g** ACA inputs strongly activated the L1-INs in L1, VIP+ neurons in L2/3 and L6, and PV+ and SST+ neurons in L6. **i–l** Similar to (**a–d**), but for ORBvl inputs. **j** Scale bars, 50 ms, 10 mV. **k** ORBvl inputs were weak in V1 inhibitory neurons across layers. **m–p** Similar to (**a–d**), but for LP inputs. **n** Scale bars, 50 ms, 10 mV. **o** LP inputs strongly activated the L1-INs in L1, VIP+ neurons from L2/3 to L6, and PV+ neurons in L5. **q** Summary of major connections of V2M and ACA inputs in V1 local circuits. V2M inputs, purple lines; ACA inputs, green lines. Line width represents the amplitude of synaptic input. Only the connections with an averaged EPSP amplitude of >3 mV were included. **r** Similar to (**q**), but for ORBvl (brown) and LP (blue) inputs. The number of neurons in each group is indicated by the numbers displayed in the figure. See Supplementary Data 3 for detailed input strengths. Source data are provided as a Source Data file.

in each recorded L6 Pyr. In each animal, we injected an AAV expressing ChR2-EYFP in the V2M (AAV-CaMKIIα-ChR2-EYFP) and also injected an AAV expressing ChrimsonR-tdTomato (AAV-hSyn-ChrimsonR-tdTomato) in the ACA (Fig. 3a). Expression of ChR2-EYFP in the V2M and ChrimsonR-tdTomato in the ACA resulted in the expected bright green and red axonal fluorescence in L6 of V1 (Fig. 3b–d).

Activation of V2M inputs (10 pulses @ 10 Hz, 488 nm) and ACA inputs (10 pulses @ 10 Hz, 647 nm) elicited robust postsynaptic currents (EPSCs) in L6 Pyrs (Fig. 3e–g). The majority of recorded L6 Pyrs (57%) received inputs from both V2M and ACA, with 40% receiving only V2M inputs and 3% not receiving inputs from either (Fig. 3h). These results show that CC top-down inputs from V2M and ACA converge on the same group of L6 Pyrs, indicating integrated processing of their top-down modulation signals in L6 of V1.

We next examined the innervation patterns of LP and ORBvl inputs in L5 Pyrs. For each animal, we injected an AAV expressing ChrimsonR-tdTomato in the LP (AAV-hSyn-ChrimsonR-tdTomato) and also injected an AAV expressing ChR2-EYFP in the ORBvl (AAV-CaMKIIα-ChR2-EYFP) (Fig. 4a). Expression of ChrimsonR-tdTomato in the LP and ChR2-EYFP in the ORBvl resulted in bright red and green axonal fluorescence in L5 of V1, with LP inputs primarily in upper L5 (red) and ORBvl inputs in lower L5 (green) (Fig. 4b–d). Activation of LP inputs (10 pulses @ 10 Hz, 647 nm) elicited robust postsynaptic currents (EPSCs) in about half of recorded L5 Pyrs (Pyr$_{\leftarrow LP}$ neurons); these neurons did not respond to activation of ORBvl inputs (10 pulses @ 10 Hz, 488 nm) (Fig. 4e–g). Intriguingly, when we activated ORBvl inputs, the remaining half of L5 Pyrs displayed robust EPSCs (Pyr$_{\leftarrow ORBvl}$ neurons) (Fig. 4e–h). These results indicate that TC and CC top-down inputs from LP and ORBvl employ parallel processing strategies by selectively activating distinct L5 Pyr populations in V1. Thus, our results reveal that examined CC and TC top-down inputs employ distinct processing strategies in their interactions, with integrated processing of V2M and ACA inputs in L6 versus parallel processing of LP and ORBvl inputs in L5.

## Pyr$_{\leftarrow LP}$ and Pyr$_{\leftarrow ORBvl}$ neurons are distinct L5 Pyr types distinguished by I$_h$

Given the notable specificity with which L5 Pyrs process LP and ORBvl inputs, we narrowed our focus to exploring the properties of these neurons: Pyr$_{\leftarrow LP}$ and Pyr$_{\leftarrow ORBvl}$ neurons. Up to this point of the study, we have identified Pyr$_{\leftarrow LP}$ and Pyr$_{\leftarrow ORBvl}$ neurons based on their input patterns, a relatively cumbersome process requiring virus injection, patch-clamp recording, and optogenetic activation. The intrinsic electrophysiological properties assessed using patch-clamp recording alone are used extensively for classifying different types of neurons, with, for example, hyperpolarization-activated inward currents (I$_h$) showing informatively varied distributions for distinct neuron types[39,50]. During our routine assessment of patch-clamp recordings by calculating access resistance, we introduced a 10-mV hyperpolarization to the recorded neurons. Pyr$_{\leftarrow ORBvl}$ neurons exhibited an I$_h$ in response to this 10-mV hyperpolarization, whereas Pyr$_{\leftarrow LP}$ neurons lacked this I$_h$—as indicated by an I$_h$ slope (see Method) close to 0 (I$_h$ slope of Pyr$_{\leftarrow LP}$, $-3.1 \pm 2.6$ pA/s, $n = 13$ neurons; I$_h$ slope of Pyr$_{\leftarrow ORBvl}$,

$82.5 \pm 8.6$ pA/s, $n = 17$ neurons; $P = 4 \times 10^{-6}$, two-sided Wilcoxon rank-sum test). Moreover, bath application of a known antagonist of hyperpolarization-activated cyclic nucleotide-gated channels (HCN channels) (ZD7288[51], 1 μM) completely eliminated the I$_h$ in Pyr$_{\leftarrow ORBvl}$ neurons, yet had no effect on I$_h$-lacking Pyr$_{\leftarrow LP}$ neurons in response to the 10-mV hyperpolarization. These findings indicate that the observed I$_h$ in Pyr$_{\leftarrow ORBvl}$ neurons is mediated by HCN channels (Fig. 5a, b).

Given the observed differences in the I$_h$ slope between Pyr$_{\leftarrow ORBvl}$ and Pyr$_{\leftarrow LP}$ neurons, we explored the use of the I$_h$ slope as a criterion to distinguish between these Pyr populations. Specifically, we revisited an aforementioned dataset for individual activation of LP and ORBvl inputs (from Fig. 1k–t) with a large set of L5 Pyrs responding to LP and ORBvl inputs (including 82 Pyr$_{\leftarrow LP}$ and 65 Pyr$_{\leftarrow ORBvl}$ neurons), recorded with an antagonist of voltage-gated sodium channels (TTX) and an antagonist of voltage-gated potassium channels (4-AP) (Fig. 5c). Control experiments showed TTX and 4-AP treatment had no effect on the I$_h$ slope (Supplementary Fig. 7). Notably, the I$_h$ slope of the Pyr$_{\leftarrow ORBvl}$ neurons was significantly steeper than that of Pyr$_{\leftarrow LP}$ neurons ($P = 7 \times 10^{-24}$, two-sided Wilcoxon rank-sum test; Fig. 5d), indicating that ORBvl and LP inputs selectively activate two types of L5 Pyrs that can be reliably characterized using I$_h$.

We subsequently used a support vector machine (SVM) approach[52] for distinguishing Pyr$_{\leftarrow LP}$ and Pyr$_{\leftarrow ORBvl}$ neurons in the aforementioned dataset. By employing the I$_h$-slope data from the 82 Pyr$_{\leftarrow LP}$ neurons and 65 Pyr$_{\leftarrow ORBvl}$ neurons, we trained a linear SVM classifier ("SVM Classifier(I$_h$)") to determine the decision boundary of the I$_h$ slope that optimally separates the two L5 Pyr types (Fig. 5e). The performance of the SVM Classifier (I$_h$) was evaluated using tenfold cross-validation, repeating the entire tenfold process 20 times with random divisions of the data. The detected average accuracy of 88.2% significantly surpassed the chance level of 50% ($P = 4 \times 10^{-28}$, two-sided one-sample $t$-test). The accuracy also exceeded the 95% confidence interval as obtained from classifiers trained on shuffled I$_h$-slope data (Fig. 5f). Next, we applied the SVM Classifier(I$_h$) trained with the complete dataset (82 Pyr$_{\leftarrow LP}$ neurons and 65 Pyr$_{\leftarrow ORBvl}$ neurons) to another dataset (the data from Fig. 4 for alternative activation of LP and ORBvl inputs in the same brain slice) (Fig. 5g). The SVM Classifier(I$_h$) informatively distinguished Pyr$_{\leftarrow LP}$ and Pyr$_{\leftarrow ORBvl}$ neurons, achieving 90% accuracy (Fig. 5h, i). Thus, the SVM Classifier(I$_h$) can effectively distinguish L5 Pyr$_{\leftarrow LP}$ and Pyr$_{\leftarrow ORBvl}$ neurons based on simple hyperpolarization data obtained through patch-clamp recordings.

## Pyr$_{\leftarrow LP}$ and Pyr$_{\leftarrow ORBvl}$ neurons employ distinct action potential modes and have unique gene-expression profiles

To investigate the functional divergence of Pyr$_{\leftarrow LP}$ and Pyr$_{\leftarrow ORBvl}$ neurons in the parallel processing of TC and CC top-down inputs, we explored the potential electrophysiological and genetic bases by conducting Patch-seq[53,54], which simultaneously yielded data for the transcriptomes and a variety of electrophysiological properties for the recorded neurons (Fig. 6a and Supplementary Figs. 8, 9). In total, we carried out patch-clamp recordings on 91 Pyrs in L5 of V1 from 15 mice.

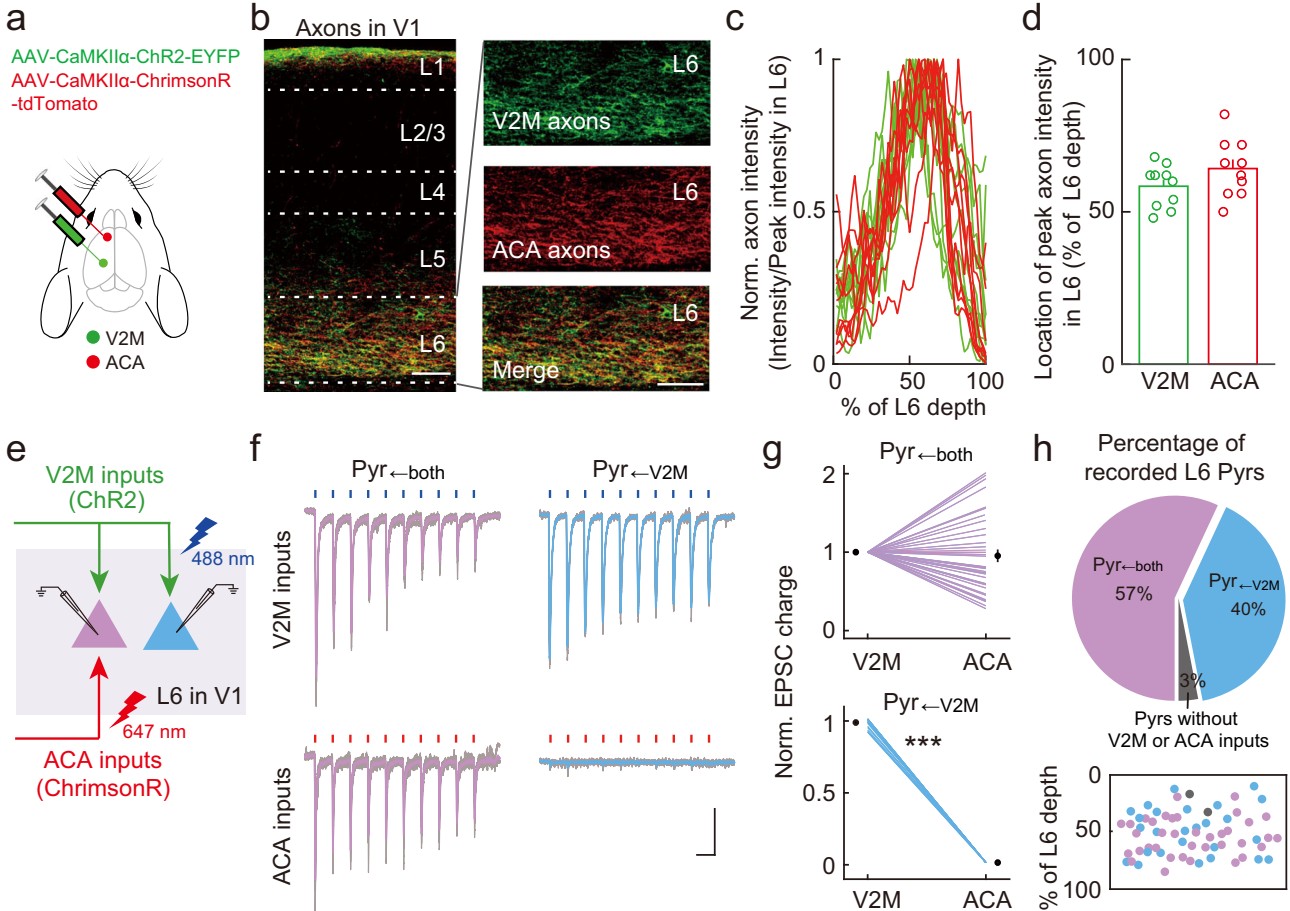

**Fig. 3 | V2M and ACA inputs converge on L6 Pyrs. a** Schematic of the viral strategy for independent optogenetic activation of V2M and ACA inputs. **b** Fluorescent images showing the distribution of V2M (green) and ACA axons (red) in V1, repeated three times with similar results. Scale bars, 100 μm. **c** Normalized axon intensity of V2M (green) and ACA (red) axons as a function of L6 depth. 0% and 100% represent the upper and lower boundaries of L6. $n = 10$ slices. **d** Summary of L6 depth for peak V2M (green) axon intensity and for peak ACA (red) axon intensity. $n = 10$ slices, $P = 0.12$, two-sided Wilcoxon sign-rank test. **e** Schematic of the slice experiment examining independent optogenetic activation of V2M and ACA inputs. **f** EPSCs evoked by independent activation of V2M and ACA inputs in L6 Pyrs. Blue dots, 488-nm light stimulation, 10 Hz, 1 ms, 3.5 mW. Red dots, 647-nm light stimulation, 10 Hz, 1 ms, 3.5 mW. Gray traces are individual trials, while purple and blue traces are averages of five trials. Scale bars, 100 ms, 50 pA. **g** Normalized EPSC charge evoked by activation of V2M and ACA inputs in Pyr$_{\leftarrow both}$ (top) and Pyr$_{\leftarrow V2M}$ neurons (bottom). The circles indicate the means. $n = 5$ mice, 7 slices. Pyr$_{\leftarrow both}$, $n = 37$ neurons, $P = 0.57$; Pyr$_{\leftarrow V2M}$, $n = 26$ neurons, $P = 3 \times 10^{-39}$. ***$P < 0.001$; two-sided paired $t$-test. **h** Percentage (top) and distribution (bottom) of L6 Pyrs activated by V2M and ACA inputs. Purple, L6 Pyrs receiving both inputs; Blue, L6 Pyrs receiving only V2M inputs; gray, L6 Pyrs without any response to V2M or ACA inputs. Data are presented as the mean ± SEM. Source data are provided as a Source Data file. Figure 3a adapted from Petrucco, L. (2020). Mouse head schema. Zenodo. https://doi.org/10.5281/zenodo.3925903 under a CC BY license: https://creativecommons.org/licenses/by/4.0/.

Among these neurons, 47 were identified as Pyr$_{\leftarrow LP}$ neurons, and 44 were identified as Pyr$_{\leftarrow ORBvl}$ neurons using the SVM Classifier(I$_h$).

Regarding their distinct action potential (AP) characteristics, Pyr$_{\leftarrow LP}$ neurons had significantly higher AP thresholds than Pyr$_{\leftarrow ORBvl}$ neurons (Supplementary Fig. 8). All recorded Pyr$_{\leftarrow LP}$ neurons exhibited regular spiking (RS) in response to a near-threshold depolarizing current pulse injected into the cell bodies (47 out of 47, 100%), whereas most of the Pyr$_{\leftarrow ORBvl}$ neurons displayed bursting spiking (BS, 29 out of 44, 66%, Fig. 6b, c). Additionally, Pyr$_{\leftarrow ORBvl}$ neurons had significantly larger depolarizing afterpotential (DAP) following a single AP as compared to Pyr$_{\leftarrow LP}$ neurons, in which DAP were largely absent (Fig. 6d). These findings indicate that Pyr$_{\leftarrow LP}$ and Pyr$_{\leftarrow ORBvl}$ neurons employ distinct AP modes, indicating that they could convey distinct signals in the parallel processing of LP and ORBvl inputs.

For the gene-expression profiles from the Patch-seq, 80 out of 91 L5 Pyrs passed the quality control (see Methods), including 41 Pyr$_{\leftarrow LP}$ neurons and 39 Pyr$_{\leftarrow ORBvl}$ neurons. The mean number of detectable genes per cell was 3247 ± 193 in Pyr$_{\leftarrow LP}$ neurons and 4013 ± 189 in Pyr$_{\leftarrow ORBvl}$ neurons (Supplementary Data 8–10). We initially assessed the

specificity of our sampling for the two neuron types based on the expression levels of a set of known markers[48,55]. As expected, we found that both Pyr$_{\leftarrow LP}$ and Pyr$_{\leftarrow ORBvl}$ neurons expressed pan-neuronal genes, such as *Snap25* (synaptosome-associated protein 25), and excitatory neuron genes, such as *Slc17a7* (vesicular glutamate transporter 1) (Supplementary Fig. 9). Moreover, and again supporting the specificity of sampling, both neuron types expressed a known L5 marker gene *Fezf2* (FEZ family zinc finger 2) (Supplementary Fig. 9).

There were 67 differentially expressed genes (DEGs) between Pyr$_{\leftarrow LP}$ neurons and Pyr$_{\leftarrow ORBvl}$ neurons: 18 with elevated expression in Pyr$_{\leftarrow LP}$ neurons and 49 with elevated expression in Pyr$_{\leftarrow ORBvl}$ neurons (Fig. 6e, Supplementary Fig. 9, and Supplementary Data 11). Recalling the known role of HCN proteins in generating I$_h$, it was unsurprising to note that expression of *Hcn1* was significantly higher in Pyr$_{\leftarrow ORBvl}$ neurons ($q$ value = 0.03, Log2FC = 3.8). The elevation of the potassium channels *Kcna6* (potassium voltage-gated channel subfamily A member 6, also known as Kv1.6) and *Kcnk2* (potassium two-pore domain channel subfamily K member 2) in Pyr$_{\leftarrow LP}$ neurons, along with the elevation of *Kcnt1* (potassium sodium-activated channel subfamily T

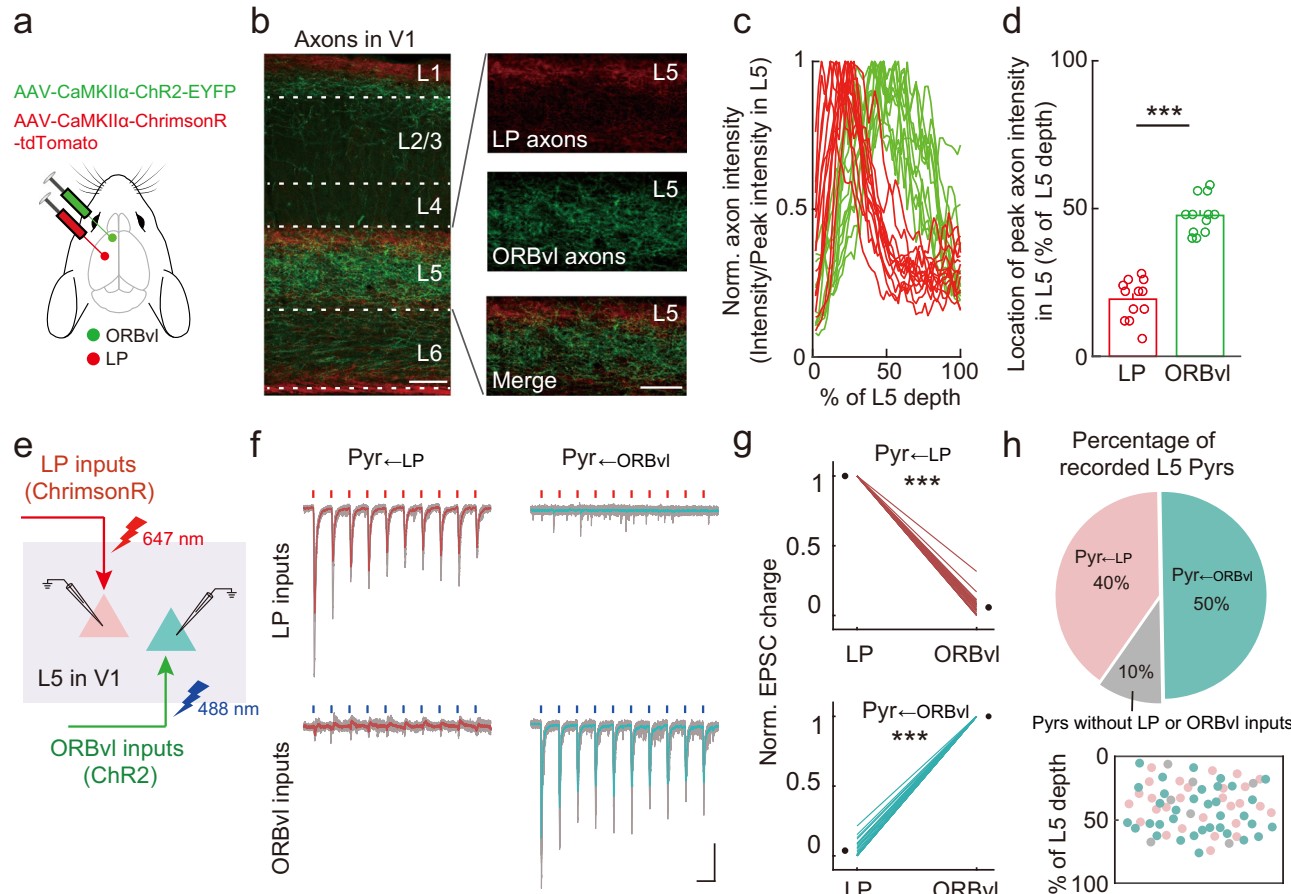

**Fig. 4 | LP and ORBvl inputs target distinct subpopulations of L5 Pyrs.**
**a** Schematic of the viral strategy for independent optogenetic activation of LP and ORBvl inputs. **b** Fluorescent images showing the distribution of LP (red) and ORBvl axons (green) in V1, repeated three times with similar results. Scale bars, 100 μm. **c** Normalized axon intensity of LP (red) and ORBvl (green) axons as a function of L5 depth. 0% and 100% represent the upper and lower boundaries of L5. $n = 12$ slices. **d** Summary of L5 depth for peak LP (red) axon intensity and for peak ORBvl (green) axon intensity. $n = 12$ slices, $P = 5 \times 10^{-4}$. ***$P < 0.001$; two-sided Wilcoxon sign-rank test. **e** Schematic of the slice experiment examining independent optogenetic activation of LP and ORBvl inputs. **f** EPSCs evoked by independent activation of LP and ORBvl inputs in L5 Pyrs. Red dots, 647-nm light stimulation, 10 Hz, 1 ms, 3.5 mW. Blue dots, 488-nm light stimulation, 10 Hz, 1 ms, 3.5 mW. Gray traces are

individual trials, while red and green traces are averages of 5 trials. Scale bars, 100 ms, 100 pA. **g** Normalized EPSC charge evoked by activation of LP and ORBvl inputs in Pyr$_{\leftarrow LP}$ (top) and Pyr$_{\leftarrow ORBvl}$ neurons (bottom). The circles indicate the means. $n = 12$ mice, 14 slices. Pyr$_{\leftarrow LP}$, $n = 28$ neurons, $P = 2 \times 10^{-4}$; Pyr$_{\leftarrow ORBvl}$, $n = 35$ neurons, $P = 9 \times 10^{-33}$. ***$P < 0.001$, two-sided paired $t$-test. **h** Percentage (top) and distribution (bottom) of L5 Pyrs activated by LP and ORBvl inputs (Pyr$_{\leftarrow LP}$ and Pyr$_{\leftarrow ORBvl}$ neurons). Red, Pyr$_{\leftarrow LP}$ neurons; green, Pyr$_{\leftarrow ORBvl}$ neurons; gray, L5 Pyrs without any response to LP or ORBvl inputs. Data were presented as the mean ± SEM. Source data are provided as a Source Data file. Figure 4a adapted from Petrucco, L. (2020). Mouse head schema. Zenodo. https://doi.org/10.5281/zenodo.3925903 under a CC BY license: https://creativecommons.org/licenses/by/4.0/.

member 1, also known as K$_{Na}$1.1) in Pyr$_{\leftarrow ORBvl}$ neurons, provides a basis for understanding the observed differences in electrophysiological properties. This is given the known roles of KCNA6, KCNK2, and KCNT1 proteins in repolarizing the membrane potential after an AP and in setting the resting membrane potential[56–59]. Moreover, the elevation of calcium channel *Cacna1h* (calcium voltage-gated channel subunit alpha1 H, which encodes Cav3.2 channels) in Pyr$_{\leftarrow ORBvl}$ neurons, compared to Pyr$_{\leftarrow LP}$ neurons, may contribute to generate burst spiking, given the known role of Cav3.2 channels in producing low-threshold calcium spikes[60]. Thus, our Patch-seq analysis characterized the distinctive AP modes used by these two L5 Pyr types and identified multiple DEGs with known functions relevant to their functional divergence.

### Pyr$_{\leftarrow LP}$ and Pyr$_{\leftarrow ORBvl}$ neurons, respectively mediate intracortical and subcortical output channels

L5 of V1 contains two major Pyr types: PT and IT neurons, mediating distinct output channels[4,37]. IT neurons preferentially targeting cortical areas and PT neurons targeting subcortical areas[39]. Both IT and PT neurons target the striatum, but IT neurons target the striatum in both

hemispheres, whereas PT neurons only target the ipsilateral striatum[40]. Our results indicate that the LP and ORBvl inputs undergo parallel processing through two distinct Pyr types in L5 of V1. Seeking a deeper understanding of information processing as carried out by these two Pyr types, we next investigated the long-range output targets of Pyr$_{\leftarrow LP}$ and Pyr$_{\leftarrow ORBvl}$ neurons. We used retrograde AAV to label V1 long-range projecting neurons targeting cortical and subcortical areas, and then conducted patch-clamp recordings on the labeled L5 Pyrs (Fig. 7a). These neurons were next classified using the aforementioned SVM Classifier(I$_h$).

Retrograde AAV expressing Cre (Retro-AAV-hSyn-Cre) was injected into nine distinct cortical and subcortical downstream targets of V1[4,16,18] in loxP-flanked tdTomato reporter mice (Ai14 mice). The downstream cortical areas included the V2M, RSP, ACA, and ORBvl, while the subcortical areas included the ipsilateral and contralateral dorsal medial striatum (ipsi-DMS and contra-DMS), Pons, Superior Colliculus (SC), and LP. We observed distinct laminar distribution specificity among retrograde-labeled V1 neurons from various downstream targets. Neurons retrograde-labeled from cortical areas and the striatum were often distributed across both superficial (L2/3) and deep

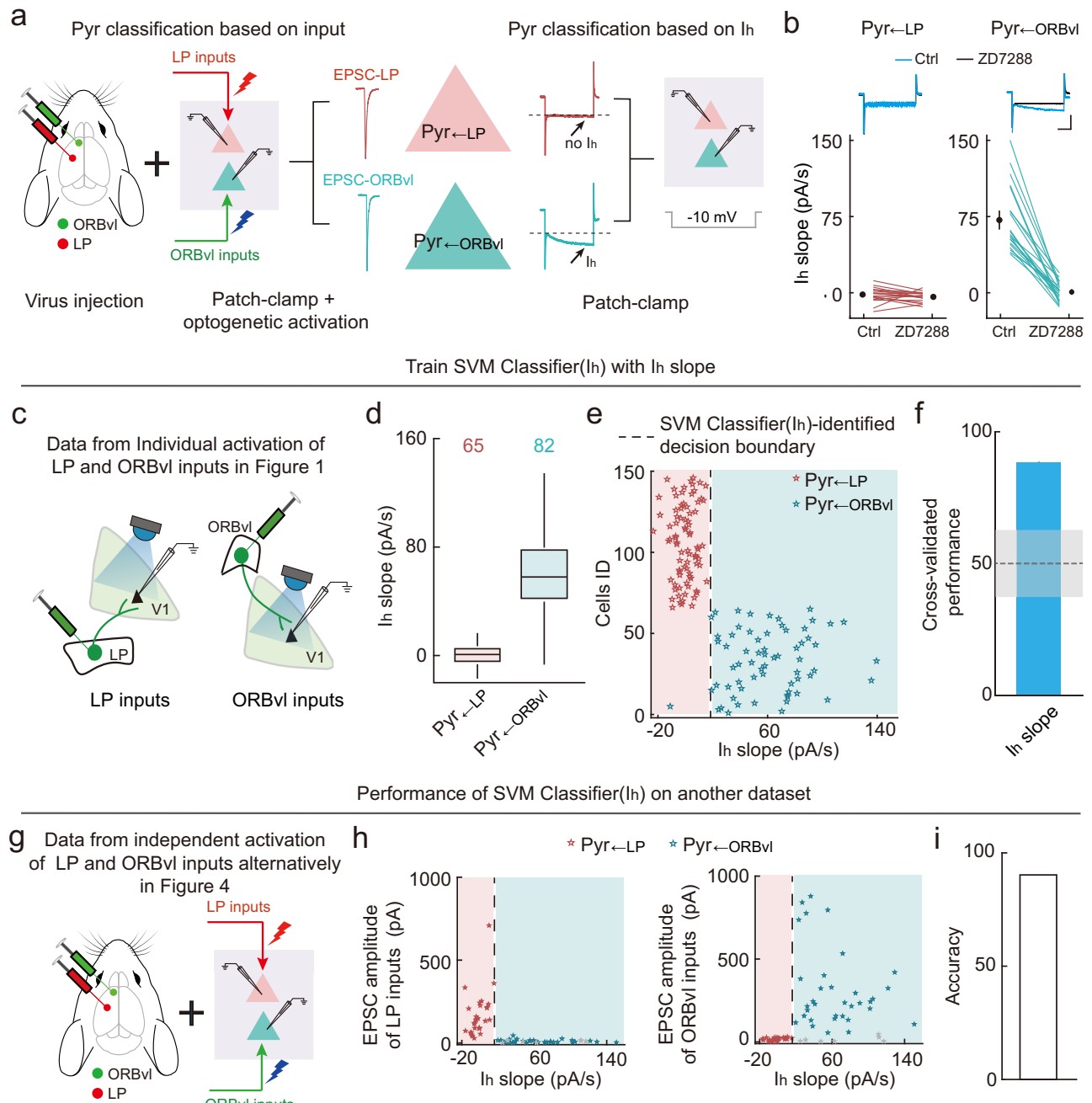

**Fig. 5 | An SVM classifier trained with $I_h$ slope can reliably distinguish Pyr←LP and Pyr←ORBvl neurons. a** Schematic of the input-pattern-based and $I_h$-based strategies for classifying Pyr←LP and Pyr←ORBvl neurons. **b** Effects of HCN1-channel antagonist (ZD7288) on $I_h$ slope of Pyr←LP and Pyr←ORBvl neurons. Left, ZD7288 had no effect in $I_h$-lacking Pyr←LP neurons ($n$ = 17 neurons, $P$ = 0.1, two-sided Wilcoxon sign-rank test). Inset, current change elicited by a 10-mV hyperpolarization in an example Pyr←LP neuron with (black) or without (blue) ZD7288. Scale bars, 100 ms, 100 pA. Right, similar to left, but for Pyr←ORBvl neurons. ZD7288 significantly decreased the $I_h$ slope of Pyr←ORBvl neurons ($n$ = 21 neurons, $P$ = 6 × 10⁻⁵). Data were presented as the mean ± SEM. **c** Schematic of the dataset used to train "SVM Classifier($I_h$)", using the individual activation of LP and ORBvl inputs data from Fig. 1 (82 Pyr←LP neurons and 65 Pyr←ORBvl neurons). **d** Box plot showing $I_h$ slope of Pyr←LP and Pyr←ORBvl neurons from Fig. 1. Edges, 25th and 75th percentiles; central line, median; whiskers, 1.5× the interquartile range of the edges. **e** Decision boundary (dashed line) for classifying Pyr←LP and Pyr←ORBvl neurons derived from SVM

Classifier($I_h$). Red and green stars, input-pattern-classified Pyr←LP and Pyr←ORBvl neurons, respectively. **f** Cross-validated classifier performance for leave-one-out data (tenfold cross-validation, $n$ = 20 times). Dashed line, chance level as 50%. The gray area indicates the 95% confidence interval of classifier performances trained on shuffled data. **g** Schematic of another dataset used to test the performance of the SVM Classifier($I_h$): this is the data from Fig. 4 regarding independent activation of LP and ORBvl inputs alternatively in the same brain slice (28 Pyr←LP neurons and 35 Pyr←ORBvl neurons). **h** Performance of the SVM Classifier($I_h$) in distinguishing Pyr←LP from Pyr←ORBvl neurons in the dataset from Fig. 4. Dashed line, decision boundary for classifying Pyr←LP and Pyr←ORBvl neurons. Red and green stars, input-pattern-classified Pyr←LP and Pyr←ORBvl neurons, respectively. Gray stars, L5 Pyrs without any response to LP or ORBvl inputs. **i** Prediction accuracy (see Methods). Source data are provided as a Source Data file. Figure 5, Panels (**a**,**g**) adapted from Petrucco, L. (2020). Mouse head schema. Zenodo. https://doi.org/10.5281/zenodo.3925903 under a CC BY license: https://creativecommons.org/licenses/by/4.0/.

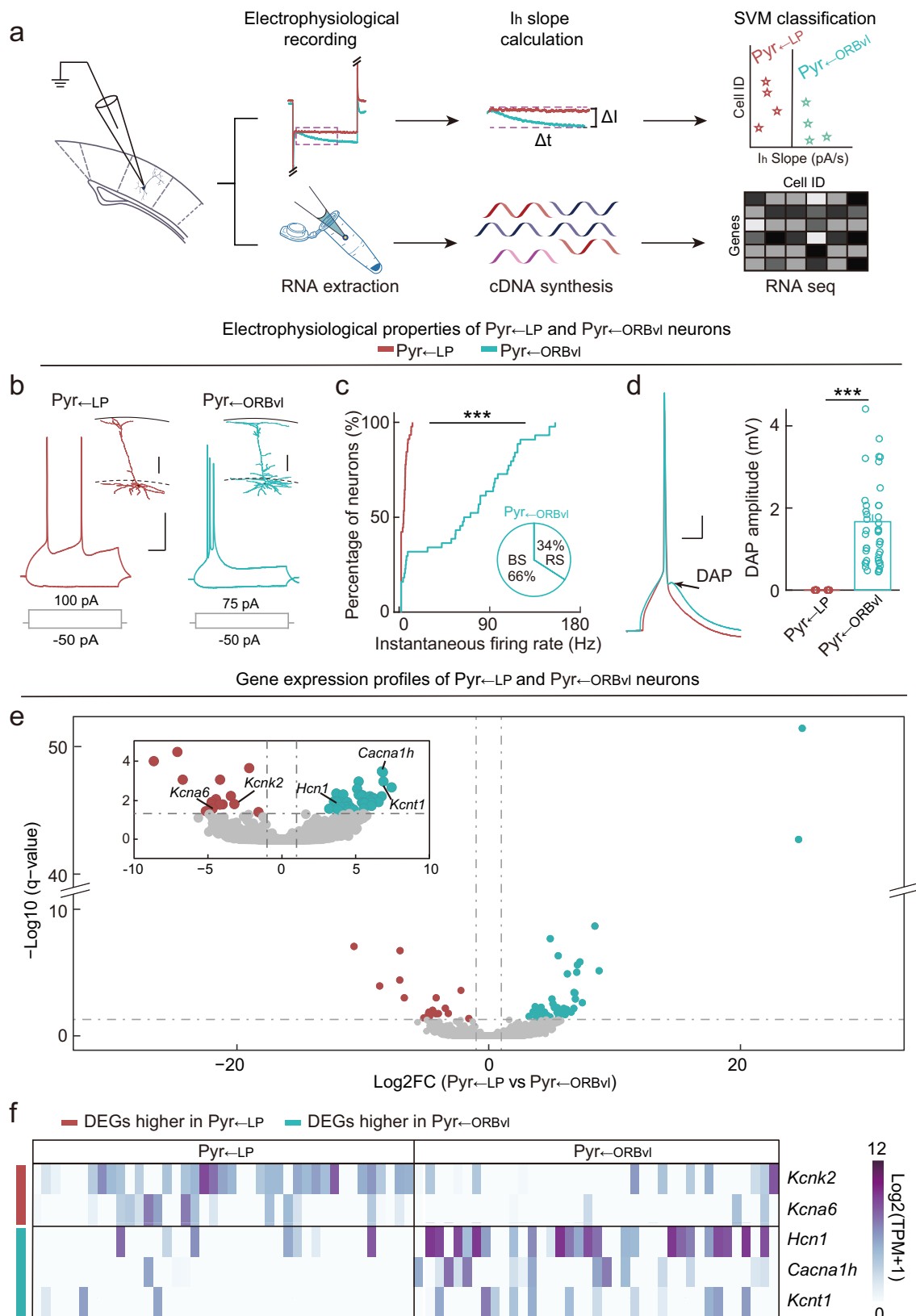

layers (L5 and L6), whereas neurons labeled from other subcortical areas (Pons, SC, and LP) exhibited a dense distribution specifically in L5 (Supplementary Fig. 10). Generally, retrograde-labeled neurons were observed in L5 (Fig. 7b), with the exception of ACA-projecting neurons, which were exclusively distributed in L2/3 and L6 (Supplementary Fig. 10).

Using the SVM Classifier(Ih), we determined that more than 78% of labeled L5 Pyrs projecting to cortical areas (V2M, RSP, and ORBvl) and contra-DMS were classified as Pyr$_{\leftarrow LP}$ neurons. In contrast, more than 82% of labeled L5 Pyrs projecting to other subcortical areas (Pons, SC, and LP) were classified as Pyr$_{\leftarrow ORBvl}$ neurons (Fig. 7c and Supplementary Fig. 11). Additionally, labeled

**Fig. 6 | Pyr$_{\leftarrow LP}$ and Pyr$_{\leftarrow ORBvl}$ neurons exhibit distinct electrophysiological properties and gene-expression profiles. a** Schematic of the Patch-seq experiment. **b** Examples of the firing pattern and morphology of Pyr$_{\leftarrow LP}$ (red) and Pyr$_{\leftarrow ORBvl}$ (green) neurons. For firing patterns, gray lines represent the current steps used to elicit the intended firing pattern; scale bars, 50 ms, 20 mV. For morphological reconstruction, the solid black line represents the pia mater; the dashed black line is the upper border of L5; the colored outline represents the somatodendritic region; scale bar, 100 μm. **c** Cumulative distributions of instantaneous firing rates of the first two action potentials (APs) elicited by positive current injection into Pyr$_{\leftarrow LP}$ (red) and Pyr$_{\leftarrow ORBvl}$ (green) neurons. All recorded Pyr$_{\leftarrow LP}$ neurons exhibited regular spiking (RS), whereas 34% of Pyr$_{\leftarrow ORBvl}$ showed RS and 66% exhibited burst spiking (BS) ($n$ = 20 mice, 20 slices; Pyr$_{\leftarrow LP}$, $n$ = 47 neurons; Pyr$_{\leftarrow ORBvl}$, $n$ = 44 neurons; P = 6 × 10$^{-10}$, two-sided Kolmogorov–Smirnov test). Inset, the percentage of Pyr$_{\leftarrow ORBvl}$ neurons exhibiting RS and BS. ***$P$ < 0.001; two-sided Kolmogorov–Smirnov test. **d** Properties of Pyr$_{\leftarrow LP}$ and Pyr$_{\leftarrow ORBvl}$ neurons after a single AP. Left, for example, AP traces of Pyr$_{\leftarrow LP}$ (red) and Pyr$_{\leftarrow ORBvl}$ (green) neurons. Single AP was induced by a brief depolarizing current step (2 pA, 25 ms). Black

arrow, DAP of Pyr$_{\leftarrow ORBvl}$ neurons. Scale bars, 20 ms, 20 mV. Right, DAP amplitude of Pyr$_{\leftarrow LP}$ and Pyr$_{\leftarrow ORBvl}$ neurons. P = 1 × 10$^{-18}$, two-sided Wilcoxon rank-sum test. ***$P$ < 0.001. **e** Volcano plot showing the differentially expressed genes (DEGs) between Pyr$_{\leftarrow LP}$ and Pyr$_{\leftarrow ORBvl}$ neurons. Red dots, DEGs with higher expression levels in Pyr$_{\leftarrow LP}$ neurons ($q$ < 0.05 and log2FC (Pyr$_{\leftarrow LP}$ vs Pyr$_{\leftarrow ORBvl}$) <−1). Green dots, DEGs with higher expression levels in Pyr$_{\leftarrow ORBvl}$ neurons ($q$ < 0.05 and log2FC (Pyr$_{\leftarrow LP}$ vs Pyr$_{\leftarrow ORBvl}$) >1). Inset, enlarged view of the genes with -log10 ($q$ value) <5 and log2FC (Pyr$_{\leftarrow LP}$ vs Pyr$_{\leftarrow ORBvl}$) ranging between −10 to 10. Pyr$_{\leftarrow LP}$, $n$ = 41 neurons; Pyr$_{\leftarrow ORBvl}$, $n$ = 39 neurons. **f** Heat map for expression of selected DEGs in all examined Pyr$_{\leftarrow LP}$ and Pyr$_{\leftarrow ORBvl}$ neurons. Source data are provided as a Source Data file. Figure 6a adapted from Claudi, F. (2020). pyramidal neuron. Zenodo. https://doi.org/10.5281/zenodo.3925905 under a CC BY license: https://creativecommons.org/licenses/by/4.0/. Figure 6a adapted from Losch De Oliveira, D. (2020). Eppendorf tube. Zenodo. https://doi.org/10.5281/zenodo.3925953 under a CC BY license: https://creativecommons.org/licenses/by/4.0/. Figure 6a adapted from Bauer Negrini, G. (2020). DNA strand. Zenodo. https://doi.org/10.5281/zenodo.3926245 under a CC BY license: https://creativecommons.org/licenses/by/4.0/.

L5 Pyrs from ipsi-DMS contained 27% Pyr$_{\leftarrow ORBvl}$ neurons and 73% Pyr$_{\leftarrow LP}$ neurons. Notably, we further conducted independent optogenetic activation experiments on L5 Pyrs projecting to a subset of these downstream targets, including the RSP, ipsi-DMS, and SC. These experiments confirmed that the recorded L5 projecting Pyrs exclusively receive either LP or ORBvl inputs, a selectivity pattern accurately predicted by the SVM Classifier(I$_h$) (Supplementary Fig. 12). Specifically, we observed that 100% of L5 RSP-projecting Pyrs receive LP inputs, 100% of SC-projecting Pyrs receive ORBvl inputs, and among ipsi-DMS-projecting Pyrs 47% receive LP inputs while 53% receive ORBvl inputs. The SVM Classifier(I$_h$) effectively distinguished Pyr$_{\leftarrow LP}$ and Pyr$_{\leftarrow ORBvl}$ neurons across these groups with 100% accuracy.

These results indicate that Pyr$_{\leftarrow LP}$ neurons preferentially innervate cortical areas and the striatum, whereas Pyr$_{\leftarrow ORBvl}$ neurons innervate subcortical areas, findings consistent with the innervation preferences of IT and PT neurons. Recent studies have shown that IT and PT neurons form parallel subnetworks for cortical processing, characterized by distinct spatiotemporal activity patterns based on behavioral state, as a basic feature of cortical functional architecture across dorsal cortical areas[61,62]. Together, our results indicate that LP and ORBvl inputs undergo parallel processing within the IT and PT subnetworks, which separately mediate intracortical and subcortical output channels.

## LP and ORBvl inputs drive mutual inhibition between two Pyr-type-specific subnetworks in L5 of V1

Recent studies have suggested that the functional integration of IT and PT subnetworks across the dorsal cortex might be dynamically gated by inhibitory and modulatory mechanisms based on brain states and behavioral demands[61,62]. To explore the interaction between distinct Pyr-type-specific subnetworks in L5 of V1 while processing LP and ORBvl inputs, we optogenetically activated each of these inputs and measured the evoked excitatory and inhibitory postsynaptic currents (EPSCs and IPSCs) in Pyr$_{\leftarrow LP}$ and Pyr$_{\leftarrow ORBvl}$ neurons.

To activate LP inputs in V1, we injected an AAV expressing ChR2 in excitatory neurons (AAV-CaMKIIα-ChR2-EYFP) into the LP (Fig. 8a). Activation of LP inputs induced both EPSCs and IPSCs in Pyr$_{\leftarrow LP}$ neurons (Fig. 8b, left). EPSCs had short onset latencies (4.5 ± 0.3 ms, mean ± SEM, $n$ = 20 neurons), suggesting monosynaptic excitatory inputs. Notably, the latencies of IPSCs were significantly longer than for EPSCs (10.4 ± 1.1 ms, $P$ = 6 × 10$^{-6}$, two-sided Wilcoxon sign-rank test), suggesting multisynaptic inhibition. We observed a net inhibition in Pyr$_{\leftarrow ORBvl}$ neurons induced by LP inputs (IPSC, latency 10.9 ± 0.6 ms, $n$ = 26 neurons, Fig. 8b, right). LP inputs drive significantly stronger inhibition relative to excitation in Pyr$_{\leftarrow ORBvl}$ neurons than in Pyr$_{\leftarrow LP}$ neurons, as shown by the I/(I + E) ratio (Fig. 8c and Supplementary

Fig. 13). This suggests that LP inputs drive simultaneous activation of Pyr$_{\leftarrow LP}$ neurons and inhibition of Pyr$_{\leftarrow ORBvl}$ neurons.

We also investigated the involvement of local GABAergic interneurons in LP-input-induced inhibition. Initially, we compared LP-input strengths on L5 Pyr$_{\leftarrow LP}$ neurons with those on major types of inhibitory neurons (PV+, SST+, and VIP+ neurons) across different layers. LP inputs strongly activated VIP+ neurons in L2-L6 and PV+ neurons in L5, doing so with comparable input strengths to Pyr$_{\leftarrow LP}$ neurons (Fig. 8d). Subsequently, we examined how each inhibitory neuron type inhibited Pyr$_{\leftarrow LP}$ and Pyr$_{\leftarrow ORBvl}$ neurons. Optogenetic activation of PV+ and SST+ neurons induced large IPSCs in both Pyr$_{\leftarrow LP}$ and Pyr$_{\leftarrow ORBvl}$ neurons, whereas activation of VIP+ neurons evoked only very weak IPSCs in both types, each constituting less than 8% of the IPSCs induced by PV+ neuron in Pyr$_{\leftarrow ORBvl}$ neurons (Supplementary Fig. 14). These results indicate that LP inputs can directly engage L5 PV + neurons to inhibit Pyrs.

We next measured responses in Pyr$_{\leftarrow LP}$ and Pyr$_{\leftarrow ORBvl}$ neurons induced by ORBvl inputs (Fig. 8e and Supplementary Fig. 13). Activation of ORBvl inputs induced monosynaptic EPSCs and multisynaptic IPSCs in Pyr$_{\leftarrow ORBvl}$ neurons (EPSC latency, 3.7 ± 0.3 ms; IPSC latency, 6.6 ± 0.4 ms, mean ± SEM; $n$ = 20 neurons, Fig. 8f). In Pyr$_{\leftarrow LP}$ neurons, only multisynaptic IPSCs were observed (IPSC latency, 7.2 ± 0.3 ms, mean ± SEM, $n$ = 19 neurons). In contrast to LP inputs, ORBvl inputs drive significantly stronger inhibition relative to excitation in Pyr$_{\leftarrow LP}$ neurons than in Pyr$_{\leftarrow ORBvl}$ neurons (Fig. 8g), indicating that ORBvl inputs may simultaneously drive activation of Pyr$_{\leftarrow ORBvl}$ neurons and inhibition of Pyr$_{\leftarrow LP}$ neurons. ORBvl inputs were relatively weak in local inhibitory neurons, with L5 inhibitory neurons receiving the strongest ORBvl inputs (-20% of the input strength in Pyr$_{\leftarrow ORBvl}$ neurons, Fig. 8h). Note that our optogenetic activation profiling of ORBvl inputs revealed activation in 33% of L5 PV+ neurons and 25% of L5 SST+ neurons (Supplementary Fig. 4); thus, the inhibition induced by ORBvl inputs may be mediated by these inhibitory neurons. Together, these results demonstrate that LP and ORBvl inputs drive mutual inhibition between the two Pyr-type-specific subnetworks in L5 of V1 (likely involving PV+ and SST+ neurons), thus enabling dynamic gating of subnetwork functions.

## Discussion

In this study, we characterized the layer- and cell-type-specific organization of multiple CC and TC top-down inputs in V1, as well as their interactions in deep layers. Each top-down input to V1 engaged both excitatory and inhibitory neurons: CC and TC top-down inputs partially overlapped in superficial layers (L1 and L2/3), bypassed the medial layer (L4), and clearly segregated in deep layers (L5 and L6). Our datasets provide a valuable resource for understanding the neuronal mechanisms of top-down modulation of visual information

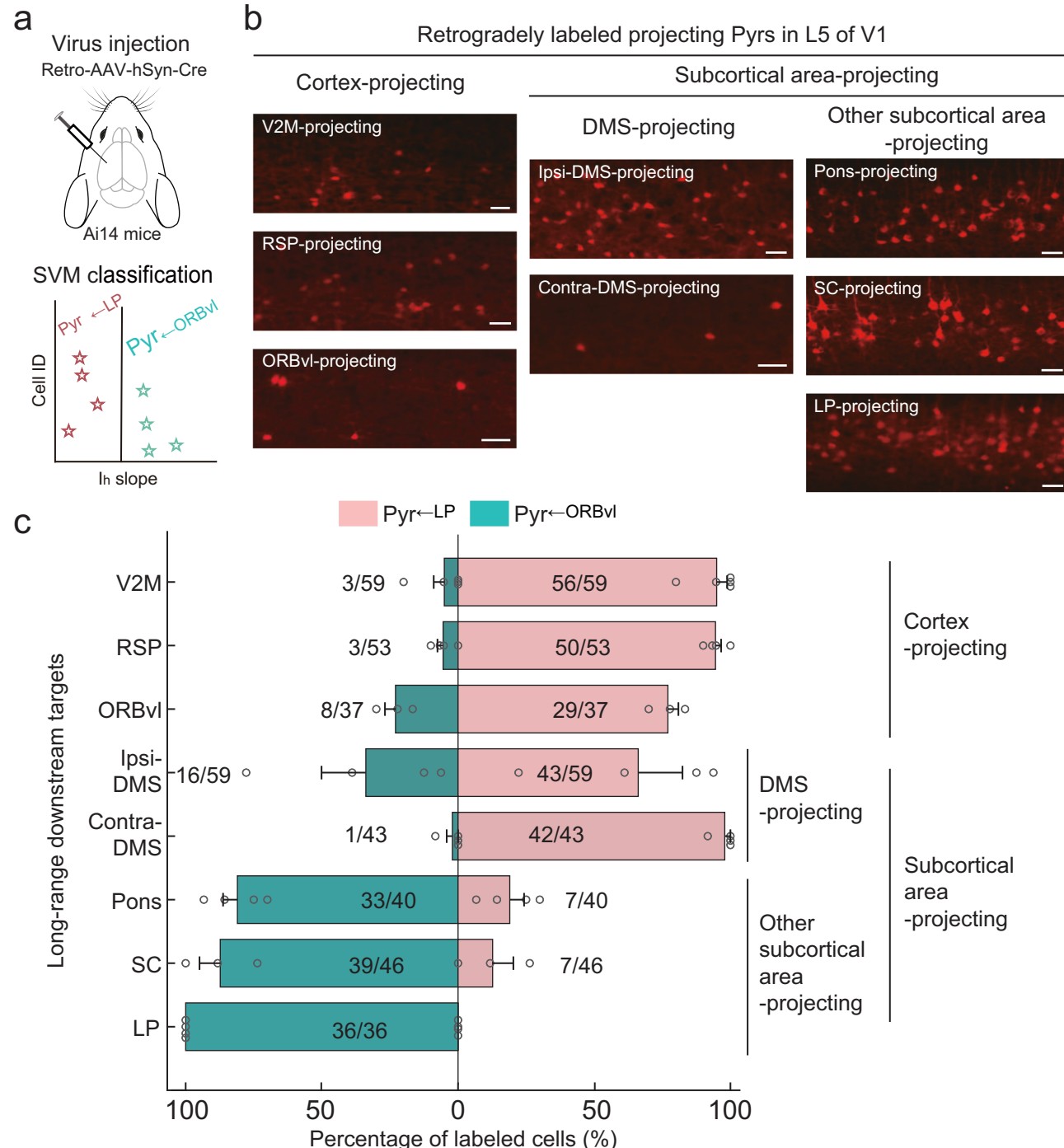

**Fig. 7 | Pyr$_{\leftarrow LP}$ and Pyr$_{\leftarrow ORBvl}$ neurons exhibit distinct long-range output patterns. a** Schematic for the Retro-AAV-mediated long-range output tracing experiment. **b** Fluorescence images showing the distribution of projecting Pyrs (red) in L5 of V1, retrogradely labeled by Retro-AAV injected in eight cortical and subcortical areas in loxP-flanked tdTomato reporter mice (Ai14 mice). Cortical areas: V2M, RSP, ORBvl. Subcortical areas: Ipsi- and contra-DMS, Pons, SC, and LP. Scale bars, 50 μm. **c** Proportion of Pyr$_{\leftarrow LP}$ and Pyr$_{\leftarrow ORBvl}$ neurons in labeled projecting Pyrs. The number of neurons in each group is indicated by the numbers displayed in the

figure. V2M, $n = 5$ mice, five slices; RSP, $n = 4$ mice, five slices; ORBvl, $n = 3$ mice, three slices; Ipsi-DMS, $n = 4$ mice, fivde slices; Contra-DMS, $n = 4$ mice, five slices; Pons, $n = 4$ mice, six slices; SC, $n = 3$ mice, three slices; LP, $n = 4$ mice, four slices. Data were presented as the mean ± SEM. Source data are provided as a Source Data file. Figure 7a adapted from Petrucco, L. (2020). Mouse head schema. Zenodo. https://doi.org/10.5281/zenodo.3925903 under a CC BY license: https://creativecommons.org/licenses/by/4.0/.

processing. Further, we found that CC top-down inputs from the V2M and ACA undergo integrated processing in L6 Pyrs, whereas TC top-down inputs from the ORBvl and LP undergo parallel processing within two L5 Pyr-type-specific subnetworks, separately mediating subcortical and intracortical V1 output channels. Notably, ORBvl and LP inputs drive mutual inhibition between these two subnetworks in L5 of

V1, potentially enabling dynamic gating of subnetwork functions in response to specific top-down modulation signals.

The visual cortex integrates both bottom-up and top-down inputs, contributing to form a unified visual perception[5,8,12,13]. Decades of studies have demonstrated the parallel processing of bottom-up inputs, starting at the retina and reaching V1 via retino-geniculo-

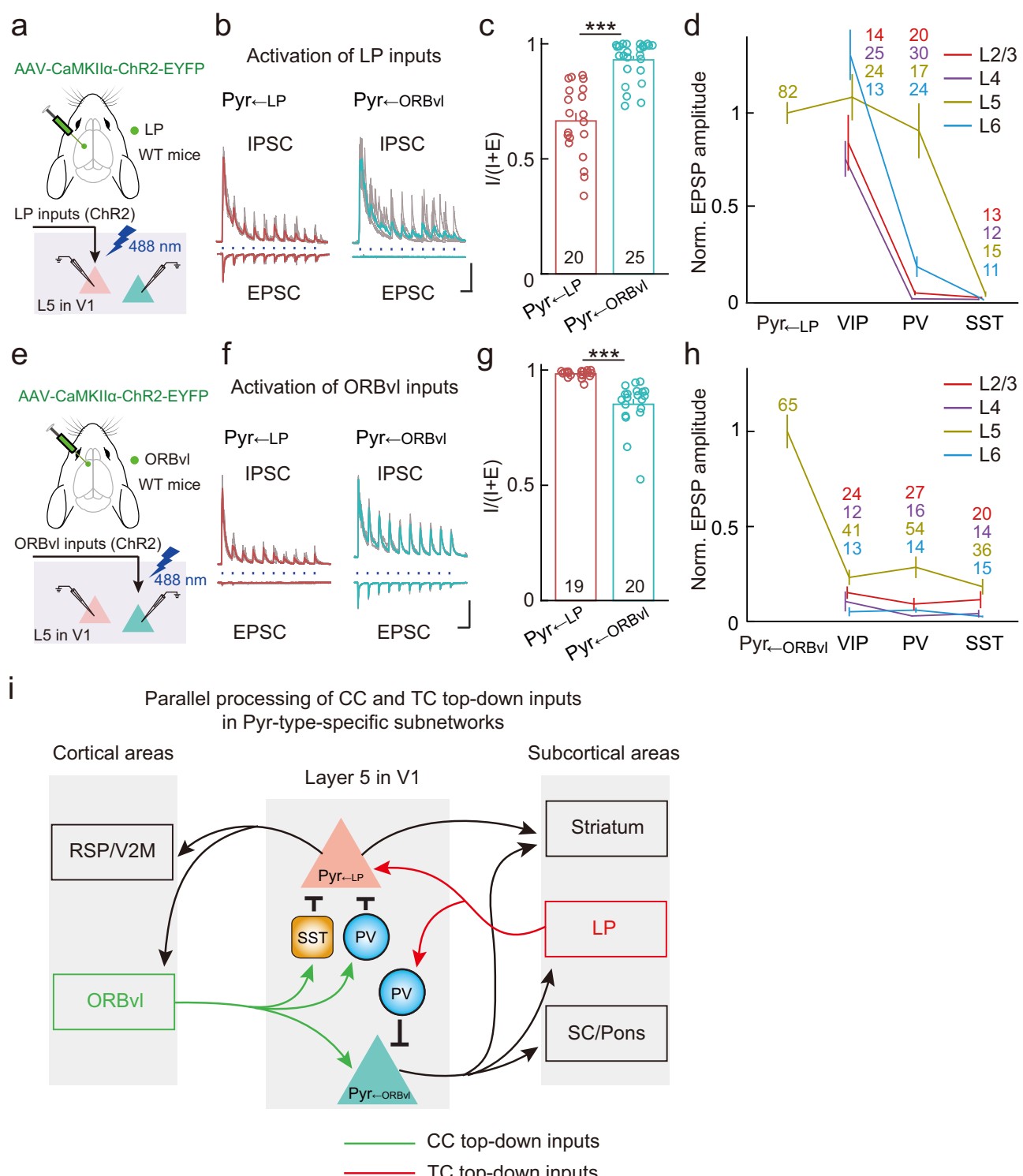

**Fig. 8 | LP and ORBvl inputs drive mutual inhibition between Pyr←LP and Pyr←ORBvl neurons in L5 of V1. a** Schematic of the slice experiment, illustrating the whole-cell recording conducted on individual Pyr←LP (red) or Pyr←ORBvl (green) neurons (one at a time), coupled with optogenetic activation of LP inputs. **b** EPSCs and IPSCs evoked by activating LP inputs in two example Pyrs. Gray traces are individual trials, while red and green traces are averages of 5 trials. Blue dots, 488-nm light stimulation, 10 Hz, 5 ms, 3.5 mW. Scale bars, 100 ms, 500 pA. Left, LP-input-evoked EPSCs (bottom) and IPSCs (top) in an example Pyr←LP neuron. Right, similar to left, but for a Pyr←ORBvl neuron. **c** Contribution of inhibitory inputs to total synaptic inputs, measured as $IPSC_{charge}/(IPSC_{charge} + EPSC_{charge})$. LP inputs drive significantly stronger inhibition relative to excitation in Pyr←ORBvl neurons than in Pyr←LP neurons. $P = 5 \times 10^{-7}$, two-sided Wilcoxon rank-sum test. $n = 4$ mice, five slices. **d** Normalized

EPSP amplitudes in different cell types evoked by LP inputs (normalized to averaged EPSP amplitude of Pyr←LP neurons in L5). The number of neurons in each group is indicated by the numbers displayed in the figure. **e–h** Similar to (**a–d**), but for ORBvl inputs. **g** ORBvl inputs drive significantly stronger inhibition relative to excitation in Pyr←LP neurons than in Pyr←ORBvl neurons. $P = 1 \times 10^{-7}$. $n = 6$ mice, seven slices. **h** Normalized to Pyr←ORBvl neurons. **i** Summary of the parallel processing of ORBvl and LP inputs within two L5 Pyr-type-specific subnetworks in V1. The number of neurons in each group is indicated by the numbers displayed in the figure. ***$P < 0.001$, two-sided Wilcoxon rank-sum test. Data were presented as the mean ± SEM. Source data are provided as a Source Data file. Figure 8a, e adapted from Petrucco, L. (2020). Mouse head schema. Zenodo. https://doi.org/10.5281/zenodo.3925903 under a CC BY license: https://creativecommons.org/licenses/by/4.0/.

cortical pathways, then advancing to higher cortical and thalamic areas[5–10]. Top-down inputs from higher-order cortical and thalamic areas to V1 are known to convey various dimensions of complex behavior-related signals to modulate visual processing[19–27,29,30], yet the neuronal mechanisms are not yet well understood. Our systematic characterization of the layer- and cell-type-specific organization of CC (V2M, ACA, and ORBvl) and TC (LP) top-down inputs in V1 offers insights into sophisticated top-down modulation mediated by distinct excitatory and inhibitory neurons in V1 local circuits. This understanding should substantially facilitate investigations into how various top-down inputs integrate with bottom-up inputs to generate task-related top-down modulations of visual processing in V1.

CC and TC top-down inputs are known to target both superficial and deep layers in V1[1,3,4]. Previous studies have demonstrated that top-down inputs preferentially activate L1-INs and L2/3 VIP+ neurons, which inhibit PV+ and SST+ neurons to disinhibit Pyrs[17,21,34,36,63–65]. Our findings extend this understanding by showing varied cell-type preferences across layers: VIP+ neurons are strongly activated in both superficial and deep layers, whereas PV+ and SST+ neurons are selectively activated in deep layers. VIP+ neurons disinhibit Pyrs, while PV+ and SST+ neurons inhibit Pyrs[32,44,66–68]. The higher density of VIP+ neurons in superficial layers and PV+ and SST+ neurons in deep layers[32] suggests a top-down modulation pattern wherein disinhibition dominates in superficial layers, gradually transitioning to increased inhibition in deep layers.

Cortical Pyrs have been divided by axonal projections into three main types: IT neurons preferentially innervating the cortex and striatum; PT neurons innervating subcortical areas; and corticothalamic neurons, which only innervate the thalamus[39]. These types overlap in deep layers, with IT and PT in L5, and IT and corticothalamic in L6. Our study has revealed the specificity of various CC and TC top-down inputs engaging distinct Pyr types in deep layers of V1. V2M and ACA inputs converge on L6 Pyrs, suggesting integrated processing of these inputs that likely involves both IT and corticothalamic neurons. And we found that ORBvl and LP inputs selectively activate PT (Pyr$_{\leftarrow ORBvl}$) and IT (Pyr$_{\leftarrow LP}$) neurons, each with distinct AP modes (BS spiking vs. RS spiking) and gene-expression profiles, indicating parallel processing of the information they convey in the PT and IT subnetworks.

The dendrites of Pyrs typically extend, and branch across multiple layers[39,44,69,70], and the spatial distribution of specific inputs within dendritic arborization serve distinct functions: spatially clustered co-active synapses are more efficacious in driving postsynaptic neurons than spatially distributed synapses[71,72]. Previous studies using subcellular ChR2-assisted circuit mapping (sCRACM) revealed the subcellular organization of local and long-range excitatory inputs to Pyrs in sensorimotor cortices, findings helpful for predicting the interactions among these inputs within the dendrites of a single neuron[43,73]. A recent study employing sCRACM has shown that top-down inputs from V2 to V1 selectively modulate apical dendrites, but not basal dendrites of L5/L6 looped IT neurons that project back to V2[28]. Future studies using sCRACM to investigate the subcellular organization of other top-down inputs in V1—specifically examining whether their synapses are clustered or segregated—are likely to provide valuable insights. Such studies could significantly enhance our understanding of dendritic integration during the interactions among these multiple top-down inputs in V1 neurons.

PT neurons receive extensive inputs from local IT neurons; however, they provide little local feedback, a phenomenon observed in various cortical regions such as the V1, somatosensory, and frontal cortices[39,43,74,75]. Moreover, PT neurons in the barrel cortex are known to receive strong bottom-up TC inputs from the VPM, yet receive few top-down TC inputs from the PO[43,76]. These findings suggest that PT neurons may act as downstream integrators in local circuits: integrating the information from local IT neurons with bottom-up TC inputs and broadcasting the results to subcortical structures.

Recent behavioral studies suggest an alternative interpretation, proposing that IT and PT neurons form parallel subnetworks capable of independent processing. For instance, only PT neurons are required for tactile or visual perception in sensory cortices, while specific PT neurons in the secondary motor cortex are involved in motor generation[38,77–79]. Moreover, widefield calcium imaging assessing the spatial and temporal dynamics of Pyr-type-specific subnetworks during sensorimotor tasks has revealed distinct cortex-wide activity patterns for IT and PT neurons[61,62]. It is possible that information flow between IT and PT in local circuits is dynamically gated by inhibitory and modulatory mechanisms according to behavioral demand.

Building on this notion, our findings suggest a mechanism for the dynamic gating of information flow between IT and PT subnetworks. We found that LP and ORBvl inputs drive mutual inhibition between IT and PT subnetworks in V1. First, we observed that ORBvl inputs drive multisynaptic inhibition in IT neurons. Second, LP inputs induce multisynaptic inhibition in PT neurons (Pyr$_{\leftarrow ORBvl}$ neurons) without multisynaptic excitation from activated IT neurons (Pyr$_{\leftarrow LP}$ neurons), suggesting that LP inputs trigger APs in L5 PV+ neurons and lead to subthreshold excitation in IT neurons. Assuming that the multisynaptic inhibition resulting from LP inputs to PT neurons precedes excitation from activated IT neurons, it is plausible that LP inputs suppress PT neuron activity and thus temporally separate the activity of IT and PT neurons.

The neuronal mechanisms of top-down modulation of visual processing are being investigated actively[19,20,24–27], yet the functional characterization of interactions between multiple top-down inputs is still in its infancy. Our study paves the way for future investigations in behaving animals using advanced imaging and optogenetic manipulation to explore how these top-down inputs interact with each other to enable nuanced task-related top-down modulations.

## Methods
The names of the companies and the catalog numbers for all reagents used in the study are listed in Supplementary Data 12.

### Animals
Animal care and the experimental protocols were approved by the Animal Committee of Shanghai Jiao Tong University School of Medicine and the Animal Committee of the Institute of Neuroscience, Chinese Academy of Sciences. Experiments were performed on wild-type (C57) and transgenic mice. The transgenic mice used were PV-Cre (Jackson lab stock #017320), SST-Cre (#013044), VIP-Cre (#010908), and loxP-flanked-tdTomato (Ai14) (#007914) mice. To visualize the interneurons, PV-, SST-, or VIP-Cre mice were crossed with Ai14 mice. Male and female mice aged 8 to 12 weeks were used. Mice were housed under a 12-h light/dark cycle, with temperatures maintained at 20–24 °C and humidity between 45 to 65%, ensuring optimal welfare and standard laboratory conditions, along with free access to food and water. Our experiments used male and female mice. We found no differences between male and female mice in the layer- and cell-type-specific innervation patterns of CC or TC top-down inputs in V1 or for interactions among these inputs in deep layers. Our experiments used male and female mice. We found no differences between male and female mice in the layer-and cell-type-specific innervation patterns of CC or TC top-down inputs in V1 or for interactions among these inputs in deep layers (Supplementary Figs. 15–18).

### Virus
The adeno-associated viruses AAV-DJ-CaMKIIα-ChR2-EYFP (genomic titer, $1 \times 10^{12}$ gc/mL) was acquired from the WZ Biosciences. Retro-AAV-hSyn-Cre ($5 \times 10^{12}$ gc/mL), Retro-AAV-hSyn-mCherry ($5 \times 10^{12}$ gc/mL), and AAV9-hSyn-ChrimsonR-tdTomato ($5 \times 10^{12}$ gc/mL) were acquired from Brain VTA. All viral vectors were stored in aliquots at −80 °C until use.

## Surgery

**Stereotaxic surgeries.** Adult mice were anesthetized with isoflurane (5% induction and 1.5% maintenance) and placed on a stereotaxic frame (Ruiwode Life Science). The body temperature was maintained at 37 °C throughout the procedure using a heating pad. Eye ointment was applied to protect the animal's eyes. After asepsis, the skin was incised to expose the skull, and the overlying connective tissue was carefully removed. A craniotomy with a diameter of ~0.5 mm was made above the injection site. Viruses were loaded in a sharp micropipette mounted on a Nanoject II attached to a micromanipulator and then injected at a speed of 60 nL per min. Coordinates used were as follows: V2M (Bregma, −2.5 mm; lateral, 1.5 mm; depth, 0.5 mm), ACA (Bregma, +0.3 mm; lateral, 0.3 mm; depth, 0.9 mm), ORBvl (Bregma, +2.5 mm; lateral, 1.3 mm; depth, 1.8 mm), LP (Bregma, −2.0 mm; lateral, 1.3 mm; depth, 2.6 mm), RSP (Bregma, −1.8 mm; lateral, 0.3 mm; depth, 0.5 mm), DMS (Bregma, 0.3 mm; lateral, 1.6 mm; depth, 1.8 mm), Pons (Bregma, −3.1 mm; lateral, 1.2 mm; depth, 4.7 mm), and SC (Bregma, −4.2 mm; lateral, 1.0 mm; depth, 1.8 mm).

**Optogenetics and slice recording.** To examine the input strengths of CC (V2M, ACA, ORBvl) and TC (LP) inputs in V1 Pyrs across layers, we injected an AAV expressing ChR2 in excitatory neurons (AAV-DJ-CaMKIIα-ChR2-EYFP) into the V2M (250 nL), ACA (300 nL), ORBvl (300 nL), and LP (75 nL) of wild-type mice. To examine the input strengths of CC and TC inputs in V1 inhibitory neurons, we injected AAV-DJ-CaMKIIα-ChR2-EYFP in the V2M, ACA, ORBvl, and LP of PV-, SST-, and VIP-Ai14 mice. To independently activate V2M and ACA inputs in the same animal, we injected an AAV expressing ChR2-EYFP in neurons in the V2M (AAV-CaMKIIα-ChR2-EYFP, 250 nL) and also injected an AAV expressing ChrimsonR-tdTomato in excitatory neurons in the ACA (AAV-hSyn-ChrimsonR-tdTomato, 300 nL) of wild-type mice. To independently activate ORBvl and LP inputs in the same animal, we injected an AAV expressing ChrimsonR-tdTomato in neurons in the LP (AAV-hSyn-ChrimsonR-tdTomato, 75 nL) and also injected an AAV expressing ChR2-EYFP in excitatory neurons in the ORBvl (AAV-CaMKIIα-ChR2-EYFP, 300 nL) of wild-type mice. To independently activate ORBvl and LP inputs and examine their input strengths in retrogradely labeled V1 projecting Pyrs, we injected an AAV expressing ChrimsonR-tdTomato in neurons in the LP (AAV-hSyn-ChrimsonR-tdTomato, 75 nL), injected an AAV expressing ChR2-EYFP in excitatory neurons in the ORBvl (AAV-CaMKIIα-ChR2-EYFP, 300 nL), and also injected a Retro-AAV expressing mCherry (Retro-AAV-hSyn-mCherry) into the RSP (300 nL), ipsi-DMS (300 nL), or SC (300 nL) of wild-type mice. To examine the EPSCs and IPSCs in L5 Pyrs elicited by ORBvl and LP inputs, we injected an AAV expressing ChR2 in excitatory neurons (AAV-DJ-CaMKIIα-ChR2-EYFP) into the ORBvl (300 nL) and LP (75 nL) of wild-type mice.

**Retro-AAV-mediated retrograde tracing.** For retrograde tracing from cortical and subcortical areas, we injected a retro-AAV expressing Cre (Retro-AAV-hSyn-Cre) into the V2M (100 nL), RSP (200 nL), ACA (300 nL), ORBvl (300 nL), ipsi-DMS (200 nL), contra-DMS (200 nL), Pons (100 nL), SC (100 nL), and LP (50 nL) of Ai14 mice. The histology experiments were performed 4 weeks after retro-AAV injection.

## In vitro electrophysiology

**Slice preparation.** Mice were anesthetized with 5% isoflurane. After decapitation, the brain was quickly dissected and immediately placed in ice-cold oxygenated NMDG-HEPES solution (in mM: NMDG 93, KCl 2.5, NaH₂PO₄ 1.2, NaHCO₃ 30, HEPES 20, glucose 25, sodium ascorbate 5, thiourea 2, sodium pyruvate 3, MgSO₄.7H₂O 10, CaCl₂.2H₂O 0.5 and NAC 12, at pH 7.4, adjusted with HCl), and coronal sections of brain slices were made with a vibratome. Slices (250-μm thick) were recovered in oxygenated NMDG-HEPES solution at 32 °C for 10 min and then maintained in an incubation chamber with oxygenated standard artificial cerebrospinal fluid (ACSF) (in mM: NaCl 125, KCl 3, CaCl₂ 2, MgCl₂ 1, NaH₂PO₄ 1.25, sodium ascorbate 1.3, NaHCO₃ 26, glucose 10) at 30 °C for 1–4 h before recording.

**Whole-cell recording.** Whole-cell recordings were made at 30 °C in oxygenated standard ACSF. To examine the monosynaptic input strength of each CC and TC input in V1 neurons, EPSPs were recorded using a potassium-based internal solution (in mM: K-gluconate 135, KCl 5, HEPES 10, EGTA 0.3, MgATP 4, Na₂GTP 0.3, and Na₂-phosphocreatine 10, at pH 7.3, adjusted with KOH, 290–300 mOsm). TTX (1 μM) and 4-aminopyridine (100 μM) were bath applied to block action potentials and permit direct depolarization of axon terminals by ChR2 activation with 5-ms pulses of blue light[43]. In experiments where LP and ORBvl inputs were activated alternately, EPSCs were also recorded using this potassium-based internal solution.

To examine the EPSCs and IPSCs in L5 Pyrs evoked by LP and ORBvl inputs, we used a cesium-based internal solution (in mM: CsMeSO₄ 125, CsCl 2, HEPES 10, EGTA 0.5, MgATP 4, Na₂GTP 0.3, Na₂-phosphocreatine 10, TEACl 5, QX-314 3.5, at pH 7.3, adjusted with CsOH, 290–300 mOsm). EPSCs and IPSCs were isolated by clamping the membrane potential of the recorded neuron at the reversal potential of inhibitory and excitatory synaptic currents, respectively. To measure the inhibition on L5 Pyrs induced by various cortical inhibitory neuron types, IPSCs were also recorded using this cesium-based internal solution.

The resistance of the patch pipette was 3–5 MΩ. The cells were excluded if the access resistance exceeded 40 MΩ or varied by more than 20% during the recording period. Data were recorded with a Multiclamp 700B amplifier (Axon Instruments), filtered at 2 kHz, and digitized with a Digidata 1550B (Axon Instruments) at 10 kHz. Recordings were analyzed using custom software.

After completion of the electrophysiological recordings, the vertical cell depth from the pial surface was measured. Based on the measured distance, we assigned the recorded neurons to specific cortical layers. The layer boundaries were determined based on Allen Mouse Brain Atlas[80].

For the experiments in Figs. 1, 2, we recorded neurons simultaneously in each layer in at least five mice in each group. For the majority of recorded animals, we recorded neurons simultaneously in three or more layers. The numbers of mice recorded with different numbers of layers in Fig. 1 are listed in Supplementary Data 13.

**Data analysis.** The resting membrane potential (RMP) was recorded soon after break-in at I = 0. During voltage-clamp recordings (holding at −70 mV), a −10 mV voltage step lasting 300 ms was applied to compute both the input resistance (Rin) and the access resistance (Ra). The initial transient current response to this voltage step ($\Delta I_{transient}$) was used to derive the Ra, while the steady-state current that followed the transient ($\Delta I_{steady}$) was used to derive the Rin. Both resistances were determined using Ohm's law: Ra = $\Delta V/\Delta I_{transient}$; Rin = $\Delta V/\Delta I_{steady}$, where $\Delta V$ indicates the −10 mV voltage change. During this −10 mV voltage step, we observed the $I_h$. The slope of $I_h$ was calculated as $I_h$ slope = $\Delta I_{5-265}/\Delta t$. Where $\Delta I_{5-265}$ is the change in current from 5 to 265 ms during the −10 mV voltage step, and $\Delta t$ is its duration (260 ms).

During current clamp recordings, we analyzed the properties of the action potential (AP). The instantaneous firing rate was calculated as the reciprocal of the inter-spike interval (ISI) of the first two spikes observed during a series of depolarizing current steps (25 pA/step, each lasting 500 ms). Regular spiking was defined as generating single APs in response to near-threshold depolarization, whereas burst spiking was defined as generating a complex of two or more APs in response to near-threshold depolarization. Single AP induced by a brief depolarizing current step (2 pA, 25 ms) was analyzed for AP threshold and depolarizing afterpolarization (DAP). The AP threshold was identified as the voltage where there was a sharp increase in the

rate of voltage change. The membrane voltages were plotted against their first-time derivative (dV/dt, phase-plane plot), and the AP threshold was selected as the voltage at which dV/dt exceeded three times the standard deviation of all the preceding data points. The amplitude of DAP was calculated as the peak amplitude of DAP relative to the minimum value of the fast afterhyperpolarizations[81]. In cases where DAPs were absent, we marked them with an amplitude of 0, as described by Hattox and Nelson (2007).

## Optogenetic manipulation

For optogenetic activation of axons from CC and TC top-down inputs in V1, we used an X-cite LED (Lumen Dynamics Group) controlled by a stimulator (Master8). We activated ChR2 using blue light bandpass filtered at 419–465 nm (Semrock) and ChrimsonR with red light filtered at 610–650 nm (Semrock). This filtered light was delivered through a 40× 0.8 NA water immersion lens. To systematically characterize CC and TC top-down inputs in V1 neurons, we delivered pulse trains of blue light (3.5 mW, 10 Hz, 5 ms). For independent activation of ORBvl and LP inputs using ChR2 and ChrimsonR within the same brain slice, we alternated between pulse trains of blue light (3.5 mW, 10 Hz, 1 ms) and red light (3.5 mW, 10 Hz, 1 ms). For measuring EPSCs and IPSCs in L5 Pyrs driven by ORBvl and LP inputs, we used pulse trains of blue light (3.5 mW, 10 Hz, 5 ms).

## Histology and anatomical data analysis

To visualize the distribution of axons from CC and TC top-down inputs in fluorescence images, we used the following procedures: Mice were deeply anesthetized using isoflurane and promptly perfused with chilled 0.1 M phosphate-buffered saline (PBS), followed by a 4% paraformaldehyde (w/v) solution in PBS. The brain was then extracted and postfixed in the same solution overnight at 4 °C. After fixation, the brain was immersed in a 30% sucrose (w/v) solution in PBS and kept for 1–2 days at 4 °C. Once embedded and frozen, the brain was sectioned into 40-μm-thick coronal slices with a cryostat. After preparing the slices, they were rehydrated in PBS for 30 min and mounted using VECTASHIELD mounting medium infused with 4′,6-diamidino-2-phenylindole (DAPI). Imaging was carried out using high-throughput slide scanners (VS120, Olympus) for comprehensive visualization and subsequent analysis. In addition, select representative slices were imaged using a confocal microscope (Olympus FV-3000). The same procedures were also used to visualize the distribution of V1 neurons retrogradely labeled from various cortical and subcortical downstream targets.

For the quantification of axon intensity in V1 shown in Figs. 1, 3 animals were analyzed for each input (two slices per animal). Vertical fluorescence profiles of EYFP- or tdTomato-expressing axons were measured using ImageJ after subtracting background fluorescence from a hippocampal area devoid of labeled axons.

To visualize the recorded Pyr$_{\leftarrow ORBvl}$ and Pyr$_{\leftarrow LP}$ neurons, we used the following procedures: After whole-cell recording, slices with cells filled with biocytin were fixed in 4% paraformaldehyde for 1 h at room temperature. Slices were then rinsed in 0.01 M PBS three times, and transferred to 0.5% Triton X-100 for 5 min and then incubated in a blocking solution (5% BSA in PBS) for 1 h. After that, slices were incubated with streptavidin (1:1000, DyLight 549 Streptavidin, Cat. No: SA-5549, Vector Laboratories, Burlingame, CA) overnight at 4 °C. For 3D reconstruction of labeled cells, z-stack images (0.5 μm per image) were acquired with a 20× air objective on a confocal microscope (Olympus FV-3000) and processed using ImageJ.

## Classification based on SVM classifier

SVM classification is implemented with the fitcsvm function in the MATLAB statistics toolbox using a linear kernel and a uniform prior. We used the I$_h$-slope data of 82 Pyr$_{\leftarrow LP}$ and 65 Pyr$_{\leftarrow ORBvl}$ neurons, derived from a dataset of individual activation of LP and ORBvl inputs

(from Fig. 1k–t), to train the "SVM Classifier(I$_h$)". The classifier performance was evaluated via tenfold cross-validation using the same I$_h$-slope data. The classifier performance is also tested on 1000 instances of shuffled labels to obtain the 95% confidence interval. We then used the SVM Classifier(I$_h$) to classify Pyr$_{\leftarrow LP}$ and Pyr$_{\leftarrow ORBvl}$ neurons based on their I$_h$ slope in subsequent experiments. The prediction accuracy of the SVM Classifier(I$_h$) is calculated as the number of correct predictions divided by the total number of predictions.

## Patch-seq recording and RNA sequencing of patched cells

Patch-seq recording was performed following the method described by ref. 53. Briefly, following electrophysiological recording, single L5 Pyrs were collected immediately by applying light negative pressure through the same glass patch pipette. Special care was taken to maintain the integrity of the seal between the pipette and the cell membrane throughout the procedure to prevent contamination from the extracellular environment. Only cells whose entire somatic compartment (including their nucleus) were visibly aspirated into the micropipette were further processed.

Following successful aspiration, the contents of the pipette were immediately ejected into an RNase-free PCR tube containing 4 μL lysis buffer and stored immediately on dry ice until −80 °C storage. cDNA libraries were produced using a Smart-seq2-based Single Cell Full-Length mRNA-Amplification Kit (N712, Vazyme) according to the manufacturer's instructions. The size distributions and concentrations of the cDNA libraries were assessed using an Agilent Bioanalyzer 2100. Samples with less than 1 ng of total cDNA in the final volume of 15 μL, or with an average size smaller than 1500 bp, were not sequenced. To construct the final sequencing libraries, 1 ng of purified cDNA from each sample was tagmented using an Illumina Nextera XT Library Preparation kit (TD503 Vazyme). The DNA was sequenced from paired ends (150 bp) with standard Illumina Nextera i5 and i7 index primers (8 bp each) using an Illumina Hiseq xten instrument.

50-bp paired-end reads were aligned to GRCm38 (mm10) using a RefSeq annotation gff file (https://www.ncbi.nlm.nih.gov/genome/annotation_euk/all). Sequence alignment was performed using STAR v2.5.3[82] in two pass mode. PCR duplicates were masked and removed using the STAR option 'bamRemoveDuplicates'. Only uniquely aligned reads were used for gene quantification. Gene counts were computed using the R Genomic Alignments package[83].

## Transcriptome data analysis

A total of 91 neurons were sequenced in our study. We excluded four cells as the sum of counts across all genes in each of these cells was below 500. Following this, we log-transformed all counts using a log2(x + 1) transformation[54] and initially assessed the specificity of our sampling based on the expression levels of pan-neuronal genes (Actb, Syt4, Atp1a3, Syt1, Calm1, Arpp21, and Snap25), cortical excitatory neuron genes (Gria2, Camk2a, and Slc17a7), GABAergic inhibitory neuron genes (Slc32a1, Gad1, and Gad2) and glial marker genes (Adarb2, Gfap, Aqp4, and Mlc1) in single neuronal transcriptomes of all neurons (Supplementary Fig. 9). An additional seven cells were excluded due to high levels of inhibitory neuron genes. This was defined as the sum of the log2(x + 1) value of the three inhibitory neuron marker genes exceeding 7.

The raw read count table was then used to identify the differentially expressed genes between the Pyr$_{\leftarrow ORBvl}$ and Pyr$_{\leftarrow LP}$ neurons using the DESsq2 v.1.40.2 R package[84]. We excluded genes expressed in fewer than ten cells (count 5). Genes with q values < 0.05 and either FC > 2 or FC < 0.5 were considered as differentially expressed genes (DEGs).

## Quantification and statistical analysis

Statistical analysis was performed using MATLAB, SPSS, and R. The selection of statistical tests was based on previous studies. All

**Article** https://doi.org/10.1038/s41467-024-48924-8

statistical tests were two-sided. The exact number of mice and recorded cells were described in figure legends and source data. Statistical method, statistics, and corresponding *P* values were reported in the figure legends and Supplementary Data.

### Reporting summary

Further information on research design is available in the Nature Portfolio Reporting Summary linked to this article.

## Data availability

The data supporting the findings of this study are included in the figures and supporting files. The sequencing data generated in this study have been deposited in the Gene-Expression Omnibus repository under the accession code GSE246589. Source data are provided with this paper.

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

## Acknowledgements

We thank J. Lu for the critical reading of our manuscript. We thank A. Fei, R. Liu, and H. Bao from Q. Li's lab for the kind help on the RNA sequencing data analysis. We thank Y. Wang from the Optical Imaging Core Facility at Shanghai Jiao Tong University School of Medicine for her help in image acquisition and analyses. This work was supported by grants from STI2030-Major Projects (2021ZD0203704 and 2021ZD0202804 to S.Z.), the National Natural Science Foundation of China (32170993 to S.Z. and 32200811 to Y.L.), the Science and Technology Commission of Shanghai Municipality (21ZR1436400 to S.Z.), China Postdoctoral Science Foundation (2022M722135 to Y.L.), the innovative research team of high-level local universities in Shanghai, and the Shanghai Frontiers Science Center of Cellular Homeostasis Regulation and Human Diseases. We thank scidraw.io for illustrations: the mouse head schema (in Figs. 3a, e, 4a, e, 5a, c, g, 7a, 8a, e, and Supplementary Fig. 10a, d, g) is from Luigi Petrucco (https://doi.org/10.5281/zenodo.3925903); the pyramidal neuron in Fig. 6a is from Federico Claudi (https://doi.org/10.5281/zenodo.3925905); the Eppendorf tube in Fig. 6a was adapted from Diogo Losch De Oliveira (https://doi.org/10.5281/zenodo.3925953); the DNA strand in Fig. 6a was adapted from Guilherme Bauer Negrini (https://doi.org/10.5281/zenodo.3926245).

## Author contributions

G.M. and S.Z. designed the experiments. G.M. performed and organized all the experiments. G.M. and Y.L. performed whole-cell recording experiments. G.M. and J.Z. performed the patch-seq experiments. G.M., Y.L., and J.Z. performed the imaging and circuit-mapping experiments

with assistance from Z.J. and M.Q. G.M. performed the data analysis. Y.L., G.M., M.X., and S.Z. wrote the manuscript. All authors read and revised the manuscript.

## Competing interests

The authors declare no competing interests.
