## [Peer Review File · Nature Communications]

Organization of Corticocortical and Thalamocortical Top-down Inputs in the Primary Visual CortexREVIEWER COMMENTS

Reviewer #1 (Remarks to the Author):

This paper utilized optogenetic stimulation to investigate the inputs to V1 neurons, with a specific focus on what can be classified as 'top-down' inputs—inputs carrying signals not directly originating from the retinogeniculate pathway. Among the four input pathways studied, two of them (V2M and ACA) targeted both L2/3 and L6 neurons, while the other two (ORBvl and LP) targeted L5 neurons, aligning with axonal projection patterns. These findings are presented in Figures 1 and 2, which could be valuable for researchers interested in these projections.

Interestingly, the remainder of the paper primarily concentrates on ORBvl and LP inputs, disregarding the V2M and ACA pathways for reasons that are not explicitly stated. Within this scope, the authors made an intriguing discovery, noting that distinct groups of neurons in L5 may receive input from one source but not the other. This observation draws parallels with previous findings in the barrel cortex, where earlier research focused on subcellular analysis, while the current study stimulates axons as a whole (Petreanu, Leopoldo, Tianyi Mao, Scott M. Sternson, and Karel Svoboda. 2009. "The Subcellular Organization of Neocortical Excitatory Connections." *Nature* 457 (7233): 1142–45. <https://doi.org/10.1038/nature07709>).

Figure 3 presents results that might not be entirely surprising, as the inputs target different sublayers, and the authors did not provide a detailed account of the recorded cell locations. Notably, some cellular distinctions emerged between neurons receiving LP and ORBvl inputs, with the latter exhibiting a distinctive sag associated with hyperpolarization. To classify cell types based on cellular properties, the authors employed a Support Vector Machine (SVM) method, which yielded satisfactory results. However, considering the simplicity of the data (with only one feature and a sample size ranging between 10 to 100), other classification methods, such as logistic regression or decision trees, could also prove effective.

Main:

The paper's overall structure may benefit from some improvements, particularly in terms of motivation. Currently, the paper lacks a compelling rationale for its focus. If the primary objective is to comprehend long-range 'top-down' inputs to V1 neurons, there appears to be no clear justification for concentrating solely on the ORBvl vs. LP pathways. The authors themselves noted in the paper:

"Note that our optogenetic activation profiling of CC and TC top-down inputs revealed that LP and ORBvl inputs each activate about half of the L5 Pyrs (Extended Data Fig. 1); this prompted us to investigate the specificities of LP and ORBvl inputs in L5 of V1."

This explanation may not provide a strong incentive for the subsequent content of the paper, as it essentially comprises the remainder of the manuscript. When reading the paper, it is challenging to become fully engaged in the questions the authors aim to answer. Instead, the paper appears to present a series of experiments without a clearly defined overarching goal.

Furthermore, the methods employed in the paper might not represent the best available tools for the job. Notably, other researchers have utilized sCRAM (as seen in the Petreanu/Svoboda paper), which offers a more detailed perspective of input locations. However, it's understandable that these experiments might not be feasible due to potential resource constraints.

Minor:

1. The authors should include information about the specific locations of the neurons they recorded, mapping them to the corresponding layers and indicating their relative positions in relation to layer boundaries. Notably, Layer 5 can often be further subdivided into multiple sublayers, with different neuron types residing in distinct sublayers. If ORBvl-receiving neurons are recorded from a different layer compared to LP-receiving neurons, it's crucial to acknowledge that the study might effectively be examining neurons from two different sublayers. The presence of distinct properties in such cases would not be surprising.

2. In relation to the SVM model training, the authors mention: "The performance of the SVM Classifier (lh) was evaluated using 10-fold cross-validation. The detected average accuracy of 88.2% (over 20 cross-validations) significantly surpassed the chance level of 50% ($P = 2 \times 10^{-26}$, paired t-test)."

a. It's important to clarify how the authors arrived at "20 cross-validations" when using the 10-fold cross-validation method. In 10-fold cross-validation, the data is typically divided into 10 sets for training (9 sets) and testing (1 set). Therefore, they should report 10 validation results, not 20.

b. Explain how the "paired t-test" was conducted on the data when the data is not intrinsically paired. If the data involved shuffling and performing k-fold cross-validation on the shuffled data, it may not qualify as a "paired" situation in the statistical sense. Paired tests generally pertain to the same sample subjected to two different conditions, with interest in the difference between them. This concept does not align with the conditions presented in this study.

c. Clarify which data were used to train the final SVM model that was later applied to another dataset. State whether all 82+65 samples were used in the training process.

Reviewer #2 (Remarks to the Author):

Using slice electrophysiology and optogenetics, Liu et al. initially examined the connectivity pattern of top-down inputs from V2M, ACA, ORBvl, and LP to pyramidal and inhibitory (VIP, PV, and SST) neurons in different layers in V1. The authors then further examined the ORB and LP inputs since the connectivity data suggested a potential cell class effect. The authors unveiled 1) V2M and ACA mainly innervated neurons in L6, 2) ORBvl and LP each innervated L5 pyramidal neurons with 50% chance, 3) ORB weakly innervates inhibitory neurons, 4) LP innervates VIP neurons in all layers (but strongest in L6) and PV

neurons in L5, 5) L5 pyramidal neurons defined by their inputs (i.e., Pyr<-LP and Pyr<-ORB) can be distinguished by their electrophysiological and genetic properties, and projection to cortical and subcortical sites, respectively, and 6) Pyr<-LP and Pyr<-ORB neurons are differentially inhibited by their upstream inputs due to recruitment of local PV neurons. Based on these, the authors claimed different top-down inputs engage different V1 subnetworks, each having an effect on different downstream targets (cortical or subcortical).

The major strength of the paper is a rigorous and laborious quantitative analysis of the specific connectivity formed between multiple top-down afferents (in particular, LP and ORB axons) with different L5 pyramidal and inhibitory cell types (Fig. 1 and 2). Moreover, the demonstration of the absence of LP and ORB input convergence at the level of single neurons (Fig. 3) and finding that each afferent exerts mostly a multisynaptic inhibitory effect on unconnected L5 pyramidal neurons (Fig. 7) is interesting.

The weakness of the paper is its novelty in several aspects. For example, V2M input to V1 has been studied with a similar technique but in slightly different contexts by Young et al. 2021 (not cited). Young et al. reported reasonable input from V2M to V1 L5 pyramidal neurons (their Fig. 4) in addition to several other connectivity details relevant to the current study, thus should be discussed. Moreover, although I appreciate the authors' laborious efforts, the finding that the Pyr<-LP and Pyr<-ORB neurons are differentiated by intrinsic properties and genetic markers (Fig. 4 and 5), and they have a different projection pattern (Fig. 6) is over-highlighted. The authors were “surprised (page 8 line 8)” by the differential expression of Ih between these cells, despite the motivation to further analyze these cells being the assumption that they fall into PT and IT cell classes. Indeed, PT and IT classes are well-known to be distinguished by the presence of Ih (and other intrinsic properties), genetic markers, and projection patterns (e.g., ref 36), thus, I do not think the findings (Fig 4-6) were as surprising as the authors claim.

Also, it is unclear whether the neurons connected and unconnected with LP axons occur at the same laminar position in L5, because there is clear projection bias for each afferent (LP axons target L5A while ORB targets L5B). L5A contains only IT neurons, so connection bias for LP axons is difficult to be explained by cell class (as the authors alluded) if the recording is done at the L5A for LP inputs assessment.

Other comments:

- In Fig 6 (and patch-seq experiment), the authors seem to define the Pyr<-LP or Pyr<-ORB neurons based solely on SVM and Ih (i.e., without verifying the connectivity). Although I understand the predictive power of the classifier, after all, it only uses one parameter (Ih slope) and I have some reservations about whether this approach qualifies to classify the neurons defined by its input type.
- The authors are focused on the idea of the subnetwork; I'm wondering if these parallel (and potentially competing) circuit may interact/converge at the level of the local circuit since Pyr<-LP neurons (IT type) is

likely to be upstream of Pyr<-OBR neurons (PT type) based on current knowledge of local circuit organization in cortex in general.

- Fig. 1 and 2: Please state how the recording is made to obtain input data across the different layers. Did the authors record at least one neuron in each layer in one slice or mice? The sample size for the cells appears to vary from layer to layer and is typically heavily skewed to the layer where the most inputs are detected. Why?

- How many animals were used for each recording (in non-supplemental figures)? What is the maximum/minimum cell number represented by one animal? Did the authors examine the sex effect?

- How many animals were used for each recording (in non-supplemental figures)? What is the maximum/minimum cell number represented by one animals? Did the authors examine the sex effect?

Reviewer #3 (Remarks to the Author):

In this manuscript, Liu et al. employed optogenetic-assisted circuit mapping to investigate top-down inputs from multiple higher-order regions into various anatomically or genetically distinct neurons in the primary visual cortex (V1). They discovered that neurons in the medial part of secondary visual cortex and the anterior cingulate area predominantly activate neurons in layer 6 (L6), whereas the ventrolateral orbital cortex (ORBvl) and the lateral posterior nucleus of the thalamus (LP) primarily target neurons in layer 5 (L5). Within L5, neurons that receive inputs from ORBvl and LP exhibit notable segregation, distinguished by distinct molecular profiles and electrophysiological characteristics. These findings offer valuable insights for those interested in information processing within V1. Nevertheless, this study constitutes an incremental intellectual progression from their previous work published in Science Advances in 2021. This manuscript might be more suited to a specialized journal.

Reviewer #4 (Remarks to the Author):

In this manuscript, Liu, Zhang, and colleagues present valuable findings on the synaptic organization of corticocortical and thalamocortical top-down inputs onto mouse V1 neurons. The authors characterized the connectivity and strengths of different synaptic inputs and made an exciting discovery of two parallel processing pathways from ORBvl to V1 L5 to subcortical areas and LP to V1 L5 to cortical areas. The evidence supporting the claims of the authors is overall solid, but requires additional validation (see below). Overall, this is an excellent study with some weaknesses that can be addressed.

1. For the experiments in Figure 1, the authors should clarify if neurons in L2/3, L4, L5, and L6 were recorded simultaneously in response to the stimulation of presynaptic terminals. This seems not the case based on very different numbers of recorded neurons from different layers. These experiments depend on the viral expression of Chr2 in different cortical and thalamic areas, which varies between mice and

slices and can contribute to the difference between different recordings. If neurons in different layers were not recorded simultaneously, then the authors should demonstrate how such variabilities did not cause the difference observed between layers for each of the input profile. In other words, the authors should validate their findings and rule out this possibility.

2. Related to Point 1, Extended Data Fig 2 shows the normalized EPSP amplitudes to L6 Pyrs recorded on the same brain slice. It is unclear how this normalization was done because one would expect that all the values from L6 would be 1.

3. The CaMKIIa promoter is not specific to excitatory neurons (see <https://www.eneuro.org/content/10/4/ENEURO.0070-23.2023>). It's not a problem to use it for ACA, ORBvl, and LP, but it could be problematic for V2M because some interneuron axons may be present in the V1, which can affect the size of EPSPs. The authors should test this possibility.

4. Similar to Point 1, for the experiments in Figure 2, the authors also need to demonstrate that the variability of viral expression of Chr2 does not cause the observed difference between layers and interneuron types. One possibility is to use an independent neuronal type as the reference for normalization.

5. The authors should also clarify if the experiments in Figure 7a/e were done on two neurons simultaneously in the same slice, because the schematics imply that.

6. The statement "...LP inputs induced significantly stronger inhibition in Pyr←ORBvl neurons compared to Pyr←LP neurons (shown by the I/(I+E) ratio, Fig. 7c)..." is not accurate because this ratio does not compare the inhibition directly. Similarly, the statement "...ORBvl inputs induced significantly stronger inhibition of Pyr←LP neurons compared to Pyr←ORBvl neurons (Fig. 6g, it should be 7g)..." is also not accurate. If the authors recorded the two neurons simultaneously, then the IPSCs can be directly compared.

7. Based on the findings in Figure 7, one would expect that activation of LP inputs would cause hyperpolarization of Pyr←ORBvl neurons, which explains the negative values in Figure 1s L5 neurons. The authors should mention this. This applies to ORBvl inputs and Figure 1n.

8. It would be helpful if the authors provide the numbers of mice and slices used in each experiment in addition to neurons.

9. The statistical methods should be justified. For example, in Figure 1, t-test is not appropriate and one-way ANOVA should be used to compare different layers.

10. The authors may consider additional motivations for this study. For example, more rationales for choosing these particular corticocortical and thalamocortical top-down inputs would be helpful to the readers.

We would like to express our sincere gratitude for the efforts of the Editor and the Reviewers in examining our study. The insightful comments have been very helpful as we conducted new experiments, new analyses, and extensively revised the manuscript. Before getting into our full point-by-point responses, we offer the following summary of our major updates to the study and manuscript:

1. We have now revised our manuscript to better clarify the motivation behind the various experiments. To justify our specific focus on the LP and ORBvl inputs in the latter half of our manuscript, we have included a new experiment employing independent optogenetic activation of V2M and ACA inputs. This experiment demonstrates that V2M and ACA inputs converge on L6 Pyrs, in sharp contrast to the parallel processing of LP and ORBvl inputs in two distinct types of L5 Pyrs. Given the notable specificity with which L5 Pyrs process LP and ORBvl inputs, we narrowed our focus to exploring the properties of $\text{Pyr}_{\leftarrow\text{LP}}$ and $\text{Pyr}_{\leftarrow\text{ORBvl}}$ neurons. In addition to the new experiment, we have added a new main figure (new Figure 3) and have made significant revisions to the original abstract, introduction, and discussion, as well as to one subsection in the results.
2. Based on the reviewer's guidance, we have reanalyzed our data and generated plots of the normalized EPSP amplitudes in V1 Pyrs and inhibitory neurons for each of the top-down inputs (see new Extended Data Figs. 3 and 4). Results obtained with this normalization approach are consistent with the previously detected differences in input strengths across layers and cell types as reported in the originally submitted manuscript.
3. We have added additional results showing the specific locations of the neurons we recorded in both the main figures and supplementary figures, including Figs. 3 and 4, as well as in Extended Data Figs. 1,2,6,9, and 11. Notably, $\text{Pyr}_{\leftarrow\text{LP}}$ and $\text{Pyr}_{\leftarrow\text{ORBvl}}$ neurons extensively overlap in L5 of V1, rather than being segregated into different sublayers.
4. We have added additional control experiments to validate the predictions of the SVM Classifier in identifying $\text{Pyr}_{\leftarrow\text{LP}}$ and $\text{Pyr}_{\leftarrow\text{ORBvl}}$ neurons in retrogradely labeled L5 Pyrs projecting to the RSP, ipsi-DMS, and SC. In each group, the SVM Classifier(I_h) achieved 100% accuracy, reinforcing that the SVM Classifier(I_h) is an effective tool for distinguishing $\text{Pyr}_{\leftarrow\text{LP}}$ and $\text{Pyr}_{\leftarrow\text{ORBvl}}$ neurons (new Extended Data Fig. 10).
5. We have added the information about the number of mice and slices used in each experiment in the figure captions and supplementary tables.
6. We have reanalyzed our data to examine any sex-specific effects. We found no differences between male and female mice in the layer- and cell-type-

specific innervation patterns of CC and TC top-down inputs in V1 (Reviewer Figs. 3,4) or for interactions among these inputs in deep layers (Reviewer Figs. 5,6).

7. We have conducted a control experiment to examine the strengths of V2M inputs in relation to the distance between the recorded neurons and the V2M border, as shown in Reviewer Fig. 1. The responses of V1 L5 Pyrs to V2M inputs are significantly stronger within 400 μm of the V2M or V2L border, compared to those located 450-900 μm away from the V2M border. Thus, the differences in V2M \rightarrow V1 input strength observed between Young *et al.*'s study (2021) and ours can be plausibly attributed to variation in the recording sites within V1.
8. We have conducted a control experiment regarding the use of the CaMKII α promoter for V2M inputs: bath application of bicuculline did not change the EPSP amplitude evoked by V2M input activation in Pyrs in V1, supporting that the recorded area in our experiments was sufficiently distant (500-1000 μm away) from the V1/V2M border to preclude influence from V2M inhibitory neurons (Reviewer Fig. 7).
9. We have removed the violin plots and retained only box plots from original Figures 1 and 2 to avoid the misleading information caused by the extension of violin plots into the negative values. The box plots clearly show that the lower limit of our data is zero.
10. We have re-examined the statistical analysis methods and corrected previous errors. First, we have replaced the previous used *t*-test with a one-way ANOVA and Tukey's post hoc test to compare the EPSP amplitudes among different layers in revised Fig.1 and Extended Data Fig. 3. Second, we have replaced the previously used paired *t*-test with a one-sample *t*-test to compare the prediction accuracy of SVM Classifier(I_h) and the chance level of 50%. The results based on these corrected statistics support our conclusions in the originally submitted version.
11. We have cited a recent study that used sCRACM to investigate V2-V1 interactions (Young *et al.*, 2021) in the revised introduction, results, and discussion sections. We have also added a new paragraph in the discussion section to elaborate on the potential insights that could be gained from employing higher-resolution stimulation techniques like sCRACM in future studies of circuitry mechanisms underlying top-down modulation in V1.
12. We have added content to the revised discussion section to elaborate: 1) the local and long-range subnetworks formed by IT and PT neurons; 2) the dynamic gating of information flow between IT and PT subnetworks

observed in behaving animals; 3) how our findings about mutual inhibition between these two subnetworks as driven by top-down inputs contributes to understanding mechanisms underlying dynamic gating.

The responses to Reviewer#1 begin on Page 4, the responses to Reviewer#2 start from Page 24, the responses to Reviewer#3 start from Page 50, and the responses to Reviewer#4 start from Page 54.

Reviewer #1 (reviewer's comments in italic):

This paper utilized optogenetic stimulation to investigate the inputs to V1 neurons, with a specific focus on what can be classified as 'top-down' inputs—inputs carrying signals not directly originating from the retinogeniculate pathway. Among the four input pathways studied, two of them (V2M and ACA) targeted both L2/3 and L6 neurons, while the other two (ORBvl and LP) targeted L5 neurons, aligning with axonal projection patterns. These findings are presented in Figures 1 and 2, which could be valuable for researchers interested in these projections.

*Interestingly, the remainder of the paper primarily concentrates on ORBvl and LP inputs, disregarding the V2M and ACA pathways for reasons that are not explicitly stated. Within this scope, the authors made an intriguing discovery, noting that distinct groups of neurons in L5 may receive input from one source but not the other. This observation draws parallels with previous findings in the barrel cortex, where earlier research focused on subcellular analysis, while the current study stimulates axons as a whole (Petreanu, Leopoldo, Tianyi Mao, Scott M. Sternson, and Karel Svoboda. 2009. "The Subcellular Organization of Neocortical Excitatory Connections." *Nature* 457 (7233): 1142–45. <https://doi.org/10.1038/nature07709>).*

Figure 3 presents results that might not be entirely surprising, as the inputs target different sublayers, and the authors did not provide a detailed account of the recorded cell locations. Notably, some cellular distinctions emerged between neurons receiving LP and ORBvl inputs, with the latter exhibiting a distinctive sag associated with hyperpolarization. To classify cell types based on cellular properties, the authors employed a Support Vector Machine (SVM) method, which yielded satisfactory results. However, considering the simplicity of the data (with only one feature and a sample size ranging between 10 to 100), other classification methods, such as logistic regression or decision trees, could also prove effective.

Main:

1) *The paper's overall structure may benefit from some improvements, particularly in terms of motivation. Currently, the paper lacks a compelling rationale for its focus. If the primary objective is to comprehend long-range 'top-down' inputs to V1 neurons, there appears to be no clear justification for concentrating solely on the ORBvl vs. LP pathways. The authors themselves noted in the paper:*

"Note that our optogenetic activation profiling of CC and TC top-down inputs revealed that LP and ORBvl inputs each activate about half of the L5 Pyrs (Extended Data Fig. 1); this prompted us to investigate the specificities of LP and ORBvl inputs in L5 of V1."

This explanation may not provide a strong incentive for the subsequent content of the

paper, as it essentially comprises the remainder of the manuscript. When reading the paper, it is challenging to become fully engaged in the questions the authors aim to answer. Instead, the paper appears to present a series of experiments without a clearly defined overarching goal.

RESPONSE:

We thank the reviewer for pointing out this issue. In response, we have revised our manuscript to better clarify the motivation behind the various experiments. Our revised story focuses on exploring how different long-range top-down modulation signals are processed in the V1 local circuit across various layers and their interactions within their most responsive layers. This exploration begins with an examination across Pyrs and the primary types of inhibitory neurons in each layer, then delves into the subpopulations of Pyrs in deep layers.

To justify our specific focus on the LP and ORBv1 inputs in the latter half of our manuscript, we have included a new experiment employing independent optogenetic activation of V2M and ACA inputs. This experiment demonstrates that V2M and ACA inputs converge on L6 Pyrs, in sharp contrast to the parallel processing of LP and ORBv1 inputs in two distinct types of L5 Pyrs. These findings demonstrate distinct processing strategies during the interaction of top-down inputs, with V2M and ACA inputs undergoing integrated processing in L6, in contrast to the parallel processing of LP and ORBv1 inputs in L5.

Given the notable specificity with which L5 Pyrs process LP and ORBv1 inputs, we narrowed our focus to exploring the properties of $\text{Pyr}_{\text{T} \leftarrow \text{LP}}$ and $\text{Pyr}_{\text{T} \leftarrow \text{ORBv1}}$ neurons. After an in-depth characterization of their electrophysiological properties, gene expression profiles, and axon innervation patterns, we concluded that they likely represent well-known Pyr subtypes: IT and PT neurons. This finding establishes a link between the parallel processing of LP and ORBv1 information in V1 L5 and IT and PT subnetworks. Such insights into the selective innervation patterns within L5 of V1, combined with the identification of distinct neuronal subclasses, add a new dimension to our understanding of the functional roles of these neurons, by revealing a more nuanced understanding of top-down modulation in visual processing.

In addition to the new experiment, we have added a new main figure and have made significant revisions to the original abstract, introduction, and discussion, as well as to one subsection in the results.

We have revised the original abstract to include the results from the new experiment testing the interaction between V2M and ACA inputs in L6. The revised abstract reads as follows:

“Unified visual perception relies on the integration of bottom-up and top-down inputs in the primary visual cortex (V1), yet the organization of

top-down inputs in V1 remains unclear. Here, we used optogenetics-assisted circuit mapping to identify how multiple top-down inputs from higher-order cortical and thalamic areas engage excitatory and inhibitory neurons in V1. Top-down inputs partially overlap in superficial layers, bypass layer 4 (L4), and clearly segregate upon reaching deep layers. Specifically, inputs from the medial secondary visual cortex (V2M) and anterior cingulate cortex (ACA) preferentially activate L6 neurons, while inputs from the ventrolateral orbitofrontal cortex (ORBvl) and lateral posterior thalamic nucleus (LP) activate L5 neurons. We also found that V2M and ACA inputs converge on L6 Pyrs, whereas ORBvl and LP inputs are processed in parallel by two types of L5 Pyrs: Pyr-ORBvl and Pyr-LP neurons, each characterized by specific electrophysiological properties and gene expression profiles. Retrograde mapping revealed that Pyr-ORBvl neurons preferentially innervate subcortical areas and Pyr-LP neurons innervate cortical areas, indicating parallel processing of ORBvl and LP inputs in Pyr-type-specific subnetworks. And we found that ORBvl and LP inputs drive mutual inhibition, mediated by local inhibitory neurons, between these two subnetworks in L5 of V1. Our study thus deepens understanding of the neuronal mechanisms involved in top-down modulation of visual processing and establishes that V2M and ACA inputs in L6 employ integrated processing that is distinct from the parallel processing of LP and ORBvl inputs in L5.”

1. We revised our introduction section to briefly introduce the parallel and integrated processing in visual information processing and to clarify our motivation in choosing to study V2M, ACA, ORBvl, and LP inputs (and the interactions among them). The revised text reads as follows (P. 3, Lines 2-31 and P. 4 Line 1):

“The visual system is hierarchically organized, with functionally related areas connecting to each other in specific laminar patterns (Coogan and Burkhalter, 1993; D'Souza et al., 2016; Felleman and Van Essen, 1991; Harris et al., 2019). Processing of various dimensions of the complex visual environment, such as color, depth, shape, and motion, occurs in parallel pathways starting from the retina and continues in specialized visual areas, followed by integration in higher-order areas to form a unified perception (Glickfeld and Olsen, 2017; Ibbotson and Meffin, 2020; Livingstone and Hubel, 1988; Nassi and Callaway, 2009; Seabrook et al., 2017; Shapley, 1990; Wassle, 2004). This involves both bottom-up inputs, which flow from the retina to higher-order areas, and top-down modulation, where signals from higher-order areas adapt visual processing to meet the requirement of the current task (Glickfeld and Olsen, 2017; Livingstone and Hubel, 1988; Nassi and Callaway, 2009; Seabrook et al., 2017; Shapley, 1990; Wassle, 2004).

V1 is the initial cortical area for visual information processing; it receives condensed and parallel bottom-up signals from the retinogeniculo-cortical pathways, extracts relevant information, and further elaborates and integrates this information with top-down inputs from higher-order cortical and thalamic areas, contributing to the formation of a unified perceptual experience (Froudarakis et al., 2019; Glickfeld and Olsen, 2017; Ibbotson and Meffin, 2020; Niell and Scanziani, 2021; Seabrook et al., 2017). In more detail, thalamocortical (TC) bottom-up inputs from the dorsal lateral geniculate thalamic nucleus (dLGN) preferentially target the medial layer (L4), while corticocortical (CC) and TC top-down inputs target superficial and deep layers in V1 (Harris et al., 2019; Hunnicutt et al., 2014; Oh et al., 2014; Zingg et al., 2014).

Previous studies have identified multiple higher-order cortical and thalamic areas that provide top-down inputs to V1, such as the secondary visual cortex (V2), posterior parietal cortex (PTLp), retrosplenial cortex (RSP), anterior cingulate cortex (ACA), ventrolateral orbitofrontal cortex (ORBvl), and lateral dorsal (LD) and lateral posterior thalamic nuclei (LP) (Harris et al., 2019; Ma et al., 2021; Oh et al., 2014; Zhang et al., 2016). These top-down inputs convey a rich array of information, including attention (Debes and Dragoi, 2023; Hu et al., 2019; Zhang et al., 2014), expectation (Fiser et al., 2016; Leinweber et al., 2017), perceptual tasks (Liu et al., 2020; Norman et al., 2021), and motor commands (Huda et al., 2020; Kim et al., 2021).

The signals from these top-down inputs are differentially processed in V1 (D'Souza et al., 2016; Gilbert and Li, 2013; Liu et al., 2020; Moore and Zirnsak, 2017; Young et al., 2021; Zhang et al., 2014). For instance, V2 inputs are known to enhance the accuracy of visual information decoding in V1 neurons without altering the average response (Javadzadeh and Hofer, 2022). ACA and LP inputs increase V1 neuron response to task-relevant visual information (Hu et al., 2019; Zhang et al., 2014), whereas ORBvl inputs suppress V1 neuron responses to filter out irrelevant visual information (Liu et al., 2020). The diverse effects of these top-down modulations on visual processing imply sophisticated mechanisms within local circuits of V1, presumably controlled by distinct sets of excitatory and inhibitory neurons across different layers. However, knowledge about how these top-down inputs differentially engage excitatory and inhibitory neurons in V1 is quite limited, particularly with regard to their innervation patterns of neurons in deep layers.”

We have also revised the last paragraph in the introduction section. The revised text reads as follows (P. 4, Lines 12-32 and P. 5, Line 1):

“In this study, we profiled the layer- and cell-type-specific innervation patterns of multiple CC and TC top-down inputs in V1, including three CC inputs (V2M, ACA, and ORBvl) and one TC input

(LP). We found distinct layer- and cell-type-specific innervation profiles for each top-down input, with profiles partially overlapping in superficial layers, bypassing L4, and clearly segregating in deep layers. Specifically, V2M and ACA inputs preferentially activate L6 Pyrs, while ORBvl and LP inputs activate L5 Pyrs. We also characterized the layer-specificity of top-down inputs on inhibitory neurons, revealing that L1-INs are strongly activated in L1 and that VIP+ neurons are strongly activated in both superficial and deep layers, while PV+ and SST+ neurons are specifically activated in the deep layers. These results provide a valuable resource for the layer- and cell-type-specific organization of top-down inputs in V1. We subsequently investigated how these inputs interact within their strongest receptive layers, L5 and L6. Using independent optogenetic activation on the same brain slice, we found that V2M and ACA inputs converge on L6 Pyrs, whereas ORBvl and LP inputs selectively activate two distinct types of L5 Pyrs: $\text{Pyr}_{\leftarrow\text{ORBvl}}$ and $\text{Pyr}_{\leftarrow\text{LP}}$ neurons, each characterized by specific electrophysiological properties and gene expression profiles. Retrograde tracing revealed that $\text{Pyr}_{\leftarrow\text{ORBvl}}$ neurons preferentially innervate subcortical areas and $\text{Pyr}_{\leftarrow\text{LP}}$ neurons innervate cortical areas, indicating parallel processing of the ORBvl and LP inputs in Pyr-type-specific subnetworks. We also found that ORBvl and LP inputs drive mutual inhibition, mediated by local inhibitory neurons, between these two subnetworks in L5 of V1. These findings deepen our understanding of neuronal mechanisms of top-down modulation of visual processing by revealing interactions of modulation signals within V1 local circuits.”

2. In the results section, we have added a main figure, revised a subheading, and added two paragraphs to introduce the results of the new experiment, in which independent optogenetic activation of V2M and ACA inputs revealed their convergence in L6 Pyrs. Additionally, we have revised the conclusion of this subsection accordingly. The revised subsection reads as follows (P. 7, Lines 14-31 and P. 8, Lines 1-17):

“Integrated processing of V2M and ACA inputs in L6 Pyrs versus parallel processing of LP and ORBvl inputs in two distinct L5 Pyr populations

The examined CC and TC top-down inputs are strongest in the deep layers, with L6 being most responsive to V2M and ACA inputs, while L5 is most responsive to LP and ORBvl inputs. Previous studies have reported that deep-layer Pyrs are a heterogeneous population, with functionally distinct subnetworks (Harris and Shepherd, 2015; Matho et al., 2021; Tasic et al., 2018). To investigate the interactions among these top-down inputs in their most responsive layers, we employed independent optogenetic activation (Rindner et al., 2022). Specifically, for V2M and ACA inputs, we alternatively activated these inputs using Chr2 and ChrimsonR in the same brain slice and measured their input strengths in each recorded L6 Pyr. In each animal, we injected an AAV

expressing ChR2-EYFP in the V2M (AAV-CaMKII α -ChR2-EYFP) and also injected an AAV expressing ChrimsonR-tdTomato (AAV-hSyn-ChrimsonR-tdTomato) in the ACA (Fig. 3a). Expression of ChR2-EYFP in the V2M and ChrimsonR-tdTomato in the ACA resulted in the expected bright green and red axonal fluorescence in L6 of V1 (Fig. 3b-d).

Activation of V2M inputs (10 pulses @ 10Hz, 488 nm) and ACA inputs (10 pulses @ 10Hz, 647 nm) elicited robust postsynaptic currents (EPSCs) in L6 Pyrs (Fig. 3e-g). The majority of recorded L6 Pyrs (57%) received inputs from both V2M and ACA, with 40% receiving only V2M inputs and with 3% not receiving inputs from either (Fig. 3h). These results show that CC top-down inputs from V2M and ACA converge on the same group of L6 Pyrs, indicating integrated processing of their top-down modulation signals in L6 of V1.

We next examined the innervation patterns of LP and ORBvl inputs in L5 Pyrs. For each animal, we injected an AAV expressing ChrimsonR-tdTomato in the LP (AAV-hSyn-ChrimsonR-tdTomato) and also injected an AAV expressing ChR2-EYFP in the ORBvl (AAV-CaMKII α -ChR2-EYFP) (Fig. 4a). Expression of ChrimsonR-tdTomato in the LP and ChR2-EYFP in the ORBvl resulted in bright red and green axonal fluorescence in L5 of V1, with LP inputs primarily in upper L5 (red) and ORBvl inputs in lower L5 (green) (Fig. 4b-d). Activation of LP inputs (10 pulses @ 10Hz, 647 nm) elicited robust postsynaptic currents (EPSCs) in about half of recorded L5 Pyrs (Pyr \leftarrow LP neurons); these neurons did not respond to activation of ORBvl inputs (10 pulses @ 10Hz, 488 nm) (Fig. 4e-g). Intriguingly, when we activated ORBvl inputs, the remaining half of L5 Pyrs displayed robust EPSCs (Pyr \leftarrow ORBvl neurons) (Fig. 4e-h). These results indicate that TC and CC top-down inputs from LP and ORBvl employ parallel processing strategies by selectively activating distinct L5 Pyr populations in V1. Thus, our results reveal examined CC and TC top-down inputs employ distinct processing strategies in their interactions, with integrated processing of V2M and ACA inputs in L6 versus parallel processing of LP and ORBvl inputs in L5.”

“Fig. 3 | V2M and ACA inputs converge on L6 Pyrs. a, Schematic of the viral strategy for independent optogenetic activation of V2M and ACA inputs. **b**, Fluorescent images showing the distribution of V2M (green) and ACA axons (red) in V1. Scale bars, 100 μ m. **c**, Normalized axon intensity of V2M (green) and ACA (red) axons as a function of L6 depth. 0% and 100% represent the upper and lower boundaries of L6. $n = 10$ slices. **d**, Summary of L6 depth for peak V2M (green) axon intensity and for peak ACA (red) axon intensity. $P = 0.12$, Wilcoxon sign-rank test. **e**, Schematic of the slice experiment examining independent optogenetic activation of V2M and ACA inputs. **f**, EPSCs evoked by independent activation of V2M and ACA inputs in L6 Pyrs. Blue dots, 488-nm light stimulation, 10 Hz, 1 ms, 3.5 mW. Red dots, 647-nm light stimulation, 10 Hz, 1 ms, 3.5 mW. Gray traces are individual trials, while purple and blue traces are averages of 5 trials. Scale bars, 100 ms, 50 pA. **g**, Normalized EPSC charge evoked by activation of V2M and ACA inputs in $\text{Pyr}_{\leftarrow\text{both}}$ (top) and $\text{Pyr}_{\leftarrow\text{V2M}}$ neurons (bottom). The circles indicate the means. $n = 5$ mice, 7 slices. $\text{Pyr}_{\leftarrow\text{both}}$, $n = 37$ neurons, $P = 0.57$; $\text{Pyr}_{\leftarrow\text{V2M}}$, $n = 26$ neurons, $P = 3 \times 10^{-39}$. $***P < 0.001$; Paired t -test. **h**, Percentage (top) and distribution (bottom) of L6 Pyrs activated by V2M and ACA inputs. Purple, L6 Pyrs receiving both inputs; Blue, L6 Pyrs receiving only V2M inputs; gray, L6 Pyrs without any response to V2M or ACA inputs. Data are presented as the mean \pm SEM. Figure3, Panel a adapted from Petrucco, L. (2020). Mouse head schema. Zenodo.

<https://doi.org/10.5281/zenodo.3925903> under a CC BY license:
[https://creativecommons.org/licenses/by/4.0/.](https://creativecommons.org/licenses/by/4.0/)”

3. In the discussion section, we have revised the first paragraph. The revised text reads as follows (P. 14, Lines 18-29):

“In this study, we characterized the layer- and cell-type-specific organization of multiple CC and TC top-down inputs in V1, as well as their interactions in deep layers. Each top-down input to V1 engaged both excitatory and inhibitory neurons: CC and TC top-down inputs partially overlapped in superficial layers (L1 and L2/3), bypassed the medial layer (L4), and clearly segregated in deep layers (L5 and L6). Our datasets provide a valuable resource for understanding the neuronal mechanisms of top-down modulation of visual information processing. Further, we found that CC top-down inputs from the V2M and ACA undergo integrated processing in L6 Pyrs, whereas TC top-down inputs from the ORBv1 and LP undergo parallel processing within two L5 Pyr-type-specific subnetworks, separately mediating subcortical and intracortical V1 output channels. Notably, ORBv1 and LP inputs drive mutual inhibition between these two subnetworks in L5 of V1, potentially enabling dynamic gating of subnetwork functions in response to specific top-down modulation signals.”

We have also revised the paragraph discussing the different Pyr types engaged in the processing of top-down inputs. The revised text reads as follows (P. 15, Lines 20-30):

“Cortical Pyrs have been divided based on axonal projections into three main types: IT neurons preferentially innervating the cortex and striatum; PT neurons innervating subcortical areas; and corticothalamic neurons, which only innervate the thalamus (Harris and Shepherd, 2015). These types overlap in deep layers, with IT and PT in L5, and IT and corticothalamic in L6. Our study has revealed the specificity of various CC and TC top-down inputs engaging distinct Pyr types in deep layers of V1. V2M and ACA inputs converge on L6 Pyrs, suggesting integrated processing of these inputs that likely involves both IT and corticothalamic neurons. And we found that ORBv1 and LP inputs selectively activate PT (Pyr-ORBv1) and IT (Pyr-LP) neurons, each with distinct AP modes (BS spiking vs. RS spiking), indicating parallel processing of the information they convey in the PT and IT subnetworks.”

2) Furthermore, the methods employed in the paper might not represent the best available tools for the job. Notably, other researchers have utilized sCRACM (as seen in the Petreanu/Svoboda paper), which offers a more detailed perspective of input locations. However, it's understandable that these experiments might not be feasible due to potential resource constraints.

RESPONSE:

We appreciate the reviewer's idea of employing the subcellular ChR2-assisted circuit mapping (sCRACM) technique, as developed in Svoboda's lab, and we understand its capacity to finely delineate innervation patterns. While we recognize the advantages of this method in detailing the innervation of apical and basal dendrites, including the subregions of the apical dendrites, we cannot currently implement sCRACM, owing to our lack of access to a two-photon microscope. Acknowledging this limitation, we have included a new paragraph in the discussion section of our revised manuscript, in which we elaborate on the potential insights that could be gained from employing higher-resolution stimulation techniques like sCRACM in future studies of circuitry mechanisms underlying top-down modulation in V1. The revised text is as follows (P. 15, Lines 31-32 and P. 16, Lines 1-12):

“The dendrites of Pyrs typically extend and branch across multiple layers (Fishell and Kepecs, 2020; Fishell and Rudy, 2011; Harris and Shepherd, 2015; Yuste et al., 2020), and the spatial distribution of specific inputs within dendritic arborization serve distinct functions: spatially clustered co-active synapses are more efficacious in driving postsynaptic neurons than spatially distributed synapses (Losonczy and Magee, 2006; Polsky et al., 2004). Previous studies using subcellular ChR2-assisted circuit mapping (sCRACM) revealed the subcellular organization of local and long-range excitatory inputs to Pyrs in sensorimotor cortices, findings helpful for predicting the interactions among these inputs within the dendrites of a single neuron (Mao et al., 2011; Petreanu et al., 2009). A recent study employing sCRACM has shown that top-down inputs from V2 to V1 selectively modulate apical dendrites, but not basal dendrites of L5/L6 looped IT neurons that project back to V2 (Young et al., 2021). Future studies using sCRACM to investigate the subcellular organization of other top-down inputs in V1—specifically examining whether their synapses are clustered or segregated—are likely to provide valuable insights. Such studies could significantly enhance our understanding of dendritic integration during the interactions among these multiple top-down inputs in V1 neurons.”

Minor:

1. The authors should include information about the specific locations of the neurons they recorded, mapping them to the corresponding layers and indicating their relative

positions in relation to layer boundaries. Notably, Layer 5 can often be further subdivided into multiple sublayers, with different neuron types residing in distinct sublayers. If ORBvl-receiving neurons are recorded from a different layer compared to LP-receiving neurons, it's crucial to acknowledge that the study might effectively be examining neurons from two different sublayers. The presence of distinct properties in such cases would not be surprising.

RESPONSE:

We thank the reviewer for this guidance, and have made revisions accordingly. Please note that we have added additional results showing the specific locations of the neurons we recorded in both the main figures and supplementary figures, including Figs. 3 and 4, as well as in Extended Data Figs. 1,2,6,9, and 11.

Regarding the distribution of the recorded Pyr \leftarrow LP (LP-receiving) and Pyr \leftarrow ORBvl (ORBvl-receiving) neurons as shown in Fig. 4h, they extensively overlap in L5 of V1, rather than being segregated into different sublayers.

“Fig. 3 | V2M and ACA inputs converge on L6 Pyrs. **a**, Schematic of the viral strategy for independent optogenetic activation of V2M and ACA inputs. **b**, Fluorescent images showing the distribution of V2M (green) and ACA axons (red) in V1. Scale bars, 100 μ m. **c**, Normalized

axon intensity of V2M (green) and ACA (red) axons as a function of L6 depth. 0% and 100% represent the upper and lower boundaries of L6. $n = 10$ slices. **d**, Summary of L6 depth for peak V2M (green) axon intensity and for peak ACA (red) axon intensity. $P = 0.12$, Wilcoxon sign-rank test. **e**, Schematic of the slice experiment examining independent optogenetic activation of V2M and ACA inputs. **f**, EPSCs evoked by independent activation of V2M and ACA inputs in L6 Pyrs. Blue dots, 488-nm light stimulation, 10 Hz, 1 ms, 3.5 mW. Red dots, 647-nm light stimulation, 10 Hz, 1 ms, 3.5 mW. Gray traces are individual trials, while purple and blue traces are averages of 5 trials. Scale bars, 100 ms, 50 pA. **g**, Normalized EPSC charge evoked by activation of V2M and ACA inputs in Pyr $_{\leftarrow}$ both (top) and Pyr $_{\leftarrow}$ V2M neurons (bottom). The circles indicate the means. $n = 5$ mice, 7 slices. Pyr $_{\leftarrow}$ both, $n = 37$ neurons, $P = 0.57$; Pyr $_{\leftarrow}$ V2M, $n = 26$ neurons, $P = 3 \times 10^{-39}$. *** $P < 0.001$; Paired t -test. **h**, Percentage (top) and distribution (bottom) of L6 Pyrs activated by V2M and ACA inputs. Purple, L6 Pyrs receiving both inputs; Blue, L6 Pyrs receiving only V2M inputs; gray, L6 Pyrs without any response to V2M or ACA inputs. Data are presented as the mean \pm SEM. Figure 3, Panel a adapted from Petrucco, L. (2020). Mouse head schema. Zenodo. <https://doi.org/10.5281/zenodo.3925903> under a CC BY license: <https://creativecommons.org/licenses/by/4.0/>.”

“Fig. 4 | LP and ORBvl inputs target distinct subpopulations of L5 Pyrs. **a**, Schematic of the viral strategy for independent optogenetic activation of LP and ORBvl inputs. **b**, Fluorescent images showing the distribution of LP (red) and ORBvl axons (green) in V1. Scale bars, 100 μm . **c**, Normalized axon intensity of LP (red) and ORBvl (green) axons as a function of L5 depth. 0% and 100% represent the upper and lower boundaries of L5. $n = 12$ slices. **d**, Summary of L5 depth for peak LP (red) axon intensity and for peak ORBvl (green) axon intensity. $P = 5 \times 10^{-4}$. $***P < 0.001$; Wilcoxon sign-rank test. **e**, Schematic of the slice experiment examining independent optogenetic activation of LP and ORBvl inputs. **f**, EPSCs evoked by independent activation of LP and ORBvl inputs in L5 Pyrs. Red dots, 647-nm light stimulation, 10 Hz, 1 ms, 3.5 mW. Blue dots, 488-nm light stimulation, 10 Hz, 1 ms, 3.5 mW. Gray traces are individual trials, while red and green traces are averages of 5 trials. Scale bars, 100 ms, 100 pA. **g**, Normalized EPSC charge evoked by activation of LP and ORBvl inputs in Pyr \leftarrow LP (top) and Pyr \leftarrow ORBvl (bottom). The circles indicate the means. $n = 12$ mice, 14 slices. Pyr \leftarrow LP, $n = 28$ neurons, $P = 2 \times 10^{-44}$; Pyr \leftarrow ORBvl, $n = 35$ neurons, $P = 9 \times 10^{-33}$. $***P < 0.001$, Paired t -test. **h**, Percentage (top) and distribution (bottom) of L5 Pyrs activated by LP and ORBvl inputs (Pyr \leftarrow LP and Pyr \leftarrow ORBvl neurons). Red, Pyr \leftarrow LP neurons; green, Pyr \leftarrow ORBvl neurons; gray, L5 Pyrs without any response to LP or ORBvl inputs. Data are presented as the mean \pm SEM. Figure4, Panel a adapted from Petrucco, L. (2020). Mouse head schema. Zenodo. <https://doi.org/10.5281/zenodo.3925903> under a CC BY license: <https://creativecommons.org/licenses/by/4.0/>.”

“**Extended Data Fig. 1 | Proportion of V1 Pyrs activated by each CC and TC input across different layers and the distribution of recorded V1 Pyrs.** **a**, Percentage of Pyrs in V1 receiving V2M inputs exceeding 3 mV across different layers. **b**, Similar to a, but for ACA inputs. **c**, Similar to a, but for ORBvl inputs. **d**, Similar to (a), but for LP inputs. **e**, Distribution of recorded Pyrs in experiments involving optogenetic activation of V2M inputs. **f-h**, Similar to e, but for the experiments involving optogenetic activation of ACA, ORBvl, and LP inputs.”

“Extended Data Fig. 2 | Proportion of V1 inhibitory neurons activated by each CC and TC input across different layers and the distribution of recorded V1 inhibitory neurons. a, Fluorescent images showing the distribution of VIP+, PV+ and SST+ neurons across different layers in V1. Red, VIP+, PV+, and SST+ neurons expressing tdTomato. Blue, DAPI. Scale bar, 100 μ m. **b,** The cell density of VIP+ (red), PV+ (blue) and SST+ (yellow) neurons across layers in V1. Colored shading, \pm SEM. VIP+ neurons, n = 5 mice; PV+ neurons, n = 6; SST+ neurons, n = 4. **c,** Percentage of four types of inhibitory neurons (L1-INs, VIP+, PV+,

and SST+ neurons) in V1 receiving V2M inputs exceeding 3 mV in each layer. **d**, Similar to c, but for ACA inputs. **e**, Similar to c, but for ORBvl inputs. **f**, Similar to c, but for LP inputs. The number of neurons in each group is indicated by the numbers displayed in the figure. **g**, Distribution of recorded inhibitory neurons in experiments involving optogenetic activation of V2M inputs. **h-j**, Similar to g, but for experiments involving optogenetic activation of ACA, ORBvl, and LP inputs.

“**Extended Data Fig. 6 | Electrophysiological properties of Pyr-LP and Pyr-ORBvl neurons.** **a**, The AP threshold of Pyr-LP and Pyr-ORBvl neurons. Pyr-LP, n = 47 neurons; Pyr-ORBvl, n = 44 neurons; P = 0.004, Wilcoxon rank-sum test. **b**, Similar to a, but for the instantaneous firing rate of the first two APs elicited by positive current injection. P = 3×10^{-13} . **c**, Similar to a, but for the resting membrane potential. P = 2×10^{-5} . **d**, Similar to a, but for the input resistance. P = 2×10^{-7} . Data are presented as the mean \pm SEM. **P < 0.01, ***P < 0.001. **e**, Distribution of recorded Pyr-LP and Pyr-ORBvl neurons.”

“**Extended Data Fig. 9 | Distribution of recorded projecting-Pyrs in L5 of V1 shown in Figure 7. a**, Distribution of recorded cortex-projecting Pyrs. **b**, Similar to a, but for DMS-projecting Pyrs. **c**, Similar to a, but for other subcortical area-projecting Pyrs.”

“**Extended Data Fig. 11 | Distribution of recorded Pyr←LP and Pyr←ORBv1 neurons in Figure 8. a**, Distribution of recorded Pyr←LP and Pyr←ORBv1 neurons in Fig.8b. **b**, Similar to a, but for recorded Pyr←LP and Pyr←ORBv1 neurons in Fig.8g.”

2. In relation to the SVM model training, the authors mention: "The performance of the SVM Classifier (I_h) was evaluated using 10-fold cross-validation. The detected average accuracy of 88.2% (over 20 cross-validations) significantly surpassed the chance level of 50% ($P = 2 \times 10^{-26}$, paired t -test)."

a. It's important to clarify how the authors arrived at "20 cross-validations" when using the 10-fold cross-validation method. In 10-fold cross-validation, the data is typically divided into 10 sets for training (9 sets) and testing (1 set). Therefore, they should report 10 validation results, not 20.

RESPONSE:

Based on the reviewer's guidance, we have now clarified the issue regarding "20 cross-validations" in the revised manuscript. In each round of the 10-fold cross-validation process, the data was divided into 10 sets for training (9 sets) and testing (1 set), and this process was repeated 10 times, ensuring that each set was used as the test set exactly once. We repeated the entire 10-fold cross-validation process 20 times, each time randomly dividing the data into 10 sets. The revised text is as follows (P. 9, Lines 22-24):

"The performance of the SVM Classifier(I_h) was evaluated using 10-fold cross-validation, repeating the entire 10-fold process 20 times with random divisions of the data."

b. Explain how the "paired t -test" was conducted on the data when the data is not intrinsically paired. If the data involved shuffling and performing k -fold cross-validation on the shuffled data, it may not qualify as a "paired" situation in the statistical sense. Paired tests generally pertain to the same sample subjected to two different conditions, with interest in the difference between them. This concept does not align with the conditions presented in this study.

RESPONSE:

We thank the reviewer for pointing out this mistake; we have now replaced paired t -test with one-sample t -test. The new statistical analysis concurs with our initial conclusions. The revised text is as follows (P. 9, Lines 24-25):

"The detected average accuracy of 88.2% significantly surpassed the chance level of 50% ($P = 4 \times 10^{-28}$, one-sample t -test)."

c. Clarify which data were used to train the final SVM model that was later applied to another dataset. State whether all 82+65 samples were used in the training process.

RESPONSE:

Following the reviewer's guidance, we have now clarified this issue in the results section of the revised manuscript. The revised text is as follows (P. 9, Lines 27-29):

“...we applied the SVM Classifier(I_h) trained with the complete data set (82 Pyr \leftarrow LP neurons and 65 Pyr \leftarrow ORBv1 neurons) to another dataset (the data from Figure 4 for alternative activation of LP and ORBv1 inputs in the same brain slice).”

We are very grateful to the reviewer for the insightful revision comments, which have been very helpful as we revised our manuscript.

References

- Coogan, T.A., and Burkhalter, A. (1993). Hierarchical organization of areas in rat visual cortex. *J. Neurosci.* *13*, 3749-3772.
- D'Souza, R.D., Meier, A.M., Bista, P., Wang, Q., and Burkhalter, A. (2016). Recruitment of inhibition and excitation across mouse visual cortex depends on the hierarchy of interconnecting areas. *Elife* *5*, e19332.
- Debes, S.R., and Dragoi, V. (2023). Suppressing feedback signals to visual cortex abolishes attentional modulation. *Science* *379*, 468-473.
- Felleman, D.J., and Van Essen, D.C. (1991). Distributed hierarchical processing in the primate cerebral cortex. *Cereb. Cortex* *1*, 1-47.
- Fiser, A., Mahringer, D., Oyibo, H.K., Petersen, A.V., Leinweber, M., and Keller, G.B. (2016). Experience-dependent spatial expectations in mouse visual cortex. *Nat. Neurosci.* *19*, 1658-1664.
- Fishell, G., and Kepecs, A. (2020). Interneuron Types as Attractors and Controllers. *Annu. Rev. Neurosci.* *43*, 1-30.
- Fishell, G., and Rudy, B. (2011). Mechanisms of inhibition within the telencephalon: "where the wild things are". *Annu. Rev. Neurosci.* *34*, 535-567.
- Froudarakis, E., Fahey, P.G., Reimer, J., Smirnakis, S.M., Tehovnik, E.J., and Tolias, A.S. (2019). The Visual Cortex in Context. *Annu. Rev. Vis. Sci.* *5*, 317-339.
- Gilbert, C.D., and Li, W. (2013). Top-down influences on visual processing. *Nat. Rev. Neurosci.* *14*, 350-363.
- Glickfeld, L.L., and Olsen, S.R. (2017). Higher-Order Areas of the Mouse Visual Cortex. *Annu. Rev. Vis. Sci.* *3*, 251-273.
- Harris, J.A., Mihalas, S., Hirokawa, K.E., Whitesell, J.D., Choi, H., Bernard, A., Bohn, P., Caldejon, S., Casal, L., Cho, A., *et al.* (2019). Hierarchical organization of cortical and thalamic connectivity. *Nature* *575*, 195-202.
- Harris, K.D., and Shepherd, G.M. (2015). The neocortical circuit: themes and variations. *Nat. Neurosci.* *18*, 170-181.
- Hu, F., Kamigaki, T., Zhang, Z., Zhang, S., Dan, U., and Dan, Y. (2019). Prefrontal Corticotectal Neurons Enhance Visual Processing through the Superior Colliculus and Pulvinar Thalamus. *Neuron* *104*, 1141-1152.
- Huda, R., Sipe, G.O., Breton-Provencher, V., Cruz, K.G., Pho, G.N., Adam, E., Gunter, L.M., Sullins, A., Wickersham, I.R., and Sur, M. (2020). Distinct prefrontal top-down circuits differentially modulate sensorimotor behavior. *Nat. Commun.* *11*, 6007.
- Hunnicutt, B.J., Long, B.R., Kusefoglu, D., Gertz, K.J., Zhong, H., and Mao, T. (2014). A comprehensive thalamocortical projection map at the mesoscopic level. *Nat. Neurosci.* *17*, 1276-1285.
- Ibbotson, M.R., and Meffin, H. (2020). Visual Information Processing. In *The Senses: A Comprehensive Reference*, pp. 36-53.
- Javadzadeh, M., and Hofer, S.B. (2022). Dynamic causal communication channels between neocortical areas. *Neuron* *110*, 2470-2483.
- Kim, J.H., Ma, D.H., Jung, E., Choi, I., and Lee, S.H. (2021). Gated feedforward inhibition in the frontal cortex releases goal-directed action. *Nat. Neurosci.* *24*, 1452-1464.
- Leinweber, M., Ward, D.R., Sobczak, J.M., Attinger, A., and Keller, G.B. (2017). A Sensorimotor

Circuit in Mouse Cortex for Visual Flow Predictions. *Neuron* 95, 1420-1432.

Liu, D., Deng, J., Zhang, Z., Zhang, Z.Y., Sun, Y.G., Yang, T., and Yao, H. (2020). Orbitofrontal control of visual cortex gain promotes visual associative learning. *Nat. Commun.* 11, 2784.

Livingstone, M., and Hubel, D. (1988). Segregation of Form, Color, Movement, and Depth: Anatomy, Physiology, and Perception. *Science* 240, 740-749.

Losonczy, A., and Magee, J.C. (2006). Integrative properties of radial oblique dendrites in hippocampal CA1 pyramidal neurons. *Neuron* 50, 291-307.

Ma, G., Liu, Y., Wang, L., Xiao, Z., Song, K., Wang, Y., Peng, W., Liu, X., Wang, Z., Jin, S., *et al.* (2021). Hierarchy in sensory processing reflected by innervation balance on cortical interneurons. *Sci. Adv.* 7, 20 eabf5676.

Mao, T., Kusefoglu, D., Hooks, B.M., Huber, D., Petreanu, L., and Svoboda, K. (2011). Long-range neuronal circuits underlying the interaction between sensory and motor cortex. *Neuron* 72, 111-123.

Matho, K.S., Huilgol, D., Galbavy, W., He, M., Kim, G., An, X., Lu, J., Wu, P., Di Bella, D.J., Shetty, A.S., *et al.* (2021). Genetic dissection of the glutamatergic neuron system in cerebral cortex. *Nature* 598, 182-187.

Moore, T., and Zirnsak, M. (2017). Neural Mechanisms of Selective Visual Attention. *Annu. Rev. Psychol.* 68, 47-72.

Nassi, J.J., and Callaway, E.M. (2009). Parallel processing strategies of the primate visual system. *Nat. Rev. Neurosci.* 10, 360-372.

Niell, C.M., and Scanziani, M. (2021). How Cortical Circuits Implement Cortical Computations: Mouse Visual Cortex as a Model. *Annu. Rev. Neurosci.* 44, 517-546.

Norman, K.J., Riceberg, J.S., Koike, H., Bateh, J., McCraney, S.E., Caro, K., Kato, D., Liang, A., Yamamuro, K., Flanigan, M.E., *et al.* (2021). Post-error recruitment of frontal sensory cortical projections promotes attention in mice. *Neuron* 109, 1202-1213.

Oh, S.W., Harris, J.A., Ng, L., Winslow, B., Cain, N., Mihalas, S., Wang, Q., Lau, C., Kuan, L., Henry, A.M., *et al.* (2014). A mesoscale connectome of the mouse brain. *Nature* 508, 207-214.

Petreanu, L., Mao, T., Sternson, S.M., and Svoboda, K. (2009). The subcellular organization of neocortical excitatory connections. *Nature* 457, 1142-1145.

Polsky, A., Mel, B.W., and Schiller, J. (2004). Computational subunits in thin dendrites of pyramidal cells. *Nat. Neurosci.* 7, 621-627.

Rindner, D.J., Proddatur, A., and Lur, G. (2022). Cell-type-specific integration of feedforward and feedback synaptic inputs in the posterior parietal cortex. *Neuron* 110, 3760-3773.

Seabrook, T.A., Burbridge, T.J., Crair, M.C., and Huberman, A.D. (2017). Architecture, Function, and Assembly of the Mouse Visual System. *Annu. Rev. Neurosci.* 40, 499-538.

Shapley, R. (1990). Visual sensitivity and parallel retinocortical channels. *Annu. Rev. Psychol.* 41, 635-658.

Tasic, B., Yao, Z., Graybiel, L.T., Smith, K.A., Nguyen, T.N., Bertagnolli, D., Goldy, J., Garren, E., Economo, M.N., Viswanathan, S., *et al.* (2018). Shared and distinct transcriptomic cell types across neocortical areas. *Nature* 563, 72-78.

Wassle, H. (2004). Parallel processing in the mammalian retina. *Nat. Rev. Neurosci.* 5, 747-757.

Yao, S., Wang, Q., Hirokawa, K.E., Ouellette, B., Ahmed, R., Bomben, J., Brouner, K., Casal, L., Caldejon, S., Cho, A., *et al.* (2023). A whole-brain monosynaptic input connectome to neuron classes in mouse visual cortex. *Nat. Neurosci.* 26, 350-364.

Yao, Z., van Velthoven, C.T.J., Nguyen, T.N., Goldy, J., Sedeno-Cortes, A.E., Baftizadeh, F., Bertagnolli, D., Casper, T., Chiang, M., Crichton, K., *et al.* (2021). A taxonomy of transcriptomic cell types across the isocortex and hippocampal formation. *Cell* *184*, 3222-3241.

Young, H., Belbut, B., Baeta, M., and Petreanu, L. (2021). Laminar-specific cortico-cortical loops in mouse visual cortex. *Elife* *10*, e59551.

Yuste, R., Hawrylycz, M., Aalling, N., Aguilar-Valles, A., Arendt, D., Arnedillo, R.A., Ascoli, G.A., Bielza, C., Bokharaie, V., Bergmann, T.B., *et al.* (2020). A community-based transcriptomics classification and nomenclature of neocortical cell types. *Nat. Neurosci.* *12*, 1456-1468.

Zhang, S., Xu, M., Chang, W.C., Ma, C., Hoang Do, J.P., Jeong, D., Lei, T., Fan, J.L., and Dan, Y. (2016). Organization of long-range inputs and outputs of frontal cortex for top-down control. *Nat. Neurosci.* *19*, 1733-1742.

Zhang, S., Xu, M., Kamigaki, T., Hoang Do, J.P., Chang, W.C., Jenvay, S., Miyamichi, K., Luo, L., and Dan, Y. (2014). Selective attention. Long-range and local circuits for top-down modulation of visual cortex processing. *Science* *345*, 660-665.

Zingg, B., Hintiryan, H., Gou, L., Song, M.Y., Bay, M., Bienkowski, M.S., Foster, N.N., Yamashita, S., Bowman, I., Toga, A.W., *et al.* (2014). Neural networks of the mouse neocortex. *Cell* *156*, 1096-1111.

Reviewer #2 (Remarks to the Author):

Using slice electrophysiology and optogenetics, Liu et al. initially examined the connectivity pattern of top-down inputs from V2M, ACA, ORBvl, and LP to pyramidal and inhibitory (VIP, PV, and SST) neurons in different layers in V1. The authors then further examined the ORB and LP inputs since the connectivity data suggested a potential cell class effect. The authors unveiled 1) V2M and ACA mainly innervated neurons in L6, 2) ORBvl and LP each innervated L5 pyramidal neurons with 50% chance, 3) ORB weakly innervates inhibitory neurons, 4) LP innervates VIP neurons in all layers (but strongest in L6) and PV neurons in L5, 5) L5 pyramidal neurons defined by their inputs (i.e., Pyr<-LP and Pyr<-ORB) can be distinguished by their electrophysiological and genetic properties, and projection to cortical and subcortical sites, respectively, and 6) Pyr<-LP and Pyr<-ORB neurons are differentially inhibited by their upstream inputs due to recruitment of local PV neurons. Based on these, the authors claimed different top-down inputs engage different V1 subnetworks, each having an effect on different downstream targets (cortical or subcortical).

The major strength of the paper is a rigorous and laborious quantitative analysis of the specific connectivity formed between multiple top-down afferents (in particular, LP and ORB axons) with different L5 pyramidal and inhibitory cell types (Fig. 1 and 2). Moreover, the demonstration of the absence of LP and ORB input convergence at the level of single neurons (Fig. 3) and finding that each afferent exerts mostly a multisynaptic inhibitory effect on unconnected L5 pyramidal neurons (Fig. 7) is interesting.

1) The weakness of the paper is its novelty in several aspects. For example, V2M input to V1 has been studied with a similar technique but in slightly different contexts by Young et al. 2021 (not cited). Young et al. reported reasonable input from V2M to V1 L5 pyramidal neurons (their Fig. 4) in addition to several other connectivity details relevant to the current study, thus should be discussed.

RESPONSE:

Following the reviewer's guidance, we have now cited this paper in the introduction, results, and discussion sections in our revised manuscript. Acknowledging our current study's limitation in lacking subcellular resolution, we have included a new paragraph in the discussion section of our revised manuscript. In this addition, we elaborate on the potential insights that could be gained from employing higher-resolution stimulation techniques like sCRACM in future studies of circuitry mechanisms underlying top-down modulation in V1.

The revised introduction reads as follows (P. 3, Lines 17-31 and P. 4, Line 1):

“Previous studies have identified multiple higher-order cortical and thalamic areas that provide top-down inputs to V1, such as the

secondary visual cortex (V2), posterior parietal cortex (PTLp), retrosplenial cortex (RSP), anterior cingulate cortex (ACA), ventrolateral orbitofrontal cortex (ORBvl), and lateral dorsal (LD) and lateral posterior thalamic nuclei (LP) (Harris et al., 2019; Ma et al., 2021; Oh et al., 2014; Zhang et al., 2016). These top-down inputs convey a rich array of information, including attention (Debes and Dragoi, 2023; Hu et al., 2019; Zhang et al., 2014), expectation (Fiser et al., 2016; Leinweber et al., 2017), perceptual tasks (Liu et al., 2020; Norman et al., 2021), and motor commands (Huda et al., 2020; Kim et al., 2021).

The signals from these top-down inputs are differentially processed in V1 (D'Souza et al., 2016; Gilbert and Li, 2013; Liu et al., 2020; Moore and Zirnsak, 2017; Young et al., 2021; Zhang et al., 2014). For instance, V2 inputs are known to enhance the accuracy of visual information decoding in V1 neurons without altering the average response (Javadzadeh and Hofer, 2022). ACA and LP inputs increase V1 neuron response to task-relevant visual information (Hu et al., 2019; Zhang et al., 2014), whereas ORBvl inputs suppress V1 neuron responses to filter out irrelevant visual information (Liu et al., 2020). The diverse effects of these top-down modulations on visual processing imply sophisticated mechanisms within local circuits of V1, presumably controlled by distinct sets of excitatory and inhibitory neurons across different layers. However, knowledge about how these top-down inputs differentially engage excitatory and inhibitory neurons in V1 is quite limited, particularly with regard to their innervation patterns of neurons in deep layers.”

The revised results reads as follows (P. 5, Lines 19-25):

“We conducted whole-cell recordings of V1 Pyrs and recorded the monosynaptic excitatory postsynaptic potential (EPSP) elicited by optogenetic activation of V2M axons, while blocking local neuron spikes with TTX (Petreanu et al., 2009). Consistent with a previous study (Young et al., 2021), we found that V2M inputs activated Pyrs in L2/3 and L6 of V1 (defined as averaged EPSP amplitude > 3 mV), with the strongest input strengths in L6 (Fig. 1d,e, Extended Data Fig. 1, and Supplementary Table 1 and 2).”

The revised discussion reads as follows (P. 15, Lines 31-32 and P. 16, Lines 1-12):

“The dendrites of Pyrs typically extend and branch across multiple layers (Fishell and Kepecs, 2020; Fishell and Rudy, 2011; Harris and Shepherd, 2015; Yuste et al., 2020), and the spatial distribution of specific

inputs within dendritic arborization serve distinct functions: spatially clustered co-active synapses are more efficacious in driving postsynaptic neurons than spatially distributed synapses (Losonczy and Magee, 2006; Polsky et al., 2004). Previous studies using subcellular ChR2-assisted circuit mapping (sCRACM) revealed the subcellular organization of local and long-range excitatory inputs to Pyrs in sensorimotor cortices, helpful for predicting the interactions among these inputs within the dendrites of a single neuron (Mao et al., 2011; Petreanu et al., 2009). A recent study employing sCRACM has shown that top-down inputs from V2 to V1 selectively modulate apical dendrites, but not basal dendrites of L5/L6 looped IT neurons, which project back to V2 (Young et al., 2021). Future studies using sCRACM to investigate the subcellular organization of other top-down inputs in V1—specifically examining whether their synapses are clustered or segregated—are likely to provide valuable insights. Such studies could significantly enhance our understanding of dendritic integration during the interactions among these multiple top-down inputs in V1 neurons.”

Regarding the differences in V2M→V1 input strength in L5 Pyrs, as reported in Young *et al.*, 2021, compared to our current study, we specifically examined the strengths of V2M input in relation to the distance between the recorded neurons and the V2M border, as shown in Reviewer Fig. 1. Our findings show that the responsiveness of V1 L5 Pyrs to V2M inputs informatively reflects the spatial distribution of V2M axons: the V2M axons are denser near the V2M and V2L borders within V1. We found that the responses of V1 L5 Pyrs to V2M inputs are significantly stronger within 400 μm of the V2M or V2L border, compared to those located 450-900 μm away from the V2M border. Therefore, the differences in V2M→V1 input strength observed between Young *et al.*'s study and ours can likely be attributed to variation in the recording sites within V1.

Reviewer Fig. 1 | Strengths of V2M inputs in relation to the distance between the recorded V1 L5 Pyrs and the V2M border. a, Fluorescent image showing the distribution of V2M axons (green) in V1. Gray dots represent the recording sites of V1 Pyrs. Scale bar, 200 μm. **b,** EPSC amplitude in V1 L5 Pyrs evoked by activation of V2M inputs. The number of neurons in each position is indicated by the numbers displayed in the figure. n = 3 mice, 3slices. Data are presented as the mean ± SEM.

I would like to take this opportunity to elaborate a bit more about the conceptual innovation of our study expanding beyond the scope of the Young *et al.* study. For context, unified visual perception is known to rely on the integration of bottom-up and top-down inputs in V1, with top-down inputs providing behavior-related modulation on visual processing. Previous studies have identified multiple higher-order cortical and thalamic areas that provide top-down inputs to V1, and the signals from these top-down inputs are differentially processed in V1 (D'Souza et al., 2016; Gilbert and Li, 2013; Javadzadeh and Hofer, 2022; Liu et al., 2020; Moore and Zirnsak, 2017; Young et al., 2021; Zhang et al., 2014). However, knowledge about how these top-down inputs differentially engage excitatory and inhibitory neurons in V1 is quite limited, particularly with regard to their innervation patterns of neurons in deep layers. Both our study and Young et al., 2021 reported on the innervation patterns of V2M to V1 Pyrs across different layers. Notably, our study also characterized the innervation patterns of V2M to various types of V1 inhibitory neurons.

Beyond V2M inputs, we also examined three other top-down inputs from the ACA, ORBv1, and LP. Our detailed profiling of layer- and cell-type-specific innervation patterns, involving various CC and TC top-down inputs in V1, provides empirical evidence for the neuronal mechanisms through which top-down visual modulation signals undergo processing. Interestingly, in contrast to similar innervation properties in superficial layers, the examined CC and TC inputs clearly segregated in deep layers. It is known that deep-layer Pyrs are a heterogeneous population with functionally distinct subnetworks (Harris and Shepherd, 2015; Matho et al., 2021; Tasic et al., 2018; Yao et al., 2021). To further investigate the interactions among these top-down inputs in their most responsive layers, we employed independent optogenetic activation. Together with new experiments added in this revision, our results demonstrated that V2M and ACA inputs converge on L6 Pyrs, in sharp contrast to the parallel processing of LP and ORBv1 inputs in two distinct types of L5 Pyrs. These findings demonstrate distinct processing strategies, with integrated processing of V2M and ACA inputs in L6, as opposed to the parallel processing of LP and ORBv1 inputs in L5.

Given the notable specificity with which L5 Pyrs process LP and ORBv1 inputs, we narrowed our focus to exploring the properties of these neurons: $\text{Pyr}_{\leftarrow\text{LP}}$ and $\text{Pyr}_{\leftarrow\text{ORBv1}}$ neurons. After an in-depth characterization of their electrophysiological properties, gene expression profiles, and axon innervation patterns, we concluded that they likely represent well-known Pyr subtypes: IT and PT neurons. Our finding establishes a link between the parallel processing of LP and ORBv1 information in V1 L5 and IT and PT subnetworks. Such insights into the selective innervation patterns within L5 of V1, combined with the identification of distinct neuronal subclasses, add a new dimension to our understanding of the functional roles of these neurons. Consequently, this contributes to a more nuanced understanding of top-down modulation in visual processing.

2) Moreover, although I appreciate the authors' laborious efforts, the finding that the $\text{Pyr}_{\leftarrow\text{LP}}$ and $\text{Pyr}_{\leftarrow\text{ORB}}$ neurons are differentiated by intrinsic properties and genetic markers (Fig. 4 and 5), and they have a different projection pattern (Fig. 6) is over-highlighted. The authors were "surprised (page 8 line 8)" by the differential expression of I_h between these cells, despite the motivation to further analyze these cells being the assumption that they fall into PT and IT cell classes. Indeed, PT and IT classes are well-known to be distinguished by the presence of I_h (and other intrinsic properties), genetic markers, and projection patterns (e.g., ref 36), thus, I do not think the findings (Fig 4-6) were as surprising as the authors claim.

RESPONSE:

Thank you for your comments regarding our characterization of $\text{Pyr}_{\leftarrow\text{LP}}$ and $\text{Pyr}_{\leftarrow\text{ORBv1}}$ neurons. We have removed the "surprise" claims from the revised manuscript. The revised text reads as " $\text{Pyr}_{\leftarrow\text{ORBv1}}$ neurons exhibited an I_h in response to this 10-mV

hyperpolarization, whereas $\text{Pyr}_{\leftarrow\text{LP}}$ neurons lacked this I_h —as indicated by an I_h slope (see Method) close to 0...” (P. 8, Lines 29-31).

We would also like to clarify that our initial focus in investigating these neurons was driven by the distinct top-down inputs they received, instead of the assumption that they fall into PT and IT cell classes. Following this line of inquiry, we conducted an investigation of these neuron’s properties, including electrophysiological properties, genetic markers, and projection patterns; and our results led to the realization of their alignment with previously established PT and IT neuron classes.

In our study, we initially defined $\text{Pyr}_{\leftarrow\text{LP}}$ and $\text{Pyr}_{\leftarrow\text{ORBv1}}$ neurons based on their inputs. However, during our investigation, we discovered that these neurons could also be informatively classified based on simple hyperpolarization data obtained through patch-clamp recordings (I_h slope), as demonstrated in our revised Fig. 5 (original Fig. 4) and our new Extended Data Fig. 10, in which we further tested the prediction accuracy of the SVM Classifier(I_h) in L5 Pyrs projecting to various downstream targets. In three groups of V1 L5 Pyrs that projecting to the RSP, ipsi-DMS, and SC, the SVM Classifier(I_h) achieved 100% prediction accuracy in each group.

This capacity to use simple hyperpolarization data to classify these neurons facilitated studies of their properties, and our data ultimately led us to conclude that these neurons, receiving distinct top-down inputs, fall into PT and IT cell classes (original Figs. 5-6, revised Figs. 6-7). We would also like to emphasize that, beyond identifying differences between cell types, our study establishes that top-down inputs from LP and ORBv1 are processed within the distinct subnetworks of IT and PT neurons in L5 of V1.

“Extended Data Fig. 10 | Performance of SVM Classifier(I_h) in distinguishing between Pyr \leftarrow LP and Pyr \leftarrow ORBvl neurons in various groups of retrogradely labeled V1 projecting-Pyrs. a, Schematic of the viral strategy (left) and the slice experiment (right) for examination of LP and ORBvl inputs in V1 L5 SC-projecting Pyrs. **b**, Performance of the SVM Classifier(I_h) in distinguishing Pyr \leftarrow LP from Pyr \leftarrow ORBvl neurons in the dataset of RSP-projecting Pyrs (the neuron types were initially defined based on their input patterns as determined upon independent optogenetic activation of LP and ORBvl inputs). Dashed line, decision boundary for classifying Pyr \leftarrow LP and Pyr \leftarrow ORBvl neurons. Red stars, input-pattern-classified Pyr \leftarrow LP neurons. n = 3 mice, 3 slices, 9 neurons. **c**, Prediction accuracy. **d-f**, Similar to a-c, but for the performance of SVM Classifier(I_h) on ipsi-DMS-projecting Pyrs. Red and green stars, input-pattern-classified Pyr \leftarrow LP and Pyr \leftarrow ORBvl neurons. n = 3 mice, 3 slices, 19 neurons. **j-i**, Similar

to a-c, but for the performance of SVM Classifier(I_h) on SC-projecting Pyrs. Green stars, input-pattern-classified $\text{Pyr}_{\leftarrow\text{ORBvI}}$ neurons. $n = 4$ mice, 4 slices, 14 neurons. In these three groups of L5 projecting Pyrs, the SVM Classifier(I_h) achieves 100% prediction accuracy in distinguishing $\text{Pyr}_{\leftarrow\text{LP}}$ from $\text{Pyr}_{\leftarrow\text{ORBvI}}$ neurons. Extended Data Fig. 10, Pannels a, d, g adapted from Petrucco, L. (2020). Mouse head schema. Zenodo. <https://doi.org/10.5281/zenodo.3925903> under a CC BY license: <https://creativecommons.org/licenses/by/4.0/>.”

3) Also, it is unclear whether the neurons connected and unconnected with LP axons occur at the same laminar position in L5, because there is clear projection bias for each afferent (LP axons target L5A while ORB targets L5B). L5A contains only IT neurons, so connection bias for LP axons is difficult to be explained by cell class (as the authors alluded) if the recording is done at the L5A for LP inputs assessment.

RESPONSE:

We thank the reviewer for highlighting the need to clarify the laminar positions of $\text{Pyr}_{\leftarrow\text{LP}}$ and $\text{Pyr}_{\leftarrow\text{ORBvI}}$ neurons in L5 of V1. Our revised main figure (Fig. 4h) demonstrates that the cell bodies of $\text{Pyr}_{\leftarrow\text{LP}}$ and $\text{Pyr}_{\leftarrow\text{ORBvI}}$ neurons extensively overlap within L5 of V1, indicating that they are not segregated into distinct sublayers.

“Fig. 4 | LP and ORBvl inputs target distinct subpopulations of L5 Pyrs. a, Schematic of the viral strategy for independent optogenetic activation of LP and ORBvl inputs. **b,** Fluorescent images showing the distribution of LP (red) and ORBvl axons (green) in V1. Scale bars, 100 μm . **c,** Normalized axon intensity of LP (red) and ORBvl (green) axons as a function of L5 depth. 0% and 100% represent the upper and lower boundaries of L5. $n = 12$ slices. **d,** Summary of L5 depth for peak LP (red) axon intensity and for peak ORBvl (green) axon intensity. $P = 5 \times 10^{-4}$. $***P < 0.001$; Wilcoxon sign-rank test. **e,** Schematic of the slice experiment examining independent optogenetic activation of LP and ORBvl inputs. **f,** EPSCs evoked by independent activation of LP and ORBvl inputs in L5 Pyrs. Red dots, 647-nm light stimulation, 10 Hz, 1 ms, 3.5 mW. Blue dots, 488-nm light stimulation, 10 Hz, 1 ms, 3.5 mW. Gray traces are individual trials, while red and green traces are averages of 5 trials. Scale bars, 100 ms, 100 pA. **g,** Normalized EPSC charge evoked by activation of LP and ORBvl inputs in Pyr $_{\leftarrow\text{LP}}$ (top) and Pyr $_{\leftarrow\text{ORBvl}}$ neurons (bottom). The circles indicate the means. $n = 12$ mice, 14 slices. Pyr $_{\leftarrow\text{LP}}$, $n = 28$ neurons, $P = 2 \times 10^{-44}$; Pyr $_{\leftarrow\text{ORBvl}}$, $n = 35$ neurons, $P = 9 \times 10^{-33}$. $***P < 0.001$, Paired t -test. **h,** Percentage (top) and distribution (bottom) of L5 Pyrs activated by LP and ORBvl inputs (Pyr $_{\leftarrow\text{LP}}$ and Pyr $_{\leftarrow\text{ORBvl}}$ neurons). Red, Pyr $_{\leftarrow\text{LP}}$ neurons; green, Pyr $_{\leftarrow\text{ORBvl}}$ neurons; gray, L5 Pyrs without any response to LP or ORBvl inputs. Data are presented as the mean \pm SEM. Figure 4, Pannels a adapted from Petrucco, L. (2020). Mouse head schema. Zenodo. <https://doi.org/10.5281/zenodo.3925903> under a CC BY license: <https://creativecommons.org/licenses/by/4.0/>.”

We would like to offer some additional context. To our understanding, it may not be the case that the L5A contains only IT neurons. For example, Mao *et al.*, 2011 reported the presence of PT neurons, which project to subcortical areas like zona incerta (ZI) and SC, in the L5A of vibrissal motor cortex (vM1) (shown in their Fig. 2). Moreover, the characteristics of L5 sublayers in V1 are less understood compared to those in the somatosensory and motor cortices. Our literature search only yield two papers differentiate L5A and L5B in V1 (Kim *et al.*, 2014; Xu *et al.*, 2016). Based on these two studies, the boundary between L5A and L5B in V1 is around the 50% of the L5 depth. Applying a similar criterion, retrogradely labeled L5 PT neurons in our current study, projecting to various subcortical regions such as the Pons, SC, and LP, exist in both L5A and L5B (Reviewer Fig. 2), consistent with the observed distribution of retrogradely labeled V1 L5 PT neurons in recent studies (Liang *et al.*, 2015; Lur *et al.*, 2016; Tang and Higley, 2020).

Additionally, Xu *et al.*, 2016 reported uniform action potential (AP)-related

electrophysiological properties in Pyramidal neurons across L5A and L5B in V1. This supports the notion of a mixed presence of IT and PT neurons within these sublayers. Therefore, given the complexity in the distribution of IT and PT neurons within V1's L5 and our observations of spatial intermingling of $\text{Pyr}_{\leftarrow\text{LP}}$ and $\text{Pyr}_{\leftarrow\text{ORBv1}}$ neurons, our results suggest that the distinct populations of L5 Pyrs innervated by LP and ORBv1 inputs cannot be solely attributed to their residing in different sublayers of L5. Instead, our findings indicate that these inputs innervate distinct cell classes within L5 of V1.

Reviewer Fig. 2 | Retrogradely labeled L5 PT neurons located in both L5A and L5B. **a**, Distribution of retrogradely labeled Pons-projecting Pyrs following injection of a Retro-AAV into the Pons. Left, Fluorescence images showing the distribution of retrogradely labeled projecting Pyrs (red) in V1. Right, the cell density of labeled projecting Pyrs across different layers in V1. The boundary between L5A and L5B in V1 is around the 50% of the L5 depth. Error bar, \pm SEM. Pons, $n = 3$ mice; Scale bar, $100 \mu\text{m}$. **b**, Similar to **a**, but for retrogradely labeled SC-projecting Pyrs. SC, $n = 3$ mice. **c**, Similar to **a**, but for retrogradely labeled LP-projecting Pyrs. LP, $n = 3$ mice.

Other comments:

4) In Fig 6 (and patch-seq experiment), the authors seem to define the $\text{Pyr}_{\leftarrow\text{LP}}$ or $\text{Pyr}_{\leftarrow\text{OBR}}$ neurons based solely on SVM and I_h (i.e., without verifying the connectivity). Although I understand the predictive power of the classifier, after all, it only uses one parameter (I_h slope) and I have some reservations about whether this approach qualifies to classify the neurons defined by its input type.

RESPONSE:

We have conducted additional control experiments to validate the predictions of the SVM Classifier in identifying $\text{Pyr}_{\leftarrow\text{LP}}$ and $\text{Pyr}_{\leftarrow\text{ORBv1}}$ neurons in retrogradely labeled L5 Pyrs projecting to the RSP, ipsi-DMS, and SC (kindly see our response to the 2nd

comment). These experiments involved independent optogenetic activation of LP and ORBv1 inputs in the same brain slice, followed by measuring input strengths in L5 Pyrs. For each animal, we injected AAVs expressing ChrimsonR-tdTomato in the LP and Chr2-EYFP in the ORBv1, alongside Retro-AAV expressing mCherry in either RSP, ipsi-DMS, or SC (Extended Data Fig. 10, kindly see our response to the 2nd comment). In L5 Pyrs retrogradely labeled by Retro-AAV, we also measured the I_h slope and applied the SVM Classifier(I_h) for neuron type prediction. We found that the SVM Classifier(I_h) effectively distinguished $\text{Pyr}_{\leftarrow\text{LP}}$ and $\text{Pyr}_{\leftarrow\text{ORBv1}}$ neurons across groups, including those projecting to the RSP, ipsi-DMS, and SC, with 100% accuracy in each group. This new data reinforces that the SVM Classifier(I_h) is an effective tool for distinguishing between L5 $\text{Pyr}_{\leftarrow\text{LP}}$ and $\text{Pyr}_{\leftarrow\text{ORBv1}}$ neurons.

The revised text is as follows (P. 12, Lines 11-24):

“Using the SVM Classifier(I_h), we determined that more than 78% of labeled L5 Pyrs projecting to cortical areas (V2M, RSP, and ORBv1) and contra-DMS were classified as $\text{Pyr}_{\leftarrow\text{LP}}$ neurons. In contrast, more than 82% of labeled L5 Pyrs projecting to other subcortical areas (Pons, SC, and LP) were classified as $\text{Pyr}_{\leftarrow\text{ORBv1}}$ neurons (Fig. 6c). Additionally, labeled L5 Pyrs from ipsi-DMS contained 27% $\text{Pyr}_{\leftarrow\text{ORBv1}}$ neurons and 73% $\text{Pyr}_{\leftarrow\text{LP}}$ neurons. Notably, we conducted independent optogenetic activation experiments on L5 Pyrs projecting to a subset of these downstream targets, including the RSP, ipsi-DMS, and SC. These experiments confirmed that the recorded L5 projecting Pyrs exclusively receive either LP or ORBv1 inputs, a selectivity pattern accurately predicted by the SVM Classifier(I_h) (Extended Data Fig. 10). Specifically, we observed that 100% of L5 RSP-projecting Pyrs receive LP inputs, 100% of SC-projecting Pyrs receive ORBv1 inputs, and among ipsi-DMS-projecting Pyrs 47% receive LP inputs while 53% receive ORBv1 inputs. The SVM Classifier(I_h) effectively distinguished $\text{Pyr}_{\leftarrow\text{LP}}$ and $\text{Pyr}_{\leftarrow\text{ORBv1}}$ neurons across these groups with 100% accuracy.”

5) The authors are focused on the idea of the subnetwork; I'm wondering if these parallel (and potentially competing) circuit may interact/converge at the level of the local circuit since $\text{Pyr}_{\leftarrow\text{LP}}$ neurons (IT type) is likely to be upstream of $\text{Pyr}_{\leftarrow\text{ORBv1}}$ neurons (PT type) based on current knowledge of local circuit organization in cortex in general.

RESPONSE:

We appreciate the reviewer's question regarding the interaction of PT and IT neurons at the local circuit level. We have revised the discussion section to address this; the revised content reads as follows (P. 16, Lines 13-29 and P. 17, Lines 1-9):

“PT neurons receive extensive inputs from local IT neurons; however, they provide little local feedback, a phenomenon observed in various cortical regions such as the V1, somatosensory, and frontal cortices (Brown and Hestrin, 2009; Harris and Shepherd, 2015; Morishima and Kawaguchi, 2006; Petreanu et al., 2009). Moreover, PT neurons in the barrel cortex are known to receive strong bottom-up TC inputs from the VPM, yet receive few top-down TC inputs from the PO (Constantinople and Bruno, 2013; Petreanu et al., 2009). These findings suggest that PT neurons may act as downstream integrators in local circuits: integrating the information from local IT neurons with bottom-up TC inputs and broadcasting the results to subcortical structures.

Recent behavioral studies suggest an alternative interpretation, proposing that IT and PT neurons form parallel subnetworks capable of independent processing. For instance, only PT neurons are required for tactile or visual perception in sensory cortices, while specific PT neurons in the secondary motor cortex are involved in motor generation (Economo et al., 2018; Li et al., 2015; Takahashi et al., 2020; Tang and Higley, 2020). Moreover, widefield calcium imaging assessing the spatial and temporal dynamics of Pyr-type-specific subnetworks during sensorimotor tasks has revealed distinct cortex-wide activity patterns for IT and PT neurons (Mohan et al., 2023; Musall et al., 2023). It is possible that information flow between IT and PT in local circuits is dynamically gated by inhibitory and modulatory mechanisms according to behavioral demand.

Building on this notion, our findings suggest a mechanism for the dynamic gating of information flow between IT and PT subnetworks. We found that that LP and ORBv1 inputs drive mutual inhibition between IT and PT subnetworks in V1. First, we observed that ORBv1 inputs drive multisynaptic inhibition in IT neurons. Second, LP inputs induce multisynaptic inhibition in PT neurons (Pyr \leftarrow ORBv1 neurons) without multisynaptic excitation from activated IT neurons (Pyr \leftarrow LP neurons), suggesting that LP inputs trigger APs in L5 PV+ neurons and lead to subthreshold excitation in IT neurons. Assuming that the multisynaptic inhibition resulting from LP inputs to PT neurons precedes excitation from activated IT neurons, it is plausible that LP inputs suppress PT neuron activity and thus temporally separate the activity of IT and PT neurons.”

6) *Fig. 1 and 2: Please state how the recording is made to obtain input data across the different layers. Did the authors record at least one neuron in each layer in one slice or mice? The sample size for the cells appears to vary from layer to layer and is typically heavily skewed to the layer where the most inputs are detected. Why?*

RESPONSE:

We thank the reviewer for pointing out this issue. In our experiments presented in Figs. 1 and 2, we recorded neurons simultaneously across different layers in each experiment. Specifically, we recorded at least one neuron in each layer in one mouse in at least 5 mice in each group. Additionally, for the majority of recorded animals, we recorded neurons simultaneously in three or more layers. We have revised the Methods section in our manuscript, which now reads: “For the experiments in Fig. 1 and Fig. 2, we recorded neurons simultaneously in each layer in at least 5 mice in each group. For the majority of recorded animals, we recorded neurons simultaneously in three or more layers.”

Given the observed variability in neuronal responses, particularly in the layers receiving the strongest inputs, we examined relatively more neurons in these layers, seeking to understand population-level responses.

Moreover, following Reviewer #4’s guidance, we reanalyzed our data and included normalized EPSP amplitude for each top-down input to validate our original findings (see new Extended Data Figs. 3 and 4). For V2M and ACA inputs, we normalized the EPSP amplitude of each recorded neuron to the averaged EPSP amplitude of L6 Pyrs in the same slice. For ORBvl and LP inputs—considering that the Pyrs in their most responsive layer (L5) can be divided into distinct subtypes according to their responses to these inputs (as shown in our revised main Fig. 4)—we normalized the EPSP amplitude against $\text{Pyr}_{\leftarrow\text{ORBvl}}$ and $\text{Pyr}_{\leftarrow\text{LP}}$ neurons, respectively. The results we obtained using this normalization approach are consistent with the differences in input strengths across layers and cell types reported in the originally submitted manuscript.

“Extended Data Fig. 3 | Layer specificity of CC and TC top-down inputs based on normalized EPSP amplitude. **a**, Normalized EPSP amplitudes evoked by optogenetic activation of V2M inputs in Pyrs across the indicated layers. Left, box plot showing distribution of EPSP amplitude from Pyrs in the indicated layers. Edges, 25th and 75th percentiles; central line, median; whiskers, 1.5× the interquartile range of the edges. Circles are outliers, defined as values more than 3 times the interquartile range (IQR) from the median. V2M inputs activated the Pyrs in L2/3 and L6, with the strongest input strength in L6 (significant differences were observed across layers $F(3,199) = 20.7$, $P = 1 \times 10^{-11}$, one-way ANOVA; L6 vs. other layers, $P < 0.002$, Tukey’s post hoc test). Right, matrix of normalized EPSP amplitude across layers of V1 for V2M inputs. **b**, Similar to **a**, but for ACA inputs. ACA inputs activated the Pyrs in L2/3 and L6, with stronger input strength in L6 (significant differences were observed across layers, $F(3,152) = 10.2$, $P = 4 \times 10^{-6}$, one-way ANOVA; L6 vs. L4 and L5, $P < 0.004$; L6 vs. L2/3, $P = 0.87$, Tukey’s post hoc test). **c**, Similar to **a**, but for ORBvl inputs. ORBvl inputs only activated the Pyrs in L5 (significant differences were observed across layers, $F(3,101) = 13.4$, $P = 2 \times 10^{-7}$, one-way ANOVA; L5 vs. other layers, $P < 5 \times 10^{-4}$, Tukey’s post hoc test). **d**, Similar to **a**, but for LP inputs. LP inputs activated the Pyrs in L2/3, L5, and L6, with the strongest input strength in L5 (significant differences were observed across layers, $F(3,204) = 15.8$, $P = 3 \times 10^{-9}$, one-way ANOVA; L5 vs. other layers, $P < 2 \times 10^{-5}$, Tukey’s post hoc test). The number of neurons in each group is indicated by the numbers displayed in the figure. The EPSP amplitudes of V2M and ACA inputs were normalized to the averaged L6 Pyr EPSP amplitude recorded the same day. Given the

observed heterogeneity in L5 Pyr subtypes responding to ORBv1 and LP inputs, we normalized the EPSP amplitudes of ORBv1 and LP inputs to the average for $\text{Pyr} \leftarrow \text{ORBv1}$ and for $\text{Pyr} \leftarrow \text{LP}$ neurons. The data included are those with Pyrs recorded from the most responsive layer in the same brain slice.”

“Extended Data Fig. 4 | Normalized input strengths of CC and

TC top-down inputs in four types of V1 inhibitory neurons across different layers. a, Distribution of normalized EPSP amplitude from inhibitory neurons in different layers. Box plots indicate the median and the respective quartiles. Circles are outliers, defined as values above 3 IQR from the median. **b**, Matrix of normalized EPSP amplitude in different types of inhibitory neurons across layers of V1 for V2M inputs. **c-d**, Similar to **a-b**, but for ACA inputs. **e-f**, Similar to **a-b**, but for ORBv1 inputs. **g-h**, Similar to **a-b**, but for LP inputs. The number of neurons in each group is indicated by the numbers displayed in the figure. The EPSP amplitudes of V2M and ACA inputs were normalized to the averaged L6 Pyr EPSP amplitude recorded the same day. Given the observed heterogeneity in L5 Pyr subtypes responding to ORBv1 and LP inputs, we normalized the EPSP amplitudes of ORBv1 and LP inputs to the average for Pyr \leftarrow ORBv1 and for Pyr \leftarrow LP neurons. The data included are those with Pyrs recorded from the most responsive layer in the same brain slice.”

7) *How many animals were used for each recording (in non-supplemental figures)? What is the maximum/minimum cell number represented by one animal? Did the authors examine the sex effect?*

RESPONSE:

Based on Reviewer#4’s and your guidance, we have added the numbers of mice and slices used in each experiment. For the experiments shown in Figs.1 and 2, as well as in Extended Data Figs. 3 and 4, the number of mice and slices have now been added to revised Supplementary Tables 1, 3, 4, and 5. For other experiments, we have added this information in the figure captions. The revised text reads as follows:

“**Fig.3 | g**, Normalized EPSC charge evoked by activation of V2M and ACA inputs in Pyr \leftarrow both (top) and Pyr \leftarrow V2M neurons (bottom). The circles indicate the means. n = 5 mice, 7 slices. Pyr \leftarrow both, n = 37 neurons; Pyr \leftarrow V2M, n = 26 neurons.”

“**Fig.4 | g**, Normalized EPSC charge evoked by activation of LP and ORBv1 inputs in Pyr \leftarrow LP (top) and Pyr \leftarrow ORBv1 neurons (bottom). The circles indicate the means. n = 12 mice, 14 slices. Pyr \leftarrow LP, n = 28 neurons; Pyr \leftarrow ORBv1, n = 35 neurons.”

“**Fig.6 | c**, All of the recorded Pyr \leftarrow LP neurons exhibited regular spiking (RS), whereas 34% of the Pyr \leftarrow ORBv1 showed RS and 66% exhibited burst spiking (BS) (n = 20 mice, 20 slices; Pyr \leftarrow LP, n = 47 neurons; Pyr \leftarrow ORBv1, n = 44 neurons; P = 6×10^{-10} , Kolmogorov-Smirnov test).”

“**Fig.7 | c**, Proportion of $\text{Pyr}_{\leftarrow\text{LP}}$ and $\text{Pyr}_{\leftarrow\text{ORBvI}}$ neurons in labeled projecting Pyrs. The number of neurons in each group is indicated by the numbers displayed in the figure. V2M, $n = 4$ mice, 5 slices; RSP, $n = 3$ mice, 5 slices; ORBvI, $n = 3$ mice, 3 slices; Ipsi-DMS, $n = 4$ mice, 5 slices; Contra-DMS, $n = 3$ mice, 5 slices; Pons, $n = 3$ mice, 3 slices; SC, $n = 3$ mice, 3 slices; LP, $n = 5$ mice, 5 slices.”

“**Fig.8 | c**, Contribution of inhibitory inputs to total synaptic inputs, measured as $\text{IPSC}_{\text{charge}}/(\text{IPSC}_{\text{charge}}+\text{EPSC}_{\text{charge}})$. LP inputs drive significantly stronger inhibition relative to excitation in $\text{Pyr}_{\leftarrow\text{ORBvI}}$ neurons than in $\text{Pyr}_{\leftarrow\text{LP}}$ neurons. $P = 5 \times 10^{-7}$, Wilcoxon rank-sum test. $n = 8$ mice, 9 slices. **g**, ORBvI inputs drive significantly stronger inhibition relative to excitation in $\text{Pyr}_{\leftarrow\text{LP}}$ neurons than in $\text{Pyr}_{\leftarrow\text{ORBvI}}$ neurons. $P = 1 \times 10^{-7}$. $n = 4$ mice, 5 slices.”

“**Extended Data Fig. 5 | b**, TTX and 4-AP had no effect on I_h slope of $\text{Pyr}_{\leftarrow\text{LP}}$ (red) and $\text{Pyr}_{\leftarrow\text{ORBvI}}$ (green) neurons. $n = 3$ mice, 3 slices. $\text{Pyr}_{\leftarrow\text{LP}}$, $n = 13$ neurons, $P = 0.12$; $\text{Pyr}_{\leftarrow\text{ORBvI}}$, $n = 14$ neurons, $P = 0.07$; Wilcoxon signed-rank test.”

“**Extended Data Fig. 10 | b**, Performance of the SVM Classifier(I_h) in distinguishing $\text{Pyr}_{\leftarrow\text{LP}}$ from $\text{Pyr}_{\leftarrow\text{ORBvI}}$ neurons in the dataset of RSP-projecting Pyrs (the neuron types were initially defined based on their input patterns as determined upon independent optogenetic activation of LP and ORBvI inputs). Dashed line, decision boundary for classifying $\text{Pyr}_{\leftarrow\text{LP}}$ and $\text{Pyr}_{\leftarrow\text{ORBvI}}$ neurons. Red stars, input-pattern-classified $\text{Pyr}_{\leftarrow\text{LP}}$ neurons. $n = 3$ mice, 3 slices, 9 neurons. **c**, Prediction accuracy. **d-f**, Similar to a-c, but for the performance of SVM Classifier(I_h) on ipsi-DMS-projecting Pyrs. Red and green stars, input-pattern-classified $\text{Pyr}_{\leftarrow\text{LP}}$ and $\text{Pyr}_{\leftarrow\text{ORBvI}}$ neurons. $n = 3$ mice, 3 slices, 19 neurons. **j-i**, Similar to a-c, but for the performance of SVM Classifier(I_h) on SC-projecting Pyrs. Green stars, input-pattern-classified $\text{Pyr}_{\leftarrow\text{ORBvI}}$ neurons. $n = 4$ mice, 4 slices, 14 neurons.”

“**Extended Data Fig. 12 | c**, Normalized PV-IPSC amplitudes in $\text{Pyr}_{\leftarrow\text{ORBvI}}$ neurons were significantly larger than in $\text{Pyr}_{\leftarrow\text{LP}}$ neurons. $n = 3$ mice, 4 slices. $\text{Pyr}_{\leftarrow\text{LP}}$, $n = 24$ neurons, $\text{Pyr}_{\leftarrow\text{ORBvI}}$, $n = 21$ neurons. **d-f**, Similar to a-c, but for SST+ neuron-induced IPSCs. $n = 5$ mice, 7 slices. $\text{Pyr}_{\leftarrow\text{LP}}$, $n = 19$ neurons, $\text{Pyr}_{\leftarrow\text{ORBvI}}$, $n = 35$ neurons. **g-i**, Similar to a-c, but for VIP+ neuron-induced IPSCs. $n = 4$ mice, 5 slices. $\text{Pyr}_{\leftarrow\text{LP}}$, $n = 16$ neurons, $\text{Pyr}_{\leftarrow\text{ORBvI}}$, $n = 13$ neurons.”

The maximum/minimum cell number represented by one animal in main figures are shown in the following table (Reviewer Table 1).

	Fig. 1	Fig. 2	Fig. 3	Fig. 4	Fig. 6	Fig. 7	Fig. 8
max (neurons/mouse)	14	13	15	10	6	18	12
min (neurons/mouse)	6	7	7	4	4	10	6

Reviewer Table 1 | The maximum/minimum cell number represented by one animal in main figures.

We incorporated both male and female mice, the number of each sex used in difference experiments are provided in the source data. We did not specifically examine the effects of sex in our experimental design. Our investigation focused on the neuronal circuits involved in top-down modulation of visual processing, and previous studies on mice have not shown any sex influence in this process (Oldenburg et al., 2024; Yoshida and Ohki, 2020; Zhuang et al., 2021).

Following reviewer’s guidance, we reanalyzed our data to examine potential sex effects. Our analysis revealed no significant differences between male and female mice in the layer- and cell-type-specific innervation patterns of CC and TC top-down inputs in V1 (Reviewer Figs. 3 and 4). Moreover, we found that both male and female mice exhibit similar processing strategies during their interaction in deep layers (Reviewer Figs. 5 and 6). These findings indicate that the innervation and interaction patterns of CC and TC top-down inputs in V1 are consistent across sexes in our study. We have revised our method section accordingly. The revised text reads as follows: “Our experiments used male and female mice. We found no differences between male and female mice in the layer- and cell-type-specific innervation patterns of CC or TC top-down inputs in V1 or for interactions among these inputs in deep layers.”

Reviewer Fig. 3 | Similar innervation patterns of CC and TC top-down inputs in V1 Pyrs across different layers in both male and female mice. **a**, EPSP amplitudes evoked by optogenetic activation of V2M inputs in Pyrs across different layers in male (blue) and female (pink) mice. Box plot showing distribution of EPSP amplitude from Pyrs in the indicated layers. Edges, 25th and 75th percentiles; central line, median; whiskers, 1.5× the interquartile range of the edges. Circles are outliers, defined as values more than 3 times the interquartile range (IQR) from the median. No significant sex effect was observed for the input strengths of V2M inputs in V1 Pyrs across different layers ($F_{sex}(1,195) = 1.2$, $P_{sex} = 0.3$; $F_{sex*layer}(3,195) = 0.7$, $P_{sex*layer} = 0.6$; two-way ANOVA). **b**, Similar to a, but for ACA inputs. No significant sex effect was observed for the input strengths of ACA inputs in V1 Pyrs across different layers ($F_{sex}(1,156) = 0.2$, $P_{sex} = 0.7$; $F_{sex*layer}(3,156) = 0.2$, $P_{sex*layer} = 0.9$; two-way ANOVA). **c**, Similar to a, but for ORBvl inputs. No significant sex effect was observed for the input strengths of ORBvl inputs in V1 Pyrs across different layers ($F_{sex}(1,176) = 0.02$, $P_{sex} = 0.9$; $F_{sex*layer}(3,176) = 0.03$, $P_{sex*layer} = 0.99$; two-way ANOVA). **d**, Similar to a, but for LP inputs. No significant sex effect was observed for the input strengths of LP inputs in V1 Pyrs across different layers ($F_{sex}(1,321) = 0.1$, $P_{sex} = 0.7$; $F_{sex*layer}(3,321) = 0.3$, $P_{sex*layer} = 0.8$; two-way ANOVA). The number of neurons in each group is indicated by the numbers displayed in the figure.

Reviewer Fig. 4 | Similar innervation patterns of CC and TC top-down inputs in V1 inhibitory neurons across different layers in both male and female mice. a, EPSP amplitudes evoked by optogenetic activation of V2M inputs in inhibitory neurons across different layers in male (blue) and female (pink) mice. Box plot showing the distribution of EPSP amplitudes from Pyrs in the indicated layers. No significant sex effect was observed for the input strengths of V2M inputs in V1 inhibitory neurons across different layers ($F_{sex}(1,180) = 1.5$, $P_{sex} = 0.2$; $F_{sex*layer}(3,180) = 0.6$, $P_{sex*layer} = 0.6$; $F_{sex*cell-type}(2,180) = 2.3$, $P_{sex*cell-type} = 0.1$; $F_{sex*layer*cell-type}(6,180) = 0.7$, $P_{sex*layer*cell-type} = 0.7$; three-way ANOVA). **b,** Similar to a, but for ACA inputs. No significant sex effect was observed for the input strengths of ACA inputs in V1 inhibitory neurons across different layers ($F_{sex}(1,176) = 0.4$, $P_{sex} = 0.6$; $F_{sex*layer}(3,176) = 0.6$, $P_{sex*layer} = 0.6$; $F_{sex*cell-type}(2,176) = 0.7$, $P_{sex*cell-type} = 0.5$; $F_{sex*layer*cell-type}(6,176) = 1.8$, $P_{sex*layer*cell-type} = 0.1$; three-way ANOVA). **c,** Similar to a, but for ORBvl inputs. No significant sex effect was observed for the input strengths of ORBvl inputs in V1 inhibitory neurons across different layers ($F_{sex}(1,275) = 1.1$, $P_{sex} = 0.3$; $F_{sex*layer}(3,275) = 0.4$, $P_{sex*layer} =$

0.8; $F_{sex*cell-type}(2,275) = 0.4$, $P_{sex*cell-type} = 0.7$; $F_{sex*layer*cell-type}(6,275) = 0.5$, $P_{sex*layer*cell-type} = 0.8$; three-way ANOVA). **d**, Similar to a, but for LP inputs. No significant sex effect was observed for the input strengths of LP inputs in V1 inhibitory neurons across different layers ($F_{sex}(1,238) = 1.1$, $P_{sex} = 0.3$; $F_{sex*layer}(3,238) = 1.4$, $P_{sex*layer} = 0.3$; $F_{sex*cell-type}(2,238) = 0.9$, $P_{sex*cell-type} = 0.4$; $F_{sex*layer*cell-type}(6,238) = 0.3$, $P_{sex*layer*cell-type} = 0.9$; three-way ANOVA). The number of neurons in each group is indicated by the numbers displayed in the figure.

Reviewer Fig. 5 | Similar innervation patterns of V2M and ACA inputs in V1 L6 Pyrs in both male and female mice. **a-b**, Normalized EPSC charge evoked by activation of V2M and ACA inputs in Pyr_{←both} (a) and Pyr_{←V2M} (b) neurons in male mice. The circles indicate the means. Pyr_{←both}, n = 21 neurons; Pyr_{←V2M}, n = 11 neurons. **c-d**, Similar to a-b, but for Pyr_{←both} (c) and Pyr_{←V2M} (d) neurons in female mice. Pyr_{←both}, n = 16 neurons; Pyr_{←V2M}, n = 15 neurons. No significant sex effect was observed for the normalized input strengths of V2M or ACA inputs in V1 L6 Pyrs (Within-Subject Effects Test: $F_{EPSC*sex}(1,59) = 0.04$, $P_{EPSC*sex} = 0.83$; $F_{EPSC*sex*CellType}(1,59) = 0.002$,

$P_{EPSC*sex*CellType} = 0.96$; Between-Subject Effects Test: $F_{sex}(1,59) = 0.04$, $P_{sex} = 0.83$; $F_{sex*CellType}(1,59) = 0.002$, $P_{sex*CellType} = 0.96$; two-way repeated ANOVA).

Reviewer Fig. 6 | Similar innervation patterns of LP and ORBv1 inputs in V1 L5 Pyrs in both male and female mice. **a-b**, Normalized EPSC charge evoked by activation of LP and ORBv1 inputs in Pyr_{-LP} (a) and Pyr_{-ORBv1} (b) neurons in male mice. The circles indicate the means. Pyr_{-LP}, n = 18 neurons; Pyr_{-ORBv1}, n = 28 neurons. **c-d**, Similar to a-b, but for Pyr_{-LP} (c) and Pyr_{-ORBv1} (d) neurons in female mice. Pyr_{-LP}, n = 10 neurons; Pyr_{-ORBv1}, n = 7 neurons. No significant sex effect was observed for the input strengths of LP or ORBv1 inputs in V1 L5 Pyrs (Within-Subject Effects Test: $F_{EPSC*sex}(1,59) = 0.5$, $P_{EPSC*sex} = 0.5$; $F_{EPSC*sex*CellType}(1,59) = 0.6$, $P_{EPSC*sex*CellType} = 0.5$; Between-Subject Effects Test: $F_{sex}(1,59) = 0.3$, $P_{sex} = 0.6$; $F_{sex*CellType}(1,59) = 0.4$, $P_{sex*CellType} = 0.5$; two-way repeated ANOVA).

Let us take this opportunity to express our appreciation for the insightful comments, which have been highly valuable during our revision process.

References

- Brown, S.P., and Hestrin, S. (2009). Intracortical circuits of pyramidal neurons reflect their long-range axonal targets. *Nature* *457*, 1133-1136.
- Constantinople, C.M., and Bruno, R.M. (2013). Deep cortical layers are activated directly by thalamus. *Science* *340*, 1591-1594.
- D'Souza, R.D., Meier, A.M., Bista, P., Wang, Q., and Burkhalter, A. (2016). Recruitment of inhibition and excitation across mouse visual cortex depends on the hierarchy of interconnecting areas. *Elife* *5*, e19332.
- Debes, S.R., and Dragoi, V. (2023). Suppressing feedback signals to visual cortex abolishes attentional modulation. *Science* *379*, 468-473.
- Economo, M.N., Viswanathan, S., Tasic, B., Bas, E., Winnubst, J., Menon, V., Graybiel, L.T., Nguyen, T.N., Smith, K.A., Yao, Z., *et al.* (2018). Distinct descending motor cortex pathways and their roles in movement. *Nature* *563*, 79-84.
- Fiser, A., Mhringer, D., Oyibo, H.K., Petersen, A.V., Leinweber, M., and Keller, G.B. (2016). Experience-dependent spatial expectations in mouse visual cortex. *Nat. Neurosci* *19*, 1658-1664.
- Fishell, G., and Kepecs, A. (2020). Interneuron Types as Attractors and Controllers. *Annu. Rev. Neurosci.* *43*, 1-30.
- Fishell, G., and Rudy, B. (2011). Mechanisms of inhibition within the telencephalon: "where the wild things are". *Annu. Rev. Neurosci.* *34*, 535-567.
- Gilbert, C.D., and Li, W. (2013). Top-down influences on visual processing. *Nat. Rev. Neurosci.* *14*, 350-363.
- Harris, J.A., Mihalas, S., Hirokawa, K.E., Whitesell, J.D., Choi, H., Bernard, A., Bohn, P., Caldejon, S., Casal, L., Cho, A., *et al.* (2019). Hierarchical organization of cortical and thalamic connectivity. *Nature* *575*, 195-202.
- Harris, K.D., and Shepherd, G.M. (2015). The neocortical circuit: themes and variations. *Nat. Neurosci.* *18*, 170-181.
- Hu, F., Kamigaki, T., Zhang, Z., Zhang, S., Dan, U., and Dan, Y. (2019). Prefrontal Corticotectal Neurons Enhance Visual Processing through the Superior Colliculus and Pulvinar Thalamus. *Neuron* *104*, 1141-1152.
- Huda, R., Sipe, G.O., Breton-Provencher, V., Cruz, K.G., Pho, G.N., Adam, E., Gunter, L.M., Sullins, A., Wickersham, I.R., and Sur, M. (2020). Distinct prefrontal top-down circuits differentially modulate sensorimotor behavior. *Nat. Commun.* *11*, 6007.
- Javadzadeh, M., and Hofer, S.B. (2022). Dynamic causal communication channels between neocortical areas. *Neuron* *110*, 2470-2483.
- Kim, J., Matney, C.J., Blankenship, A., Hestrin, S., and Brown, S.P. (2014). Layer 6 corticothalamic neurons activate a cortical output layer, layer 5a. *J. Neurosci.* *34*, 9656-9664.
- Kim, J.H., Ma, D.H., Jung, E., Choi, I., and Lee, S.H. (2021). Gated feedforward inhibition in the frontal cortex releases goal-directed action. *Nat. Neurosci.* *24*, 1452-1464.
- Leinweber, M., Ward, D.R., Sobczak, J.M., Attinger, A., and Keller, G.B. (2017). A Sensorimotor Circuit in Mouse Cortex for Visual Flow Predictions. *Neuron* *95*, 1420-1432.
- Li, N., Chen, T.W., Guo, Z.V., Gerfen, C.R., and Svoboda, K. (2015). A motor cortex circuit for motor planning and movement. *Nature* *519*, 51-56.
- Liang, F., Xiong, X.R., Zingg, B., Ji, X.Y., Zhang, L.I., and Tao, H.W. (2015). Sensory Cortical Control

of a Visually Induced Arrest Behavior via Corticotectal Projections. *Neuron* 86, 755-767.

Liu, D., Deng, J., Zhang, Z., Zhang, Z.Y., Sun, Y.G., Yang, T., and Yao, H. (2020). Orbitofrontal control of visual cortex gain promotes visual associative learning. *Nat. Commun.* 11, 2784.

Losonczy, A., and Magee, J.C. (2006). Integrative properties of radial oblique dendrites in hippocampal CA1 pyramidal neurons. *Neuron* 50, 291-307.

Lur, G., Vinck, M.A., Tang, L., Cardin, J.A., and Higley, M.J. (2016). Projection-Specific Visual Feature Encoding by Layer 5 Cortical Subnetworks. *Cell Rep.* 14, 2538-2545.

Ma, G., Liu, Y., Wang, L., Xiao, Z., Song, K., Wang, Y., Peng, W., Liu, X., Wang, Z., Jin, S., *et al.* (2021). Hierarchy in sensory processing reflected by innervation balance on cortical interneurons. *Sci. Adv.* 7, abf5676.

Mao, T., Kusefoglou, D., Hooks, B.M., Huber, D., Petreanu, L., and Svoboda, K. (2011). Long-range neuronal circuits underlying the interaction between sensory and motor cortex. *Neuron* 72, 111-123.

Matho, K.S., Huilgol, D., Galbavy, W., He, M., Kim, G., An, X., Lu, J., Wu, P., Di Bella, D.J., Shetty, A.S., *et al.* (2021). Genetic dissection of the glutamatergic neuron system in cerebral cortex. *Nature* 598, 182-187.

Mohan, H., An, X., Xu, X.H., Kondo, H., Zhao, S., Matho, K.S., Wang, B.S., Musall, S., Mitra, P., and Huang, Z.J. (2023). Cortical glutamatergic projection neuron types contribute to distinct functional subnetworks. *Nat. Neurosci.* 26, 481-494.

Moore, T., and Zirnsak, M. (2017). Neural Mechanisms of Selective Visual Attention. *Annu. Rev. Psychol.* 68, 47-72.

Morishima, M., and Kawaguchi, Y. (2006). Recurrent connection patterns of corticostriatal pyramidal cells in frontal cortex. *J. Neurosci.* 26, 4394-4405.

Musall, S., Sun, X.R., Mohan, H., An, X., Gluf, S., Li, S.J., Drewes, R., Cravo, E., Lenzi, I., Yin, C., *et al.* (2023). Pyramidal cell types drive functionally distinct cortical activity patterns during decision-making. *Nat. Neurosci.* 26, 495-505.

Norman, K.J., Riceberg, J.S., Koike, H., Bateh, J., McCraney, S.E., Caro, K., Kato, D., Liang, A., Yamamuro, K., Flanigan, M.E., *et al.* (2021). Post-error recruitment of frontal sensory cortical projections promotes attention in mice. *Neuron* 109, 1202-1213.

Oh, S.W., Harris, J.A., Ng, L., Winslow, B., Cain, N., Mihalas, S., Wang, Q., Lau, C., Kuan, L., Henry, A.M., *et al.* (2014). A mesoscale connectome of the mouse brain. *Nature* 508, 207-214.

Oldenburg, I.A., Hendricks, W.D., Handy, G., Shamardani, K., Bounds, H.A., Doiron, B., and Adesnik, H. (2024). The logic of recurrent circuits in the primary visual cortex. *Nat. Neurosci.* 27, 137-147.

Petreanu, L., Mao, T., Sternson, S.M., and Svoboda, K. (2009). The subcellular organization of neocortical excitatory connections. *Nature* 457, 1142-1145.

Polsky, A., Mel, B.W., and Schiller, J. (2004). Computational subunits in thin dendrites of pyramidal cells. *Nat. Neurosci.* 7, 621-627.

Takahashi, N., Ebner, C., Sigl-Glockner, J., Moberg, S., Nierwetberg, S., and Larkum, M.E. (2020). Active dendritic currents gate descending cortical outputs in perception. *Nat. Neurosci.* 23, 1277-1285.

Tang, L., and Higley, M.J. (2020). Layer 5 Circuits in V1 Differentially Control Visuomotor Behavior. *Neuron* 105, 346-354.

Tasic, B., Yao, Z., Graybiel, L.T., Smith, K.A., Nguyen, T.N., Bertagnolli, D., Goldy, J., Garren, E., Economo, M.N., Viswanathan, S., *et al.* (2018). Shared and distinct transcriptomic cell types across

neocortical areas. *Nature* *563*, 72-78.

Xu, X., Olivas, N.D., Ikrar, T., Peng, T., Holmes, T.C., Nie, Q., and Shi, Y. (2016). Primary visual cortex shows laminar-specific and balanced circuit organization of excitatory and inhibitory synaptic connectivity. *J. Physiol.* *594*, 1891-1910.

Yao, S., Wang, Q., Hirokawa, K.E., Ouellette, B., Ahmed, R., Bomben, J., Brouner, K., Casal, L., Caldejon, S., Cho, A., *et al.* (2023). A whole-brain monosynaptic input connectome to neuron classes in mouse visual cortex. *Nat. Neurosci.* *26*, 350-364.

Yao, Z., van Velthoven, C.T.J., Nguyen, T.N., Goldy, J., Sedeno-Cortes, A.E., Baftizadeh, F., Bertagnolli, D., Casper, T., Chiang, M., Crichton, K., *et al.* (2021). A taxonomy of transcriptomic cell types across the isocortex and hippocampal formation. *Cell* *184*, 3222-3241.

Yoshida, T., and Ohki, K. (2020). Natural images are reliably represented by sparse and variable populations of neurons in visual cortex. *Nat. Commun.* *11*, 872.

Young, H., Belbut, B., Baeta, M., and Petreanu, L. (2021). Laminar-specific cortico-cortical loops in mouse visual cortex. *Elife* *10*, e59551.

Yuste, R., Hawrylycz, M., Aalling, N., Aguilar-Valles, A., Arendt, D., Arnedillo, R.A., Ascoli, G.A., Bielza, C., Bokharaie, V., Bergmann, T.B., *et al.* (2020). A community-based transcriptomics classification and nomenclature of neocortical cell types. *Nat. Neurosci.* *12*, 1456-1468.

Zhang, S., Xu, M., Chang, W.C., Ma, C., Hoang Do, J.P., Jeong, D., Lei, T., Fan, J.L., and Dan, Y. (2016). Organization of long-range inputs and outputs of frontal cortex for top-down control. *Nat. Neurosci.* *19*, 1733-1742.

Zhang, S., Xu, M., Kamigaki, T., Hoang Do, J.P., Chang, W.C., Jenvay, S., Miyamichi, K., Luo, L., and Dan, Y. (2014). Selective attention. Long-range and local circuits for top-down modulation of visual cortex processing. *Science* *345*, 660-665.

Zhuang, J., Wang, Y., Ouellette, N.D., Turschak, E.E., Larsen, R.S., Takasaki, K.T., Daigle, T.L., Tasic, B., Waters, J., Zeng, H., *et al.* (2021). Laminar distribution and arbor density of two functional classes of thalamic inputs to primary visual cortex. *Cell Rep.* *37*, 109826.

Reviewer #3 (Remarks to the Author):

*In this manuscript, Liu et al. employed optogenetic-assisted circuit mapping to investigate top-down inputs from multiple higher-order regions into various anatomically or genetically distinct neurons in the primary visual cortex (V1). They discovered that neurons in the medial part of secondary visual cortex and the anterior cingulate area predominantly activate neurons in layer 6 (L6), whereas the ventrolateral orbital cortex (ORBvl) and the lateral posterior nucleus of the thalamus (LP) primarily target neurons in layer 5 (L5). Within L5, neurons that receive inputs from ORBvl and LP exhibit notable segregation, distinguished by distinct molecular profiles and electrophysiological characteristics. These findings offer valuable insights for those interested in information processing within V1. Nevertheless, this study constitutes an incremental intellectual progression from their previous work published in *Science Advances* in 2021. This manuscript might be more suited to a specialized journal.*

RESPONSE:

The primary visual cortex (V1) is fundamental to our understanding of visual perception. At this site, both bottom-up and top-down inputs are integrated, and this is known to help shape our perception of the visual world (Froudarakis et al., 2019; Glickfeld and Olsen, 2017; Niell and Scanziani, 2021; Seabrook et al., 2017). Thus, deepening understanding of V1's processing mechanisms contribute to basic neuroscience and have potential for clinical applications for treating visual disorders.

Recent research in the field has been intensely focused on elucidating the neuronal circuits and mechanisms underlying visual processing, including studies assessing how different higher-order cortical and thalamic areas modulate V1 activity in response to various behavioral states and needs (Debes and Dragoi, 2023; Fiser et al., 2016; Gilbert and Li, 2013; Glickfeld and Olsen, 2017; Hu et al., 2019; Huda et al., 2020; Kim et al., 2021; Leinweber et al., 2017; Liu et al., 2020; Makino and Komiyama, 2015; Maunsell, 2015; Moore and Zirnsak, 2017; Norman et al., 2021; Steinmetz et al., 2019; Zatka-Haas et al., 2021; Zhang et al., 2014). Our study contributes to this active area of research by revealing how specific top-down inputs modulate V1.

I would like to elaborate further on the conceptual innovation of our current study, which was informed by our study published in *Science Advances* in 2021. In the 2021 study, we quantitatively analyzed CC and TC long-range inputs to cortical inhibitory neurons at different hierarchical stages in the visual network, using rabies-virus tracing combined with optogenetics-assisted electrophysiological recording. While both our previous and current studies present innervation patterns of ACA and LP inputs to V1 inhibitory neurons, our previous work was limited to exploring innervation patterns in L2/3. By contrast, our current study systematically characterized the layer- and cell-type-specific organization of CC (V2M, ACA, ORBvl) and TC (LP) top-down inputs in V1, which revealed insights into the sophisticated mechanisms of

top-down modulation.

Our study deepens understanding about how V1 integrates these top-down inputs with bottom-up signals to modulate visual processing. Interestingly, in contrast to similar innervation properties in superficial layers, the examined CC and TC inputs clearly segregated in deep layers. It is known that deep-layer Pyrs are a heterogeneous population with functionally distinct subnetworks (Harris and Shepherd, 2015; Matho et al., 2021; Tasic et al., 2018; Yao et al., 2021). To further investigate the interactions among these top-down inputs in their most responsive layers, we employed independent optogenetic activation. Together with new experiments added in this revision, our results demonstrate that V2M and ACA inputs converge on L6 Pyrs, in sharp contrast to the parallel processing of LP and ORBv1 inputs in two distinct types of L5 Pyrs. These findings demonstrate distinct processing strategies, with integrated processing of V2M and ACA inputs in L6, as opposed to the parallel processing of LP and ORBv1 inputs in L5.

Given the notable specificity with which L5 Pyrs process LP and ORBv1 inputs, we narrowed our focus on the properties of Pyr_{-LP} and Pyr_{-ORBv1} neurons. After an in-depth characterization of their electrophysiological properties, gene expression profiles, and axon innervation patterns, we concluded that they likely represent Pyr subtypes: IT and PT neurons, thus establishing a link between the parallel processing of LP and ORBv1 information in V1 L5 and IT and PT subnetworks. Finally, it bears emphasis that our discovery of mutual inhibition between PT and IT subnetworks, driven by ORBv1 and LP inputs, establishes that dynamic gating occurs in V1, thus adding a new dimension to the current knowledge of visual processing. The insights gained from our study open new avenues for future research, particularly for determining the functional consequences of these interactions in V1 and for exploring their implications for visual perception and behavior.

References

- Debes, S.R., and Dragoi, V. (2023). Suppressing feedback signals to visual cortex abolishes attentional modulation. *Science* *379*, 468-473.
- Fiser, A., Mahringer, D., Oyibo, H.K., Petersen, A.V., Leinweber, M., and Keller, G.B. (2016). Experience-dependent spatial expectations in mouse visual cortex. *Nat. Neurosci.* *19*, 1658-1664.
- Froudarakis, E., Fahey, P.G., Reimer, J., Smirnakis, S.M., Tehovnik, E.J., and Tolias, A.S. (2019). The Visual Cortex in Context. *Annu. Rev. Vis. Sci.* *5*, 317-339.
- Gilbert, C.D., and Li, W. (2013). Top-down influences on visual processing. *Nat. Rev. Neurosci.* *14*, 350-363.
- Glickfeld, L.L., and Olsen, S.R. (2017). Higher-Order Areas of the Mouse Visual Cortex. *Annu. Rev. Vis. Sci.* *3*, 251-273.
- Harris, K.D., and Shepherd, G.M. (2015). The neocortical circuit: themes and variations. *Nat. Neurosci.* *18*, 170-181.
- Hu, F., Kamigaki, T., Zhang, Z., Zhang, S., Dan, U., and Dan, Y. (2019). Prefrontal Corticotectal Neurons Enhance Visual Processing through the Superior Colliculus and Pulvinar Thalamus. *Neuron* *104*, 1141-1152.
- Huda, R., Sipe, G.O., Breton-Provencher, V., Cruz, K.G., Pho, G.N., Adam, E., Gunter, L.M., Sullins, A., Wickersham, I.R., and Sur, M. (2020). Distinct prefrontal top-down circuits differentially modulate sensorimotor behavior. *Nat. Commun.* *11*, 6007.
- Kim, J.H., Ma, D.H., Jung, E., Choi, I., and Lee, S.H. (2021). Gated feedforward inhibition in the frontal cortex releases goal-directed action. *Nat. Neurosci.* *24*, 1452-1464.
- Leinweber, M., Ward, D.R., Sobczak, J.M., Attinger, A., and Keller, G.B. (2017). A Sensorimotor Circuit in Mouse Cortex for Visual Flow Predictions. *Neuron* *95*, 1420-1432.
- Liu, D., Deng, J., Zhang, Z., Zhang, Z.Y., Sun, Y.G., Yang, T., and Yao, H. (2020). Orbitofrontal control of visual cortex gain promotes visual associative learning. *Nat. Commun.* *11*, 2784.
- Ma, G., Liu, Y., Wang, L., Xiao, Z., Song, K., Wang, Y., Peng, W., Liu, X., Wang, Z., Jin, S., *et al.* (2021). Hierarchy in sensory processing reflected by innervation balance on cortical interneurons. *Sci. Adv.* *7*, abf5676.
- Makino, H., and Komiyama, T. (2015). Learning enhances the relative impact of top-down processing in the visual cortex. *Nat. Neurosci.* *18*, 1116-1122.
- Matho, K.S., Huilgol, D., Galbavy, W., He, M., Kim, G., An, X., Lu, J., Wu, P., Di Bella, D.J., Shetty, A.S., *et al.* (2021). Genetic dissection of the glutamatergic neuron system in cerebral cortex. *Nature* *598*, 182-187.
- Maunsell, J.H.R. (2015). Neuronal Mechanisms of Visual Attention. *Annu. Rev. Vis. Sci.* *1*, 373-391.
- Moore, T., and Zirnsak, M. (2017). Neural Mechanisms of Selective Visual Attention. *Annu. Rev. Psychol.* *68*, 47-72.
- Niell, C.M., and Scanziani, M. (2021). How Cortical Circuits Implement Cortical Computations: Mouse Visual Cortex as a Model. *Annu. Rev. Neurosci.* *44*, 517-546.
- Norman, K.J., Riceberg, J.S., Koike, H., Bateh, J., McCraney, S.E., Caro, K., Kato, D., Liang, A., Yamamuro, K., Flanigan, M.E., *et al.* (2021). Post-error recruitment of frontal sensory cortical projections promotes attention in mice. *Neuron* *109*, 1202-1213.
- Seabrook, T.A., Burbridge, T.J., Crair, M.C., and Huberman, A.D. (2017). Architecture, Function, and Assembly of the Mouse Visual System. *Annu. Rev. Neurosci.* *40*, 499-538.

Steinmetz, N.A., Zatka-Haas, P., Carandini, M., and Harris, K.D. (2019). Distributed coding of choice, action and engagement across the mouse brain. *Nature* *576*, 266-273.

Tasic, B., Yao, Z., Graybiel, L.T., Smith, K.A., Nguyen, T.N., Bertagnolli, D., Goldy, J., Garren, E., Economo, M.N., Viswanathan, S., *et al.* (2018). Shared and distinct transcriptomic cell types across neocortical areas. *Nature* *563*, 72-78.

Yao, Z., van Velthoven, C.T.J., Nguyen, T.N., Goldy, J., Sedenio-Cortes, A.E., Baftizadeh, F., Bertagnolli, D., Casper, T., Chiang, M., Crichton, K., *et al.* (2021). A taxonomy of transcriptomic cell types across the isocortex and hippocampal formation. *Cell* *184*, 3222-3241.

Zatka-Haas, P., Steinmetz, N.A., Carandini, M., and Harris, K.D. (2021). Sensory coding and the causal impact of mouse cortex in a visual decision. *Elife* *10*, e63163.

Zhang, S., Xu, M., Kamigaki, T., Hoang Do, J.P., Chang, W.C., Jenvay, S., Miyamichi, K., Luo, L., and Dan, Y. (2014). Selective attention. Long-range and local circuits for top-down modulation of visual cortex processing. *Science* *345*, 660-665.

Reviewer #4 (Remarks to the Author):

In this manuscript, Liu, Zhang, and colleagues present valuable findings on the synaptic organization of corticocortical and thalamocortical top-down inputs onto mouse V1 neurons. The authors characterized the connectivity and strengths of different synaptic inputs and made an exciting discovery of two parallel processing pathways from ORBv1 to V1 L5 to subcortical areas and LP to V1 L5 to cortical areas. The evidence supporting the claims of the authors is overall solid, but requires additional validation (see below). Overall, this is an excellent study with some weaknesses that can be addressed.

1. For the experiments in Figure 1, the authors should clarify if neurons in L2/3, L4, L5, and L6 were recorded simultaneously in response to the stimulation of presynaptic terminals. This seems not the case based on very different numbers of recorded neurons from different layers. These experiments depend on the viral expression of ChR2 in different cortical and thalamic areas, which varies between mice and slices and can contribute to the difference between different recordings. If neurons in different layers were not recorded simultaneously, then the authors should demonstrate how such variabilities did not cause the difference observed between layers for each of the input profile. In other words, the authors should validate their findings and rule out this possibility.

RESPONSE:

We thank the reviewer for pointing out this issue. Briefly, we recorded neurons simultaneously in L2/3, L4, L5, and L6 in at least 5 mice in each group (V2M, ACA, ORBv1, and LP inputs). For the majority of recorded animals, we recorded neurons simultaneously in more than 3 layers. Based on reviewer's guidance, we have clarified this issue in the revised methods. The revised text reads as follows: "For the experiments in Fig. 1 and Fig. 2, we recorded neurons simultaneously in each layer in at least 5 mice in each group. For the majority of recorded animals, we recorded neurons simultaneously in three or more layers."

Moreover, following the reviewer's guidance in the 4th comment, we reanalyzed our data and generated plots of the normalized EPSP amplitude for each top-down input (see new Extended Data Fig. 3). For V2M and ACA inputs, we normalized the EPSP amplitude of each recorded neuron to the averaged EPSP amplitude of L6 Pyrs in the same slice. For ORBv1 and LP inputs, considering that the Pyrs in their most responsive layer (L5) can be divided into distinct subtypes according to their responses to these inputs (as shown in our revised main Fig. 4), we normalized the EPSP amplitude against $\text{Pyr}_{\leftarrow\text{ORBv1}}$ and $\text{Pyr}_{\leftarrow\text{LP}}$ neurons, respectively. Results obtained with this normalization approach are consistent with the previously detected differences in input strengths across layers and cell types as reported in the originally submitted manuscript. CC top-down inputs (V2M, ACA, ORBv1) and TC top-down inputs (LP) inputs are partially overlapped in L2/3 with relatively weak input strength in L2/3 Pyrs. These inputs then

bypass L4 Pyrs before segregating in L5 and L6: V2M and ACA inputs strongly activate L6 Pyrs, while ORBv1 and LP inputs strongly activate L5 Pyrs.

“Extended Data Fig. 3 | Layer specificity of CC and TC top-down inputs based on normalized EPSP amplitude.

a, Normalized EPSP amplitudes evoked by optogenetic activation of V2M inputs in Pyrs across the indicated layers. Left, box plot showing distribution of EPSP amplitude from Pyrs in the indicated layers. Edges, 25th and 75th percentiles; central line, median; whiskers, 1.5× the interquartile range of the edges. Circles are outliers, defined as values more than 3 times the interquartile range (IQR) from the median. V2M inputs activated the Pyrs in L2/3 and L6, with the strongest input strength in L6 (significant differences were observed across layers $F(3,199) = 20.7$, $P = 1 \times 10^{-11}$, one-way ANOVA; L6 vs. other layers, $P < 0.002$, Tukey’s post hoc test). Right, matrix of normalized EPSP amplitude across layers of V1 for V2M inputs. **b**, Similar to **a**, but for ACA inputs. ACA inputs activated the Pyrs in L2/3 and L6, with stronger input strength in L6 (significant differences were observed across layers, $F(3,152) = 10.2$, $P = 4 \times 10^{-6}$, one-way ANOVA; L6 vs. L4 and L5, $P < 0.004$; L6 vs. L2/3, $P = 0.87$, Tukey’s post hoc test). **c**, Similar to **a**, but for ORBv1 inputs. ORBv1 inputs only activated the Pyrs in L5 (significant differences were observed across layers, $F(3,101) = 13.4$, $P = 2 \times 10^{-7}$, one-way ANOVA; L5 vs. other layers, $P < 5 \times 10^{-4}$, Tukey’s post hoc test). **d**, Similar to **a**, but for LP inputs. LP inputs activated the Pyrs in L2/3, L5, and L6, with the strongest input strength in L5

(significant differences were observed across layers, $F(3,204) = 15.8$, $P = 3 \times 10^{-9}$, one-way ANOVA; L5 vs. other layers, $P < 2 \times 10^{-5}$, Tukey's post hoc test). The number of neurons in each group is indicated by the numbers displayed in the figure. The EPSP amplitudes of V2M and ACA inputs were normalized to the averaged L6 Pyr EPSP amplitude recorded the same day. Given the observed heterogeneity in L5 Pyr subtypes responding to ORBv1 and LP inputs, we normalized the EPSP amplitudes of ORBv1 and LP inputs to the average for Pyr \leftarrow ORBv1 and for Pyr \leftarrow LP neurons. The data included are those with Pyrs recorded from the most responsive layer in the same brain slice.”

2. Related to Point 1, Extended Data Fig 2 shows the normalized EPSP amplitudes to L6 Pyrs recorded on the same brain slice. It is unclear how this normalization was done because one would expect that all the values from L6 would be 1.

RESPONSE:

The original Extended Data Fig. 2 has been replaced with revised Extended Data Fig. 3 in our revised manuscript (kindly see our response to the 1st comment). To clarify this issue, I would like to show the original Extended Data Fig.2 here.

“Original Extended Data Fig. 2 | Layer specificity of V2M inputs derived from raw and normalized EPSP amplitude. **a**, Raw EPSP amplitudes evoked by V2M inputs in V1 Pyrs across different layers. L6 Pyrs received the strongest inputs. L2/3, n = 62 neurons; L4, n = 11 neurons; L5, n = 36 neuron; L6, n = 84 neurons. **b**, Similar to a, but for normalizing EPSP amplitude, which was normalized to L6 Pyrs recorded on the same brain slice. Normalized input strength data also showed that L6 Pyrs received the strongest inputs. Data are presented as the mean \pm SEM.”

Here, the averaged Norm. EPSP amplitude of L6 is 1. The number of L6 Pyrs recorded each day range from 2 to 12. And the EPSP amplitude of each recorded neuron, including those recorded in L6, was normalized to the averaged EPSP amplitude of all L6 Pyrs recorded in the same brain slice. We observed some variations in the normalized EPSP amplitudes within the L6 Pyr population, consistent with previous reports (Ji et al., 2016; Naskar et al., 2021).

3. *The CaMKII α promoter is not specific to excitatory neurons (see <https://www.eneuro.org/content/10/4/ENEURO.0070-23.2023>). It's not a problem to use it for ACA, ORBvl, and LP, but it could be problematic for V2M because some interneuron axons may be present in the V1, which can affect the size of EPSPs. The authors should test this possibility.*

RESPONSE:

We appreciate the reviewer highlighting the specificity concern regarding the use of the CaMKII α promoter. To address this, we conducted additional experiments to determine if the responses from V2M inputs recorded in our study were influenced by inhibitory neurons expressing ChR2 under the CaMKII α promoter in the V2M area. Specifically, we compared the EPSP amplitudes in V1 Pyrs evoked by optogenetic activation of V2M inputs both with and without the application of a GABA α receptor antagonist (bicuculline, 20 μ M) in the presence of TTX and 4-AP (blocking local neuron spikes). The bath application of bicuculline did not change the EPSP amplitude evoked by V2M input activation in either L2/3 or L6 Pyrs in V1 (Reviewer Fig. 7). These findings suggest that the recorded area in our experiments was sufficiently distant (500-1000 μ m away) from the V1/V2M border to preclude influence from V2M inhibitory neurons.

Reviewer Fig. 7 | Antagonist of GABA_A receptors has no effect on the EPSP amplitude of V2M inputs on V1 Pyrs. Top, example EPSPs recorded with (black) and without (blue) bicuculline. Bottom, bath application of bicuculline (a GABA_A receptor antagonist, 20 μM), in the presence of TTX and 4-AP (blocking local neuron spikes), caused no significant changes in the EPSP amplitudes of V2M inputs on both L2/3 and L6 Pyrs in V1. L2/3 Pyrs, n = 4 mice, 10 slices, 10 neurons, P = 0.28, Wilcoxon signed-rank test. L6 Pyrs, n = 4 mice, 17 slices, 17 neurons, P = 0.79. Scale bars, 5 mV, 25 ms. Data are presented as the mean ± SEM.

4. Similar to Point 1, for the experiments in Figure 2, the authors also need to demonstrate that the variability of viral expression of ChR2 does not cause the observed difference between layers and interneuron types. One possibility is to use an independent neuronal type as the reference for normalization.

RESPONSE:

We appreciate reviewer's guidance about using an independent neuronal type as the reference for normalization. Previous studies on inhibitory neurons have used Pyrs as a reference for normalization (Combe and Gasparini, 2021; Hu et al., 2019; Ibrahim et al., 2016; Ji et al., 2016; Ma et al., 2021; Naskar et al., 2021; Zhang et al., 2014). Following the reviewer's guidance, we reanalyzed our data and generated plots of the normalized EPSP amplitude for each top-down input (new Extended Data Fig. 4). For V2M and ACA inputs, we normalized the EPSP amplitude of each recorded neuron to the averaged EPSP amplitude of L6 Pyrs in the same slice. For ORBvl and LP inputs—considering that the Pyrs in their most responsive layer (L5) can be divided into distinct subtypes according to their responses to these inputs (as shown in our revised main Fig. 4)—we normalized the EPSP amplitude against Pyr_{T←ORBvl} and Pyr_{T←LP} neurons,

respectively. Results obtained with this normalization approach are consistent with the previously detected differences in input strengths across layers and cell types as reported in the originally submitted manuscript.

We have revised our results section accordingly. The revised text reads as follows (P. 7, Lines 5-7): “Note that the layer and cell-type specificity of these top-down inputs is evident based on both the raw and normalized input strengths (Extended Data Figs. 3 and 4, Supplementary Tables 4 and 5).”

We would like to take this opportunity to clarify our methodology related to these experiments. In alignment with established practices in mouse studies that assess raw input strengths across various neuron types, cortical layers, and developmental stages (Ibrahim et al., 2021; Kim et al., 2018; Nagai et al., 2019; Paniccia et al., 2023), we carefully controlled several parameters to minimize variations caused by ChR2 expression. These included standardizing the titer, injection volume, and expression time of the AAV vector in each experimental series, ensuring consistent ChR2 expression levels. Moreover, we consistently applied the same optical stimulation parameters for ChR2 activation across all experiments presented in Figs. 1 and 2. The top-down inputs we examined preferentially activate various subpopulations of Pyrs across different layers, necessitating layer- and subpopulation-specific reference points for each top-down input. While normalized data facilitates comparison across different types of inhibitory neurons responding to the same inputs, this approach loses some information regarding the potential differences among various inputs. Therefore, we present both raw and normalized data in our revised manuscript as shown in Figs. 1 and 2, and Extended Data Figs. 3 and 4.

“Extended Data Fig. 4 | Normalized input strengths of CC and TC top-down inputs in four types of V1 inhibitory neurons across different layers. a, Distribution of normalized EPSP amplitude from inhibitory neurons in different layers. Box plots indicate the median and the respective quartiles. Circles are outliers, defined as values above 3 IQR from the median. **b**, Matrix of normalized EPSP amplitude in different

types of inhibitory neurons across layers of V1 for V2M inputs. **c-d**, similar to **a-b**, but for ACA inputs. **e-f**, similar to **a-b**, but for ORBv1 inputs. **g-h**, similar to **a-b**, but for LP inputs. The number of neurons in each group is indicated by the numbers displayed in the figure. The EPSP amplitudes of V2M and ACA inputs were normalized to the averaged L6 Pyr EPSP amplitude recorded the same day. Given the observed heterogeneity in L5 Pyr subtypes responding to ORBv1 and LP inputs, we normalized the EPSP amplitudes of ORBv1 and LP inputs to the average for Pyr \leftarrow ORBv1 and for Pyr \leftarrow LP neurons. The data included are those with Pyrs recorded from the most responsive layer in the same brain slice.”

5. *The authors should also clarify if the experiments in Figure 7a/e were done on two neurons simultaneously in the same slice, because the schematics imply that.*

RESPONSE:

Following the reviewer’s guidance, we have revised the legend to original Fig. 7 (revised Fig. 8) to clarify our methodology. We recorded from one neuron at a time, randomly sampling from L5 Pyrs. The revised text is as follows “**a**, Schematic of the slice experiment, illustrating the whole-cell recording conducted on individual Pyr \leftarrow LP (red) or Pyr \leftarrow ORBv1 (green) neurons (one at a time), coupled with optogenetic activation of LP inputs.”

6. *The statement “...LP inputs induced significantly stronger inhibition in Pyr \leftarrow ORBv1 neurons compared to Pyr \leftarrow LP neurons (shown by the $I/(I+E)$ ratio, Fig. 7c)...” is not accurate because this ratio does not compare the inhibition directly. Similarly, the statement “...ORBv1 inputs induced significantly stronger inhibition of Pyr \leftarrow LP neurons compared to Pyr \leftarrow ORBv1 neurons (Fig. 6g, it should be 7g)...” is also not accurate. If the authors recorded the two neurons simultaneously, then the IPSCs can be directly compared.*

RESPONSE:

We thank the reviewer for pointing out this mistake. As we have stated in the response to 5th comment, we recorded from one neuron at a time, randomly sampling from L5 Pyrs. Our original findings, as indicated by the $I/(I+E)$ ratio in the original Fig. 7c (revised Fig. 8c), suggest that LP inputs result in a relatively higher proportion of inhibition in Pyr \leftarrow ORBv1 neurons compared to Pyr \leftarrow LP neurons. This implies that, in response to LP inputs, there is a tendency for increased inhibitory influence relative to excitatory influence in Pyr \leftarrow ORBv1 neurons. Conversely, the $I/(I+E)$ ratio observed for ORBv1 inputs (Fig. 8g) suggests a relatively higher proportion of inhibition in Pyr \leftarrow LP

neurons compared to Pyr \leftarrow ORBvl neurons. However, it is important to note that these ratios do not directly compare the absolute levels of inhibition among these neuron types. Instead, they represent the ratio of inhibitory to total synaptic inputs (both inhibitory and excitatory) within each neuron type, in response to specific inputs.

We have now revised the manuscript accordingly, and the revised text reads as follows (P. 13, Lines 11-20, and 32, as well as P. 14, Lines 1-8):

“Activation of LP inputs induced both EPSCs and IPSCs in Pyr \leftarrow LP neurons (Fig. 8b, left). EPSCs had short onset latencies (4.5 ± 0.3 ms, mean \pm SEM, $n = 20$ neurons), suggesting monosynaptic excitatory inputs. Notably, the latencies of IPSCs were significantly longer than for EPSCs (10.4 ± 1.1 ms, $P = 6 \times 10^{-6}$, Wilcoxon sign-rank test), suggesting multisynaptic inhibition. We observed a net inhibition in Pyr \leftarrow ORBvl neurons induced by LP inputs (IPSC, latency 10.9 ± 0.6 ms, $n = 26$ neurons, Fig. 8b, right). LP inputs drive significantly stronger inhibition relative to excitation in Pyr \leftarrow ORBvl neurons than in Pyr \leftarrow LP neurons, as shown by the I/(I+E) ratio (Fig. 8c, Extended Data Fig. 11). This suggests that LP inputs drive simultaneous activation of Pyr \leftarrow LP neurons and inhibition of Pyr \leftarrow ORBvl neurons.”

“We next measured responses in Pyr \leftarrow LP and Pyr \leftarrow ORBvl neurons induced by ORBvl inputs (Fig. 8e, Extended Data Fig. 11). Activation of ORBvl inputs induced monosynaptic EPSCs and multisynaptic IPSCs in Pyr \leftarrow ORBvl neurons (EPSC latency, 3.7 ± 0.3 ms; IPSC latency, 6.6 ± 0.4 ms, mean \pm SEM; $n = 20$ neurons, Fig. 8f). In Pyr \leftarrow LP neurons, only multisynaptic IPSCs were observed (IPSC latency, 7.2 ± 0.3 ms, mean \pm SEM, $n = 19$ neurons). In contrast to LP inputs, ORBvl inputs drive significantly stronger inhibition relative to excitation in Pyr \leftarrow LP neurons than in Pyr \leftarrow ORBvl neurons (Fig. 8g), indicating that ORBvl inputs may simultaneously drive activation of Pyr \leftarrow ORBvl neurons and inhibition of Pyr \leftarrow LP neurons.”

Additionally, we have also revised related figure caption of revised Fig.8. The revised text reads as follows:

“**c**, Contribution of inhibitory inputs to total synaptic inputs, measured as $IPSC_{charge}/(IPSC_{charge}+EPSC_{charge})$. LP inputs drive significantly stronger inhibition relative to excitation in Pyr \leftarrow ORBvl neurons than in Pyr \leftarrow LP neurons. $P = 5 \times 10^{-7}$, Wilcoxon rank-sum test. $n = 8$ mice, 9 slices. **d**, Normalized EPSP amplitudes in different cell types evoked by LP inputs (normalized to Pyr \leftarrow LP neurons in L5). **e-h**, similar to **a-d**, but for ORBvl inputs. **g**, ORBvl inputs drive significantly stronger inhibition

relative to excitation in Pyr←LP neurons than in Pyr←ORBv1 neurons. $P = 1 \times 10^{-7}$. n = 4 mice, 5 slices.”

7. Based on the findings in Figure 7, one would expect that activation of LP inputs would cause hyperpolarization of Pyr←ORBv1 neurons, which explains the negative values in Figure 1s L5 neurons. The authors should mention this. This applies to ORBv1 inputs and Figure 1n.

RESPONSE:

We thank the reviewer for pointing out this problem. In our original Figs. 1 and 2, we show both box plots and violin plots. The box plots clearly show that the lower limit of our data is zero, but the violin plots extend into the negative values, likely due to the kernel density estimation algorithm extending beyond the actual range of data to fit a smooth curve. To avoid the misleading information in violin plots, we have revised the original Figs. 1 and 2, removing violin plots and retained only box blots.

We would like to take this opportunity to clarify our methodology related to the experiments in Figs. 1 and 2. In these experiments, we used TTX and 4-AP (bath application), blocking local neuron spikes (Petreanu et al., 2009), to measure the monosynaptic inputs from various sources to V1 neurons. Therefore, despite inhibitory neurons receiving excitatory inputs from LP or ORBv1, the action potentials in these neurons are blocked by TTX. Consequently, local inhibitory neurons should not cause hyperpolarization in Pys.

“Fig. 1 | Systematic characterization of the input strengths of CC and TC top-down inputs in V1 Pyrs across different layers. a, Schematic of the slice experiment, with whole-cell recording of V1 Pyrs and optogenetic activation of V2M inputs. **b,** Injection sites in the V2M. Scale bar, 500 μ m. Arrowhead, AAV injection site. Blue, DAPI staining. Inset, location of the coronal section. **c,** Distribution of V2M axons in V1. Left, fluorescence image showing V2M axons (green) in V1. Scale bar, 50 μ m. Right, normalized green fluorescence intensity from L1 to L6 (normalized by the peak intensity in V1). **d,** Monosynaptic EPSP amplitudes in Pyrs across different layers. Left, monosynaptic EPSPs from one example Pyr in each layer with TTX and 4-AP treatment to block local neuron spiking (gray, raw traces; purple, averaged traces). Blue dots, 5-ms blue light stimulation (474 nm, 3.5 mW). Scale bars, 50 ms, 5 mV. Right, box plot showing distribution of EPSP amplitude from Pyrs in the indicated layers. Edges, 25th and 75th percentiles; central line, median; whiskers, 1.5 \times the interquartile range of the edges. Black circles are

outliers, defined as values more than 3 times the interquartile range (IQR) from the median. V2M inputs activated the Pyrs in L2/3 and L6, with the strongest input strength in L6 (significant differences were observed across layers, $F(3,199) = 17.8$, $P = 3 \times 10^{-10}$, one-way ANOVA; L6 vs. other layers, $P < 0.004$, Tukey's post hoc test). **e**, Matrix of EPSP amplitude (left) and normalized axon intensity (right) across layers of V1 for V2M inputs. **f-j**, similar to **a-e**, but for ACA inputs. **i**, Scale bars, 50 ms, 5 mV. ACA inputs activated the Pyrs in L2/3 and L6, with stronger input strength in L6 (significant differences were observed across layers, $F(3,160) = 8.1$, $P = 5 \times 10^{-5}$, one-way ANOVA; L6 vs. L4 and L5, $P < 0.007$; L6 vs. L2/3, $P = 0.23$, Tukey's post hoc test). L2/3, 4.8 ± 1.2 mV; L6, 7.6 ± 1 mV. **k-o**, similar to **a-e**, but for ORBvl inputs. **n**, Scale bars, 50 ms, 5 mV. ORBvl inputs only activated the Pyrs in L5 (significant differences were observed across layers, $F(3,180) = 10.8$, $P = 1 \times 10^{-6}$, one-way ANOVA; L5 vs. other layers, $P < 0.02$, Tukey's post hoc test). **p-t**, similar to **a-e**, but for LP inputs. **s**, Scale bars, 50 ms, 5 mV. LP inputs activated the Pyrs in L2/3, L5, and L6, with the strongest input strength in L5 (significant differences were observed across layers, $F(3,301) = 17.1$, $P = 3 \times 10^{-10}$, one-way ANOVA; L5 vs. other layers, $P < 7 \times 10^{-5}$, Tukey's post hoc test). The number of neurons in each group is indicated by the numbers displayed in the figure. See Extended Data Table 1 and 2 for detailed input strengths and ANOVA parameters.”

‘Fig. 2 | Input strengths of CC and TC top-down inputs in four types of V1 inhibitory neurons across different layers. a, Schematic of the slice experiment, with whole-cell recording of V1 inhibitory neurons and optogenetic activation of V2M inputs. b, Monosynaptic EPSPs from

one example inhibitory neuron (L1-IN, VIP+, PV+, and SST+ neurons) in each layer with TTX and 4-AP treatment to block local neuron spiking (gray traces, raw traces; colored traces, averaged traces). Blue dots, 5-ms blue light stimulation (474 nm, 3.5 mW). Scale bars, 50 ms, 10 mV. **c**, Distribution of EPSP amplitude from inhibitory neurons in different layers. Box plots indicate the median and the respective quartiles. Black dots are outliers, defined as values above 3 IQR from the median. V2M inputs strongly activated the L1-INs in L1, and VIP+ and PV+ neurons in L2/3 and L6. **d**, Matrix of EPSP amplitude in different types of inhibitory neurons across layers of V1 for V2M inputs. **e-h**, similar to **a-d**, but for ACA inputs. **f**, Scale bars, 50 ms, 10 mV. **g**, ACA inputs strongly activated the L1-INs in L1, VIP+ neurons in L2/3 and L6, and PV+ and SST+ neurons in L6. **i-l**, similar to **a-d**, but for ORBvl inputs. **j**, Scale bars, 50 ms, 10 mV. **k**, ORBvl inputs were weak in V1 inhibitory neurons across layers. **m-p**, similar to **a-d**, but for LP inputs. **n**, Scale bars, 50 ms, 10 mV. **o**, LP inputs strongly activated the L1-INs in L1, VIP+ neurons from L2/3 to L6, and PV+ neurons in L5. **q**, Summary of major connections of V2M and ACA inputs in V1 local circuits. V2M inputs, purple lines; ACA inputs, green lines. Line width represents the amplitude of synaptic input. Only the connections with an averaged EPSP amplitude of >3 mV were included. **r**, similar to **q**, but for ORBvl (brown) and LP (blue) inputs. The number of neurons in each group is indicated by the numbers displayed in the figure. See Extended Data Table 3 for detailed input strengths.”

8. *It would be helpful if the authors provide the numbers of mice and slices used in each experiment in addition to neurons.*

RESPONSE:

Based on the reviewer’s guidance, we have added the numbers of mice and slices used in each experiment. For the experiments shown in Figs.1 and 2, as well as in Extended Data Figs. 3 and 4, the number of mice and slices have now been added to revised Supplementary Tables 1, 3, 4, and 5. For other experiments, we have added this information in the figure captions. The revised text reads as follows:

“**Fig.3 | g**, Normalized EPSC charge evoked by activation of V2M and ACA inputs in Pyr_{-both} (top) and Pyr_{-V2M} neurons (bottom). The circles indicate the means. n = 5 mice, 7 slices. Pyr_{-both}, n = 37 neurons; Pyr_{-V2M}, n = 26 neurons.”

“**Fig.4 | g**, Normalized EPSC charge evoked by activation of LP and ORBvl inputs in Pyr_{-LP} (top) and Pyr_{-ORBvl} neurons (bottom). The circles indicate the means. n = 12 mice, 14 slices. Pyr_{-LP}, n = 28 neurons; Pyr_{-ORBvl}, n = 35 neurons.”

“**Fig.6 | c**, All of the recorded Pyr_{-LP} neurons exhibited regular

spiking (RS), whereas 34% of the Pyr \leftarrow ORBvl showed RS and 66% exhibited burst spiking (BS) (n = 20 mice, 20 slices; Pyr \leftarrow LP, n = 47 neurons; Pyr \leftarrow ORBvl, n = 44 neurons; P = 6×10^{-10} , Kolmogorov-Smirnov test).”

“**Fig.7 | c**, Proportion of Pyr \leftarrow LP and Pyr \leftarrow ORBvl neurons in labeled projecting Pyrs. The number of neurons in each group is indicated by the numbers displayed in the figure. V2M, n = 4 mice, 5 slices; RSP, n = 3 mice, 5 slices; ORBvl, n = 3 mice, 3 slices; Ipsi-DMS, n = 4 mice, 5 slices; Contra-DMS, n = 3 mice, 5 slices; Pons, n = 3 mice, 3 slices; SC, n = 3 mice, 3 slices; LP, n = 5 mice, 5 slices.”

“**Fig.8 | c**, Contribution of inhibitory inputs to total synaptic inputs, measured as $IPSC_{charge}/(IPSC_{charge}+EPSC_{charge})$. LP inputs drive significantly stronger inhibition relative to excitation in Pyr \leftarrow ORBvl neurons than in Pyr \leftarrow LP neurons. P = 5×10^{-7} , Wilcoxon rank-sum test. n = 8 mice, 9 slices. **g**, ORBvl inputs drive significantly stronger inhibition relative to excitation in Pyr \leftarrow LP neurons than in Pyr \leftarrow ORBvl neurons. P = 1×10^{-7} . n = 4 mice, 5 slices.”

“**Extended Data Fig. 5 | b**, TTX and 4-AP had no effect on I_h slope of Pyr \leftarrow LP (red) and Pyr \leftarrow ORBvl (green) neurons. n = 3 mice, 3 slices. Pyr \leftarrow LP, n = 13 neurons, P = 0.12; Pyr \leftarrow ORBvl, n = 14 neurons, P = 0.07; Wilcoxon signed-rank test.”

“**Extended Data Fig. 10 | b**, Performance of the SVM Classifier(I_h) in distinguishing Pyr \leftarrow LP from Pyr \leftarrow ORBvl neurons in the dataset of RSP-projecting Pyrs (the neuron types were initially defined based on their input patterns as determined upon independent optogenetic activation of LP and ORBvl inputs). Dashed line, decision boundary for classifying Pyr \leftarrow LP and Pyr \leftarrow ORBvl neurons. Red stars, input-pattern-classified Pyr \leftarrow LP neurons. n = 3 mice, 3 slices, 9 neurons. **c**, Prediction accuracy. **d-f**, Similar to a-c, but for the performance of SVM Classifier(I_h) on ipsi-DMS-projecting Pyrs. Red and green stars, input-pattern-classified Pyr \leftarrow LP and Pyr \leftarrow ORBvl neurons. n = 3 mice, 3 slices, 19 neurons. **j-i**, Similar to a-c, but for the performance of SVM Classifier(I_h) on SC-projecting Pyrs. Green stars, input-pattern-classified Pyr \leftarrow ORBvl neurons. n = 4 mice, 4 slices, 14 neurons.”

“**Extended Data Fig. 12 | c**, Normalized PV-IPSC amplitudes in Pyr \leftarrow ORBvl neurons were significantly larger than in Pyr \leftarrow LP neurons. n = 3 mice, 4 slices. Pyr \leftarrow LP, n = 24 neurons, Pyr \leftarrow ORBvl, n = 21 neurons. **d-f**, Similar to a-c, but for SST+ neuron-induced IPSCs. n = 5 mice, 7 slices. Pyr \leftarrow LP, n = 19 neurons, Pyr \leftarrow ORBvl, n = 35 neurons. **g-i**, Similar to a-c, but for VIP+ neuron-induced IPSCs. n = 4 mice, 5 slices. Pyr \leftarrow LP, n = 16 neurons, Pyr \leftarrow ORBvl, n = 13 neurons.”

9. *The statistical methods should be justified. For example, in Figure 1, t-test is not appropriate and one-way ANOVA should be used to compare different layers.*

RESPONSE:

We have re-examined the statistical analysis methods and corrected previous errors. And all the detailed information for ANOVA, including the F ratio, the associated degrees of freedom, and the exact associated P value for ANOVA are now provided in Supplementary Table 2.

In Fig. 1, we have replaced the previously used t-test with a one-way ANOVA and Tukey's post hoc test to compare the raw EPSP amplitudes among different layers, and have corrected the figure legends accordingly. The results of corrected statistics support our conclusions in the originally submitted version.

In revised Extended Data Fig. 3, we used a one-way ANOVA and Tukey's post hoc test to compare the normalized EPSP amplitudes among different layers, leading to a similar conclusion as for Fig. 1.

Kindly see the corrections to the statistical analyses in Figure 1 and the revised Extended Data Fig. 3 in our responses to the 7th and 1st comments, respectively.

10. *The authors may consider additional motivations for this study. For example, more rationales for choosing these particular corticocortical and thalamocortical top-down inputs would be helpful to the readers.*

RESPONSE:

We thank the reviewer for pointing out this issue. Combining Reviewer #1's and your suggestions, we have revised our manuscript to better clarify the motivation behind the various experiments. To justify our specific focus on the LP and ORBv1 inputs in the latter half of our manuscript, we have included a new experiment employing independent optogenetic activation of V2M and ACA inputs. This experiment demonstrates that V2M and ACA inputs converge on L6 Pyrs, in sharp contrast to the parallel processing of LP and ORBv1 inputs in two distinct types of L5 Pyrs. Given the notable specificity with which L5 Pyrs process LP and ORBv1 inputs, we narrowed our focus to exploring the properties of $\text{Pyr}_{\leftarrow\text{LP}}$ and $\text{Pyr}_{\leftarrow\text{ORBv1}}$ neurons. In addition to the new experiment, we have added a new main figure (new Figure 3) and have made significant revisions to the original abstract, introduction, and discussion, as well as to one subsection in the results.

We have revised the original abstract to include the results from the new experiment testing the interaction between V2M and ACA inputs in L6. The revised abstract reads as follows:

“Unified visual perception relies on the integration of bottom-up and top-down inputs in the primary visual cortex (V1), yet the organization of top-down inputs in V1 remains unclear. Here, we used optogenetics-assisted circuit mapping to identify how multiple top-down inputs from higher-order cortical and thalamic areas engage excitatory and inhibitory neurons in V1. Top-down inputs partially overlap in superficial layers, bypass layer 4 (L4), and clearly segregate upon reaching deep layers. Specifically, inputs from the medial secondary visual cortex (V2M) and anterior cingulate cortex (ACA) preferentially activate L6 neurons, while inputs from the ventrolateral orbitofrontal cortex (ORBvl) and lateral posterior thalamic nucleus (LP) activate L5 neurons. We also found that V2M and ACA inputs converge on L6 Pyrs, whereas ORBvl and LP inputs are processed in parallel by two types of L5 Pyrs: Pyr-ORBvl and Pyr-LP neurons, each characterized by specific electrophysiological properties and gene expression profiles. Retrograde mapping revealed that Pyr-ORBvl neurons preferentially innervate subcortical areas and Pyr-LP neurons innervate cortical areas, indicating parallel processing of ORBvl and LP inputs in Pyr-type-specific subnetworks. And we found that ORBvl and LP inputs drive mutual inhibition, mediated by local inhibitory neurons, between these two subnetworks in L5 of V1. Our study thus deepens understanding of the neuronal mechanisms involved in top-down modulation of visual processing and establishes that V2M and ACA inputs in L6 employ integrated processing that is distinct from the parallel processing of LP and ORBvl inputs in L5.”

4. We revised our introduction section to briefly introduce the parallel and integrated processing in visual information processing and to clarify our motivation in choosing to study V2M, ACA, ORBvl, and LP inputs (and the interactions among them). The revised text reads as follows (P. 3, Lines 2-31 and P. 4 Line 1):

“The visual system is hierarchically organized, with functionally related areas connecting to each other in specific laminar patterns (Coogan and Burkhalter, 1993; D'Souza et al., 2016; Felleman and Van Essen, 1991; Harris et al., 2019). Processing of various dimensions of the complex visual environment, such as color, depth, shape, and motion, occurs in parallel pathways starting from the retina and continues in specialized visual areas, followed by integration in higher-order areas to form a unified perception (Glickfeld and Olsen, 2017; Ibbotson and Meffin, 2020; Livingstone and Hubel, 1988; Nassi and Callaway, 2009; Seabrook et al., 2017; Shapley, 1990; Wassle, 2004). This involves both bottom-up inputs,

which flow from the retina to higher-order areas, and top-down modulation, where signals from higher-order areas adapt visual processing to meet the requirement of the current task (Glickfeld and Olsen, 2017; Livingstone and Hubel, 1988; Nassi and Callaway, 2009; Seabrook et al., 2017; Shapley, 1990; Wässle, 2004).

V1 is the initial cortical area for visual information processing; it receives condensed and parallel bottom-up signals from the retinogeniculo-cortical pathways, extracts relevant information, and further elaborates and integrates this information with top-down inputs from higher-order cortical and thalamic areas, contributing to the formation of a unified perceptual experience (Froudarakis et al., 2019; Glickfeld and Olsen, 2017; Ibbotson and Meffin, 2020; Niell and Scanziani, 2021; Seabrook et al., 2017). In more detail, thalamocortical (TC) bottom-up inputs from the dorsal lateral geniculate thalamic nucleus (dLGN) preferentially target the medial layer (L4), while corticocortical (CC) and TC top-down inputs target superficial and deep layers in V1 (Harris et al., 2019; Hunnicutt et al., 2014; Oh et al., 2014; Zingg et al., 2014).

Previous studies have identified multiple higher-order cortical and thalamic areas that provide top-down inputs to V1, such as the secondary visual cortex (V2), posterior parietal cortex (PTLp), retrosplenial cortex (RSP), anterior cingulate cortex (ACA), ventrolateral orbitofrontal cortex (ORBvl), and lateral dorsal (LD) and lateral posterior thalamic nuclei (LP) (Harris et al., 2019; Ma et al., 2021; Oh et al., 2014; Zhang et al., 2016). These top-down inputs convey a rich array of information, including attention (Debes and Dragoi, 2023; Hu et al., 2019; Zhang et al., 2014), expectation (Fiser et al., 2016; Leinweber et al., 2017), perceptual tasks (Liu et al., 2020; Norman et al., 2021), and motor commands (Huda et al., 2020; Kim et al., 2021).

The signals from these top-down inputs are differentially processed in V1 (D'Souza et al., 2016; Gilbert and Li, 2013; Liu et al., 2020; Moore and Zirnsak, 2017; Young et al., 2021; Zhang et al., 2014). For instance, V2 inputs are known to enhance the accuracy of visual information decoding in V1 neurons without altering the average response (Javadzadeh and Hofer, 2022). ACA and LP inputs increase V1 neuron response to task-relevant visual information (Hu et al., 2019; Zhang et al., 2014), whereas ORBvl inputs suppress V1 neuron responses to filter out irrelevant visual information (Liu et al., 2020). The diverse effects of these top-down modulations on visual processing imply sophisticated mechanisms within local circuits of V1, presumably controlled by distinct sets of excitatory and inhibitory neurons across different layers. However, knowledge about how these top-down inputs differentially engage excitatory and inhibitory neurons in V1 is quite limited, particularly with regard to their innervation patterns of neurons in deep layers.”

We have also revised the last paragraph in the introduction section. The revised text reads as follows (P. 4, Lines 12-32 and P. 5, Line 1):

“In this study, we profiled the layer- and cell-type-specific innervation patterns of multiple CC and TC top-down inputs in V1, including three CC inputs (V2M, ACA, and ORBv1) and one TC input (LP). We found distinct layer- and cell-type-specific innervation profiles for each top-down input, with profiles partially overlapping in superficial layers, bypassing L4, and clearly segregating in deep layers. Specifically, V2M and ACA inputs preferentially activate L6 Pyrs, while ORBv1 and LP inputs activate L5 Pyrs. We also characterized the layer-specificity of top-down inputs on inhibitory neurons, revealing that L1-INs are strongly activated in L1 and that VIP+ neurons are strongly activated in both superficial and deep layers, while PV+ and SST+ neurons are specifically activated in the deep layers. These results provide a valuable resource for the layer- and cell-type-specific organization of top-down inputs in V1. We subsequently investigated how these inputs interact within their strongest receptive layers, L5 and L6. Using independent optogenetic activation on the same brain slice, we found that V2M and ACA inputs converge on L6 Pyrs, whereas ORBv1 and LP inputs selectively activate two distinct types of L5 Pyrs: $\text{Pyr}_{\leftarrow\text{ORBv1}}$ and $\text{Pyr}_{\leftarrow\text{LP}}$ neurons, each characterized by specific electrophysiological properties and gene expression profiles. Retrograde tracing revealed that $\text{Pyr}_{\leftarrow\text{ORBv1}}$ neurons preferentially innervate subcortical areas and $\text{Pyr}_{\leftarrow\text{LP}}$ neurons innervate cortical areas, indicating parallel processing of the ORBv1 and LP inputs in Pyr-type-specific subnetworks. We also found that ORBv1 and LP inputs drive mutual inhibition, mediated by local inhibitory neurons, between these two subnetworks in L5 of V1. These findings deepen our understanding of neuronal mechanisms of top-down modulation of visual processing by revealing interactions of modulation signals within V1 local circuits.”

5. In the results section, we have added a main figure, revised a subheading, and added two paragraphs to introduce the results of the new experiment, in which independent optogenetic activation of V2M and ACA inputs revealed their convergence in L6 Pyrs. Additionally, we have revised the conclusion of this subsection accordingly. The revised subsection reads as follows (P. 7, Lines 14-31 and P. 8, Lines 1-17):

“Integrated processing of V2M and ACA inputs in L6 Pyrs versus parallel processing of LP and ORBv1 inputs in two distinct L5 Pyr populations

The examined CC and TC top-down inputs are strongest in the deep layers, with L6 being most responsive to V2M and ACA inputs, while L5 is most responsive to LP and ORBv1 inputs. Previous studies have reported that deep-layer Pyrs are a heterogeneous population, with functionally distinct subnetworks (Harris and Shepherd, 2015; Matho et

al., 2021; Tasic et al., 2018). To investigate the interactions among these top-down inputs in their most responsive layers, we employed independent optogenetic activation (Rindner et al., 2022). Specifically, for V2M and ACA inputs, we alternatively activated these inputs using ChR2 and ChrimsonR in the same brain slice and measured their input strengths in each recorded L6 Pyr. In each animal, we injected an AAV expressing ChR2-EYFP in the V2M (AAV-CaMKII α -ChR2-EYFP) and also injected an AAV expressing ChrimsonR-tdTomato (AAV-hSyn-ChrimsonR-tdTomato) in the ACA (Fig. 3a). Expression of ChR2-EYFP in the V2M and ChrimsonR-tdTomato in the ACA resulted in the expected bright green and red axonal fluorescence in L6 of V1 (Fig. 3b-d).

Activation of V2M inputs (10 pulses @ 10Hz, 488 nm) and ACA inputs (10 pulses @ 10Hz, 647 nm) elicited robust postsynaptic currents (EPSCs) in L6 Pyrs (Fig. 3e-g). The majority of recorded L6 Pyrs (57%) received inputs from both V2M and ACA, with 40% receiving only V2M inputs and with 3% not receiving inputs from either (Fig. 3h). These results show that CC top-down inputs from V2M and ACA converge on the same group of L6 Pyrs, indicating integrated processing of their top-down modulation signals in L6 of V1.

We next examined the innervation patterns of LP and ORBvl inputs in L5 Pyrs. For each animal, we injected an AAV expressing ChrimsonR-tdTomato in the LP (AAV-hSyn-ChrimsonR-tdTomato) and also injected an AAV expressing ChR2-EYFP in the ORBvl (AAV-CaMKII α -ChR2-EYFP) (Fig. 4a). Expression of ChrimsonR-tdTomato in the LP and ChR2-EYFP in the ORBvl resulted in bright red and green axonal fluorescence in L5 of V1, with LP inputs primarily in upper L5 (red) and ORBvl inputs in lower L5 (green) (Fig. 4b-d). Activation of LP inputs (10 pulses @ 10Hz, 647 nm) elicited robust postsynaptic currents (EPSCs) in about half of recorded L5 Pyrs (Pyr \leftarrow LP neurons); these neurons did not respond to activation of ORBvl inputs (10 pulses @ 10Hz, 488 nm) (Fig. 4e-g). Intriguingly, when we activated ORBvl inputs, the remaining half of L5 Pyrs displayed robust EPSCs (Pyr \leftarrow ORBvl neurons) (Fig. 4e-h). These results indicate that TC and CC top-down inputs from LP and ORBvl employ parallel processing strategies by selectively activating distinct L5 Pyr populations in V1. Thus, our results reveal examined CC and TC top-down inputs employ distinct processing strategies in their interactions, with integrated processing of V2M and ACA inputs in L6 versus parallel processing of LP and ORBvl inputs in L5.”

“Fig. 3 | V2M and ACA inputs converge on L6 Pyrs. a, Schematic of the viral strategy for independent optogenetic activation of V2M and ACA inputs. **b**, Fluorescent images showing the distribution of V2M (green) and ACA axons (red) in V1. Scale bars, 100 μ m. **c**, Normalized axon intensity of V2M (green) and ACA (red) axons as a function of L6 depth. 0% and 100% represent the upper and lower boundaries of L6. V2M, $n = 11$ slices; ACA, $n = 9$ slices. **d**, Summary of L6 depth for peak V2M (green) axon intensity and for peak ACA (red) axon intensity. $P = 0.21$, Wilcoxon sign-rank test. **e**, Schematic of the slice experiment examining independent optogenetic activation of V2M and ACA inputs. **f**, EPSCs evoked by independent activation of V2M and ACA inputs in L6 Pyrs. Blue dots, 488-nm light stimulation, 10 Hz, 1 ms, 3.5 mW. Red dots, 647-nm light stimulation, 10 Hz, 1 ms, 3.5 mW. Gray traces are individual trials, while purple and blue traces are averages of 5 trials. Scale bars, 100 ms, 50 pA. **g**, Normalized EPSC charge evoked by activation of V2M and ACA inputs in Pyr \leftarrow both (top) and Pyr \leftarrow V2M neurons (bottom). The circles indicate the means. $n = 5$ mice, 7 slices. Pyr \leftarrow both, $n = 37$ neurons, $P = 0.57$; Pyr \leftarrow V2M, $n = 26$ neurons, $P = 3 \times 10^{-39}$. *** $P < 0.001$; Paired t -test. **h**, Percentage (top) and distribution (bottom) of L6 Pyrs activated by V2M and ACA inputs. Purple, L6 Pyrs receiving both inputs; Blue, L6 Pyrs receiving only V2M inputs; gray, L6 Pyrs without any response to V2M or ACA inputs. Data are presented as the mean \pm SEM. Figure 3, Pannel a adapted from Petrucco, L. (2020). Mouse head schema. Zenodo.

<https://doi.org/10.5281/zenodo.3925903> under a CC BY license:
[https://creativecommons.org/licenses/by/4.0/.](https://creativecommons.org/licenses/by/4.0/)”

6. In the discussion section, we have revised the first paragraph. The revised text reads as follows (P. 14, Lines 18-29):

“In this study, we characterized the layer- and cell-type-specific organization of multiple CC and TC top-down inputs in V1, as well as their interactions in deep layers. Each top-down input to V1 engaged both excitatory and inhibitory neurons: CC and TC top-down inputs partially overlapped in superficial layers (L1 and L2/3), bypassed the medial layer (L4), and clearly segregated in deep layers (L5 and L6). Our datasets provide a valuable resource for understanding the neuronal mechanisms of top-down modulation of visual information processing. Further, we found that CC top-down inputs from the V2M and ACA undergo integrated processing in L6 Pyrs, whereas TC top-down inputs from the ORBv1 and LP undergo parallel processing within two L5 Pyr-type-specific subnetworks, separately mediating subcortical and intracortical V1 output channels. Notably, ORBv1 and LP inputs drive mutual inhibition between these two subnetworks in L5 of V1, potentially enabling dynamic gating of subnetwork functions in response to specific top-down modulation signals.”

We have also revised the paragraph discussing the different Pyr types engaged in the processing of top-down inputs. The revised text reads as follows (P. 15, Lines 20-30):

“Cortical Pyrs have been divided based on axonal projections into three main types: IT neurons preferentially innervating the cortex and striatum; PT neurons innervating subcortical areas; and corticothalamic neurons, which only innervate the thalamus (Harris and Shepherd, 2015). These types overlap in deep layers, with IT and PT in L5, and IT and corticothalamic in L6. Our study has revealed the specificity of various CC and TC top-down inputs engaging distinct Pyr types in deep layers of V1. V2M and ACA inputs converge on L6 Pyrs, suggesting integrated processing of these inputs that likely involves both IT and corticothalamic neurons. And we found that ORBv1 and LP inputs selectively activate PT (Pyr-ORBv1) and IT (Pyr-LP) neurons, each with distinct AP modes (BS spiking vs. RS spiking), indicating parallel processing of the information they convey in the PT and IT subnetworks.”

We would like to thank the reviewer for the excellent guidance about how to improve our study.

References

- Combe, C.L., and Gasparini, S. (2021). I(h) from synapses to networks: HCN channel functions and modulation in neurons. *Prog. Biophys. Mol. Biol.* *166*, 119-132.
- Coogan, T.A., and Burkhalter, A. (1993). Hierarchical organization of areas in rat visual cortex. *J. Neurosci.* *13*, 3749-3772.
- D'Souza, R.D., Meier, A.M., Bista, P., Wang, Q., and Burkhalter, A. (2016). Recruitment of inhibition and excitation across mouse visual cortex depends on the hierarchy of interconnecting areas. *Elife* *5*, e19332.
- Debes, S.R., and Dragoi, V. (2023). Suppressing feedback signals to visual cortex abolishes attentional modulation. *Science* *379*, 468-473.
- Felleman, D.J., and Van Essen, D.C. (1991). Distributed hierarchical processing in the primate cerebral cortex. *Cereb. Cortex* *1*, 1-47.
- Fiser, A., Mahringer, D., Oyibo, H.K., Petersen, A.V., Leinweber, M., and Keller, G.B. (2016). Experience-dependent spatial expectations in mouse visual cortex. *Nat. Neurosci.* *19*, 1658-1664.
- Froudarakis, E., Fahey, P.G., Reimer, J., Smirnakis, S.M., Tehovnik, E.J., and Tolias, A.S. (2019). The Visual Cortex in Context. *Annu. Rev. Vis. Sci.* *5*, 317-339.
- Gilbert, C.D., and Li, W. (2013). Top-down influences on visual processing. *Nat. Rev. Neurosci.* *14*, 350-363.
- Glickfeld, L.L., and Olsen, S.R. (2017). Higher-Order Areas of the Mouse Visual Cortex. *Annu. Rev. Vis. Sci.* *3*, 251-273.
- Harris, J.A., Mihalas, S., Hirokawa, K.E., Whitesell, J.D., Choi, H., Bernard, A., Bohn, P., Caldejon, S., Casal, L., Cho, A., *et al.* (2019). Hierarchical organization of cortical and thalamic connectivity. *Nature* *575*, 195-202.
- Harris, K.D., and Shepherd, G.M. (2015). The neocortical circuit: themes and variations. *Nat. Neurosci.* *18*, 170-181.
- Hu, F., Kamigaki, T., Zhang, Z., Zhang, S., Dan, U., and Dan, Y. (2019). Prefrontal Corticotectal Neurons Enhance Visual Processing through the Superior Colliculus and Pulvinar Thalamus. *Neuron* *104*, 1141-1152.
- Huda, R., Sipe, G.O., Breton-Provencher, V., Cruz, K.G., Pho, G.N., Adam, E., Gunter, L.M., Sullins, A., Wickersham, I.R., and Sur, M. (2020). Distinct prefrontal top-down circuits differentially modulate sensorimotor behavior. *Nat. Commun.* *11*, 6007.
- Hunnicut, B.J., Long, B.R., Kusefoglu, D., Gertz, K.J., Zhong, H., and Mao, T. (2014). A comprehensive thalamocortical projection map at the mesoscopic level. *Nat. Neurosci.* *17*, 1276-1285.
- Ibbotson, M.R., and Meffin, H. (2020). Visual Information Processing. In *The Senses: A Comprehensive Reference*, pp. 36-53.
- Ibrahim, L.A., Huang, S., Fernandez-Otero, M., Sherer, M., Qiu, Y., Vemuri, S., Xu, Q., Machold, R., Pouchelon, G., Rudy, B., *et al.* (2021). Bottom-up inputs are required for establishment of top-down connectivity onto cortical layer 1 neurogliaform cells. *Neuron* *109*, 3473-3485.
- Ibrahim, L.A., Mesik, L., Ji, X.Y., Fang, Q., Li, H.F., Li, Y.T., Zingg, B., Zhang, L.I., and Tao, H.W. (2016).

Cross-Modality Sharpening of Visual Cortical Processing through Layer-1-Mediated Inhibition and Disinhibition. *Neuron* *89*, 1031-1045.

Javadzadeh, M., and Hofer, S.B. (2022). Dynamic causal communication channels between neocortical areas. *Neuron* *110*, 2470-2483.

Ji, X.Y., Zingg, B., Mesik, L., Xiao, Z., Zhang, L.I., and Tao, H.W. (2016). Thalamocortical Innervation Pattern in Mouse Auditory and Visual Cortex: Laminar and Cell-Type Specificity. *Cereb. Cortex* *26*, 2612-2625.

Kim, J.H., Ma, D.H., Jung, E., Choi, I., and Lee, S.H. (2021). Gated feedforward inhibition in the frontal cortex releases goal-directed action. *Nat. Neurosci.* *24*, 1452-1464.

Kim, M.H., Znamenskiy, P., Iacaruso, M.F., and Mrsic-Flogel, T.D. (2018). Segregated Subnetworks of Intracortical Projection Neurons in Primary Visual Cortex. *Neuron* *100*, 1313-1321.

Leinweber, M., Ward, D.R., Sobczak, J.M., Attinger, A., and Keller, G.B. (2017). A Sensorimotor Circuit in Mouse Cortex for Visual Flow Predictions. *Neuron* *95*, 1420-1432.

Liu, D., Deng, J., Zhang, Z., Zhang, Z.Y., Sun, Y.G., Yang, T., and Yao, H. (2020). Orbitofrontal control of visual cortex gain promotes visual associative learning. *Nat. Commun.* *11*, 2784.

Livingstone, M., and Hubel, D. (1988). Segregation of Form, Color, Movement, and Depth: Anatomy, Physiology, and Perception. *Science* *240*, 740-749.

Ma, G., Liu, Y., Wang, L., Xiao, Z., Song, K., Wang, Y., Peng, W., Liu, X., Wang, Z., Jin, S., *et al.* (2021). Hierarchy in sensory processing reflected by innervation balance on cortical interneurons. *Sci. Adv.* *7*, eabf5676

Matho, K.S., Huilgol, D., Galbavy, W., He, M., Kim, G., An, X., Lu, J., Wu, P., Di Bella, D.J., Shetty, A.S., *et al.* (2021). Genetic dissection of the glutamatergic neuron system in cerebral cortex. *Nature* *598*, 182-187.

Moore, T., and Zirnsak, M. (2017). Neural Mechanisms of Selective Visual Attention. *Annu. Rev. Psychol.* *68*, 47-72.

Nagai, J., Rajbhandari, A.K., Gangwani, M.R., Hachisuka, A., Coppola, G., Masmanidis, S.C., Faselow, M.S., and Khakh, B.S. (2019). Hyperactivity with Disrupted Attention by Activation of an Astrocyte Synaptogenic Cue. *Cell* *177*, 1280-1292.

Naskar, S., Qi, J., Pereira, F., Gerfen, C.R., and Lee, S. (2021). Cell-type-specific recruitment of GABAergic interneurons in the primary somatosensory cortex by long-range inputs. *Cell Rep.* *34*, 108774.

Nassi, J.J., and Callaway, E.M. (2009). Parallel processing strategies of the primate visual system. *Nat. Rev. Neurosci.* *10*, 360-372.

Niell, C.M., and Scanziani, M. (2021). How Cortical Circuits Implement Cortical Computations: Mouse Visual Cortex as a Model. *Annu. Rev. Neurosci.* *44*, 517-546.

Norman, K.J., Riceberg, J.S., Koike, H., Bateh, J., McCraney, S.E., Caro, K., Kato, D., Liang, A., Yamamuro, K., Flanigan, M.E., *et al.* (2021). Post-error recruitment of frontal sensory cortical projections promotes attention in mice. *Neuron* *109*, 1202-1213.

Oh, S.W., Harris, J.A., Ng, L., Winslow, B., Cain, N., Mihalas, S., Wang, Q., Lau, C., Kuan, L., Henry, A.M., *et al.* (2014). A mesoscale connectome of the mouse brain. *Nature* *508*, 207-214.

Paniccia, J.E., Vollmer, K.M., Green, L.M., Grant, R.I., Winston, K.T., Buchmaier, S., Westphal, A.M., Clarke, R.E., Doncheck, E.M., Bordieanu, B., *et al.* (2023). Restoration of a paraventricular thalamo-accumbal behavioral suppression circuit prevents reinstatement of heroin seeking. *Neuron*. S0896-6273(23)00928-5.

Petreanu, L., Mao, T., Sternson, S.M., and Svoboda, K. (2009). The subcellular organization of neocortical excitatory connections. *Nature* *457*, 1142-1145.

Rindner, D.J., Proddatur, A., and Lur, G. (2022). Cell-type-specific integration of feedforward and feedback synaptic inputs in the posterior parietal cortex. *Neuron* *110*, 3760-3773.

Seabrook, T.A., Burbridge, T.J., Crair, M.C., and Huberman, A.D. (2017). Architecture, Function, and Assembly of the Mouse Visual System. *Annu. Rev. Neurosci.* *40*, 499-538.

Shapley, R. (1990). Visual sensitivity and parallel retinocortical channels. *Annu. Rev. Psychol.* *41*, 635-658.

Tasic, B., Yao, Z., Graybiel, L.T., Smith, K.A., Nguyen, T.N., Bertagnolli, D., Goldy, J., Garren, E., Economo, M.N., Viswanathan, S., *et al.* (2018). Shared and distinct transcriptomic cell types across neocortical areas. *Nature* *563*, 72-78.

Wassle, H. (2004). Parallel processing in the mammalian retina. *Nat. Rev. Neurosci.* *5*, 747-757.

Yao, S., Wang, Q., Hirokawa, K.E., Ouellette, B., Ahmed, R., Bomben, J., Brouner, K., Casal, L., Caldejon, S., Cho, A., *et al.* (2023). A whole-brain monosynaptic input connectome to neuron classes in mouse visual cortex. *Nat. Neurosci.* *26*, 350-364.

Yao, Z., van Velthoven, C.T.J., Nguyen, T.N., Goldy, J., Sedenio-Cortes, A.E., Baftizadeh, F., Bertagnolli, D., Casper, T., Chiang, M., Crichton, K., *et al.* (2021). A taxonomy of transcriptomic cell types across the isocortex and hippocampal formation. *Cell* *184*, 3222-3241.

Young, H., Belbut, B., Baeta, M., and Petreanu, L. (2021). Laminar-specific cortico-cortical loops in mouse visual cortex. *Elife* *10*, e59551.

Zhang, S., Xu, M., Chang, W.C., Ma, C., Hoang Do, J.P., Jeong, D., Lei, T., Fan, J.L., and Dan, Y. (2016). Organization of long-range inputs and outputs of frontal cortex for top-down control. *Nat. Neurosci.* *19*, 1733-1742.

Zhang, S., Xu, M., Kamigaki, T., Hoang Do, J.P., Chang, W.C., Jenvay, S., Miyamichi, K., Luo, L., and Dan, Y. (2014). Selective attention. Long-range and local circuits for top-down modulation of visual cortex processing. *Science* *345*, 660-665.

Zingg, B., Hintiryan, H., Gou, L., Song, M.Y., Bay, M., Bienkowski, M.S., Foster, N.N., Yamashita, S., Bowman, I., Toga, A.W., *et al.* (2014). Neural networks of the mouse neocortex. *Cell* *156*, 1096-1111.

REVIEWERS' COMMENTS

Reviewer #1 (Remarks to the Author):

Thank the authors for putting in the work to revise the manuscript, including adding new experiments on V2M and ACA inputs. I agree that the paper can serve as a valuable resource for future investigations on the V1 function. As of now, I have no future questions or concerns.

Reviewer #2 (Remarks to the Author):

The authors adequately addressed my concern. I have no further comments.

Reviewer #3 (Remarks to the Author):

the author have successfully addressed my concerns with new experiments and analysis.

I have no further questions.

Reviewer #4 (Remarks to the Author):

In this revised manuscript, Liu, Zhang, and colleagues adequately addressed most of my previous concerns except the following two.

1. Regarding the impact of variable expression of Chr2 on the results, the authors stated that "For the experiments in Fig. 1 and Fig. 2, we recorded neurons simultaneously in each layer in at least 5 mice in each group. For the majority of recorded animals, we recorded neurons simultaneously in three or more layers." This method section needs to be clarified with more details, as it is puzzling when comparing to the distribution of recorded neurons in different layers. For example, in Fig1k, there are 23, 12, 124, and 25 neurons in L2/3, L4, L5, and L6, respectively. It is unclear how "for the majority of recorded animals, we recorded neurons simultaneously in three or more layers" gives arise to such an uneven distribution.

This also applies to other panels in Figure 1. The authors stated that “we carefully controlled several parameters to minimize variations caused by ChR2 expression. These included standardizing the titer, injection volume, and expression time of the AAV vector in each experimental series, ensuring consistent ChR2 expression levels”, which are good practices, but the outcome of these practices (i.e., the ChR2 expression levels) was not measured or shown to demonstrate that they have achieved consistent ChR2 expression levels in these experiments. The important question is if the results from those neurons simultaneously recorded in each layer in the same slice are consistent with the overall results reported in the figures. This will help address the concern how the variability of ChR2 expression affects the differences among layers.

2. Were statistical comparisons done for the results of different interneuron types in different layers in Figure 2 and Extended Data Figure 4? Was none of the differences reach statistical significance?

Reviewer #1 (reviewer' s comments in italic):

Thank the authors for putting in the work to revise the manuscript, including adding new experiments on V2M and ACA inputs. I agree that the paper can serve as a valuable resource for future investigations on the VI function. As of now, I have no future questions or concerns.

Reviewer #2 (reviewer' s comments in italic):

The authors adequately addressed my concern. I have no further comments.

Reviewer #3 (reviewer' s comments in italic):

The authors have successfully addressed my concerns with new experiments and analysis.

I have no further questions.

Reviewer #4 (reviewer' s comments in italic):

In this revised manuscript, Liu, Zhang, and colleagues adequately addressed most of my previous concerns except the following two.

1. Regarding the impact of variable expression of ChR2 on the results, the authors stated that “For the experiments in Fig. 1 and Fig. 2, we recorded neurons simultaneously in each layer in at least 5 mice in each group. For the majority of recorded animals, we recorded neurons simultaneously in three or more layers.” This method section needs to be clarified with more details, as it is puzzling when comparing to the distribution of recorded neurons in different layers. For example, in Fig1k, there are 23, 12, 124, and 25 neurons in L2/3, L4, L5, and L6, respectively. It is unclear how “for the majority of recorded animals, we recorded neurons simultaneously in three or more layers” gives arise to such an uneven distribution. This also applies to other panels in Figure 1.

The authors stated that “we carefully controlled several parameters to minimize variations caused by ChR2 expression. These included standardizing the titer, injection volume, and expression time of the AAV vector in each experimental series, ensuring consistent ChR2 expression levels”, which are good practices, but the outcome of these practices (i.e., the ChR2 expression levels) was not measured or shown to demonstrate that they have achieved consistent ChR2 expression levels in these

experiments. The important question is if the results from those neurons simultaneously recorded in each layer in the same slice are consistent with the overall results reported in the figures. This will help address the concern how the variability of Chr2 expression affects the differences among layers.

RESPONSE:

We thank the reviewer for bringing this issue to our attention. We have now added a supplementary table to provide the detailed information on the number of mice recorded with different numbers of layers (Supplementary table 13). Given our interest in Pyr-type classification for the most responsive layers, we collected additional data from these layers (L6 for V2M and ACA inputs, and L5 for ORBvl and LP inputs). In Fig. 1, where we compared differences in raw EPSP amplitudes recorded from Pyrs across layers, we included data from some mice where only Pyrs in the most responsive layers were recorded, which caused a biased data distribution.

Following the reviewer’s guidance, we reanalyzed our data in Fig.1 and generated plots of the normalized EPSP amplitude for each top-down input (see revised Supplementary Fig. 5, original Extended Data Fig. 3) to address potential influences from the variability of Chr2 expression. In Supplementary Fig. 5, the data are for Pyrs recorded in at least two layers of the same brain slice. By excluding data from mice where only Pyrs in the most responsive layers were recorded, the bias in data distribution is reduced. For V2M and ACA inputs, we normalized the EPSP amplitude of each recorded neuron to the averaged EPSP amplitude of L6 Pyrs in the same slice. For ORBvl and LP inputs—considering that the Pyrs in their most responsive layer (L5) can be divided into distinct subtypes according to their responses to these inputs (as shown in Fig. 4)—we normalized the EPSP amplitude against $Py_{T \leftarrow ORBvl}$ and $Py_{T \leftarrow LP}$ neurons, respectively. The results we obtained using this normalization approach are consistent with the differences in input strengths across layers reported in the originally submitted manuscript.

We have revised the caption for Supplementary fig.5 to clarify our exclusion of data from mice for which only Pyrs from the most responsive layers were recorded. The updated caption states: “The included data are from Pyrs recorded from at least two layers in the same brain slice, including the layer with the strongest response (L6 for V2M and ACA inputs, and L5 for ORBvl and LP inputs).”

Number of mice recorded with different numbers of layers in Fig.1				
Simultaneously recorded layer number	V2M input (mice)	ACA input (mice)	ORBvl input (mice)	LP input (mice)

4 layers	5	5	5	7
3 layers	10	9	5	8
2 layers	2	2	2	4
1 layer	0	1	6	5

“Supplementary table 13. Number of mice recorded with different numbers of layers in Fig.1.”

“Supplementary Fig. 5 | Layer specificity of CC and TC top-down inputs based on normalized EPSP amplitude. **a**, Normalized EPSP amplitudes evoked by optogenetic activation of V2M inputs in Pyrs across the indicated layers. Left, box plot showing distribution of EPSP amplitude from Pyrs in the indicated layers. Edges, 25th and 75th percentiles; central line, median; whiskers, 1.5× the interquartile range of the edges. Circles are outliers, defined as values more than 3 times the interquartile range (IQR) from the median. V2M inputs activated the Pyrs in L2/3 and L6, with the strongest input strength in L6 (significant differences were observed across layers $F(3,199) = 20.7$, $P = 1 \times 10^{-11}$, one-way ANOVA; L6 vs. other layers, $P < 0.002$, Tukey’s post hoc test). Right, matrix of normalized EPSP amplitude across layers of V1 for V2M inputs. **b**, Similar to **a**, but for ACA inputs. ACA inputs activated the Pyrs in L2/3 and L6, with stronger input strength in L6 (significant differences were observed across layers, $F(3,152) = 10.2$, $P = 4 \times 10^{-6}$,

one-way ANOVA; L6 vs. L4 and L5, $P < 0.004$; L6 vs. L2/3, $P = 0.87$, Tukey's post hoc test). **c**, Similar to **a**, but for ORBvl inputs. ORBvl inputs only activated the Pyrs in L5 (significant differences were observed across layers, $F(3,101) = 13.4$, $P = 2 \times 10^{-7}$, one-way ANOVA; L5 vs. other layers, $P < 5 \times 10^{-4}$, Tukey's post hoc test). **d**, Similar to **a**, but for LP inputs. LP inputs activated the Pyrs in L2/3, L5, and L6, with the strongest input strength in L5 (significant differences were observed across layers, $F(3,204) = 15.8$, $P = 3 \times 10^{-9}$, one-way ANOVA; L5 vs. other layers, $P < 2 \times 10^{-5}$, Tukey's post hoc test). The number of neurons in each group is indicated by the numbers displayed in the figure. The EPSP amplitudes of V2M and ACA inputs were normalized to the averaged L6 Pyr EPSP amplitude recorded the same day. Given the observed heterogeneity in L5 Pyr subtypes responding to ORBvl and LP inputs, we normalized the EPSP amplitudes of ORBvl and LP inputs to the average for Pyr \leftarrow ORBvl and for Pyr \leftarrow LP neurons. The included data are from Pyrs recorded from at least two layers in the same brain slice, including the layer with the strongest response (L6 for V2M and ACA inputs, and L5 for ORBvl and LP inputs). Source data are provided as a Source Data file."

2. Were statistical comparisons done for the results of different interneuron types in different layers in Figure 2 and Extended Data Figure 4? Was none of the differences reach statistical significance?

RESPONSE:

Following this guidance, we have now performed statistical analyses for raw EPSP data in Fig. 2 and normalized EPSP data in revised Supplementary fig. 6 (original Extended Data Fig. 4), including:

1. A one-way ANOVA to assess single effects across layers (L2/3, L4, L5, and L6) and cell types (PV+, SST+, and VIP+).
2. Tukey's HSD tests for pairwise comparisons.

The detailed parameters for the ANOVA and Tukey's HSD tests are included in newly revised supplementary tables (Supplementary Tables 4 and 5).

Supplementary Table 4 presents analyses based on both raw and normalized EPSP amplitudes across different layers for the same type of neurons. The results from these comparisons are consistent: for raw data, V2M inputs showed 5 significant differences, ACA inputs showed 8, ORBvl inputs showed 4, and LP inputs showed 4. Normalized data similarly showed 5, 8, 3, and 3 significant differences, respectively, for these inputs. Notably, analyses based on both raw and normalized EPSP amplitudes indicate that LP

inputs to PV+ neurons are strongest in L5, surpassing those in L2/3, L4, and L6, findings supporting a role for PV+ neurons in inhibiting L5 Pyrs when activated by LP inputs.

Supplementary Table 5 presents analyses based on both raw and normalized EPSP amplitudes for different types of neurons within the same layer. Analyses of both raw and normalized data show similar results: for raw data, V2M inputs showed 3 significant differences, ACA inputs showed 4, ORBvl inputs showed 0, and LP inputs showed 8. Normalized data similarly revealed 0, 2, 0, 8 significant differences, respectively, for these inputs. Notably, analyses of both raw and normalized EPSP amplitudes show that LP inputs to PV+ neurons are significantly weaker than LP inputs to VIP+ neurons in layers L2/3, L4, and L6. However, in layer L5, LP inputs to both PV+ and VIP+ neurons are strong (and with no significant differences), supporting that PV+ neurons inhibit L5 Pyrs after activation by LP inputs.

The tables below provide a simplified overview of the results from our multiple comparisons, adapted from Supplementary Tables 4 and 5.

Multiple comparisons (Tukey HSD)							
V2M inputs							
		VIP+		PV+		SST+	
		Raw_EPSP	Norm_EPSP	Raw_EPSP	Norm_EPSP	Raw_EPSP	Norm_EPSP
Layer2/3	Layer4	ns	ns	ns	ns	ns	ns
	Layer5	ns	ns	ns	ns	ns	ns
	Layer6	**	**	ns	ns	ns	ns
Layer4	Layer5	ns	ns	ns	ns	ns	ns
	Layer6	***	***	*	*	ns	ns
Layer5	Layer6	***	**	**	**	ns	ns
ACA inputs							
		VIP+		PV+		SST+	
		Raw_EPSP	Norm_EPSP	Raw_EPSP	Norm_EPSP	Raw_EPSP	Norm_EPSP
Layer2/3	Layer4	ns	ns	ns	ns	ns	ns
	Layer5	ns	ns	ns	ns	ns	ns
	Layer6	ns	ns	***	**	***	*
Layer4	Layer5	ns	ns	ns	ns	ns	ns
	Layer6	*	**	***	***	***	**
Layer5	Layer6	*	*	***	***	***	***
ORBvl inputs							
		VIP+		PV+		SST+	
		Raw_EPSP	Norm_EPSP	Raw_EPSP	Norm_EPSP	Raw_EPSP	Norm_EPSP
Layer2/3	Layer4	ns	ns	ns	ns	ns	ns
	Layer5	ns	ns	*	*	ns	ns
	Layer6	ns	ns	ns	ns	ns	ns
Layer4	Layer5	ns	ns	*	*	ns	ns
	Layer6	ns	ns	ns	Ns	ns	ns
Layer5	Layer6	*	*	ns	Ns	*	ns
LP inputs							
		VIP+		PV+		SST+	
		Raw_EPSP	Norm_EPSP	Raw_EPSP	Norm_EPSP	Raw_EPSP	Norm_EPSP
Layer2/3	Layer4	ns	ns	ns	ns	ns	ns
	Layer5	ns	ns	***	***	ns	ns
	Layer6	ns	ns	ns	ns	ns	ns
Layer4	Layer5	ns	ns	***	***	ns	ns
	Layer6	*	ns	ns	ns	ns	ns
Layer5	Layer6	ns	ns	***	***	ns	ns

Reviewer Table. 1. Comparison of CC and TC top-down input strengths in V1 interneurons across layers, adjusted from Supplementary Table 4.

*P < 0.05, **P < 0.01, ***P < 0.001.

Multiple comparisons (Tukey HSD)								
V2M inputs								
	L2/3		L4		L5		L6	
	Raw	Norm	Raw	Norm	Raw	Norm	Raw	Norm
VIP+ vs. PV+	ns	ns	ns	ns	ns	ns	ns	ns
VIP+ vs. SST+	ns	ns	ns	ns	*	ns	***	ns
PV+ vs. SST+	ns	ns	ns	ns	ns	ns	*	ns
ACA inputs								
	L23		L4		L5		L6	
	Raw	Norm	Raw	Norm	Raw	Norm	Raw	Norm
VIP+ vs. PV+	***	ns	ns	ns	***	***	ns	ns
VIP+ vs. SST+	***	ns	ns	ns	***	***	ns	ns
PV+ vs. SST+	ns	ns	ns	ns	ns	ns	ns	ns
ORBvl inputs								
	L23		L4		L5		L6	
	Raw	Norm	Raw	Norm	Raw	Norm	Raw	Norm
VIP+ vs. PV+	ns	ns	ns	ns	ns	ns	ns	ns
VIP+ vs. SST+	ns	ns	ns	ns	ns	ns	ns	ns
PV+ vs. SST+	ns	ns	ns	ns	ns	ns	ns	ns
LP inputs								
	L23		L4		L5		L6	
	Raw	Norm	Raw	Norm	Raw	Norm	Raw	Norm
VIP+ vs. PV+	***	***	***	***	ns	ns	***	***
VIP+ vs. SST+	***	***	***	***	***	***	***	***
PV+ vs. SST+	ns	ns	ns	ns	***	***	ns	ns

Reviewer Table. 2. Comparison of CC and TC top-down input strengths among different types of interneurons within the same layer of V1, adjusted from Supplementary Table 5.

*P < 0.05, **P < 0.01, ***P < 0.001.